# Controllable 3D Molecular Generation for Structure-Based Drug Design Through Bayesian Flow Networks and Gradient Integration

**Seungyeon Choi**[1], **Hwanhee Kim**[1], **Chihyun Park**[2,3], **Dahyeon Lee**[3], **Seungyong Lee**[1]
**Yoonju Kim**[1], **Hyoungjoon Park**[1], **Sein Kwon**[1], **Youngwan Jo**[1], **Sanghyun Park**[1†]
[1] Yonei University [2] UBLBio [3] Kangwon National University
{tmddus1553,sanghyun}@yonsei.ac.kr

## Abstract

Recent advances in Structure-based Drug Design (SBDD) have leveraged generative models for 3D molecular generation, predominantly evaluating model performance by binding affinity to target proteins. However, practical drug discovery necessitates high binding affinity along with synthetic feasibility and selectivity, critical properties that were largely neglected in previous evaluations. To address this gap, we identify fundamental limitations of conventional diffusion-based generative models in effectively guiding molecule generation toward these diverse pharmacological properties. We propose CBYG, a novel framework extending Bayesian Flow Network into a gradient-based conditional generative model that robustly integrates property-specific guidance. Additionally, we introduce a comprehensive evaluation scheme incorporating practical benchmarks for binding affinity, synthetic feasibility, and selectivity, overcoming the limitations of conventional evaluation methods. Extensive experiments demonstrate that our proposed CBYG framework significantly outperforms baseline models across multiple essential evaluation criteria, highlighting its effectiveness and practicality for real-world drug discovery applications.

## 1 Introduction

In Structure-based Drug Design (SBDD), generative models capable of designing 3D molecules that selectively bind target proteins have emerged as essential tools in drug discovery [3, 50]. Initial approaches primarily utilized voxel-grid representations [33], evolving through autoregressive architectures [34, 43], to recent high-performing non-autoregressive, diffusion-based methods [20, 21, 19]. Despite significant progress, practical drug discovery requires more than just binding affinity; viable drug candidates must also satisfy critical pharmacological constraints such as synthetic feasibility and selectivity [29, 26]. To address this, recent diffusion-based studies have incorporated gradient-based guidance strategies for enhanced property control during generation [11, 22].

However, it remains unclear whether diffusion-based guidance represents an optimal and practically reliable strategy for controllable molecule generation. Moreover, widely used evaluation metrics in existing SBDD research have rarely been rigorously examined regarding their adequacy in assessing realistic drug candidate properties. In this study, we systematically investigate the theoretical and practical limitations of diffusion-based guidance strategies and conventional evaluation metrics. To overcome these problems, we introduce CBYG (Controllable Bayesian Flow Network with Integrated Guidance), an extended Bayesian Flow Network [18] framework utilizing gradient-based conditional generation, accompanied by comprehensive evaluation benchmarks targeting synthetic feasibility, selectivity, and binding affinity. Our contributions provide essential insights and methodological

---

†Correspondence to: sanghyun@yonsei.ac.kr.

39th Conference on Neural Information Processing Systems (NeurIPS 2025).

advancements toward more robust, practical, and effective AI-driven molecular generation for real-world drug discovery applications.

## 2   Rethinking of 3D Molecule Modeling in SBDD

**Problem of guidance in hybrid modalities.**
In SBDD, generating molecules with desired properties that bind to target proteins is a primary objective. To this end, employing guidance strategies with external predictors within diffusion models is a considerable approach for property-driven sampling [22]. However, the hybrid nature of 3D molecular data, which comprises continuous Cartesian coordinates (typically modeled with Gaussian distributions) and categorical atom types (modeled with categorical distributions), presents significant limitations to the conventional application of guidance. The primary challenges are as follows:

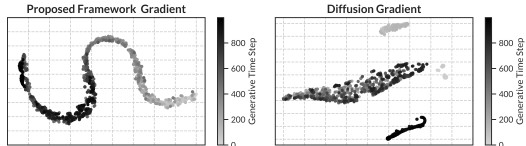

Figure 1: Gradient trajectory for target properties throughout the Generative Process for Proposed model and diffusion model. Further Interpretation on these results is provided in Section 6.4.

1. *Difficulty in Capturing Interactions:* As continuous coordinates and categorical atom types are sampled from fundamentally different types of distributions, the guidance mechanism may fail to accurately reflect the interactions between them [35, 46]. This can lead to a loss of crucial chemical context during the generation process.

2. *Ineffective and Unstable Categorical Variable Guidance:* Applying gradient-based guidance to categorical variables in diffusion models is inherently problematic. The argmax operation in the reverse process [19, 20, 21], coupled with the non-differentiability of categorical distributions, renders guidance signals either ineffective or a source of instability, while workaround strategies often introduce unnatural representations and increase model complexity, as shown in Figure 1.

3. *Loss of Chemical and Structural Validity:* Direct injection of gradients calculated during the denoising process into the sample space can readily compromise the chemical and structural validity of 3D molecules, which are highly sensitive to numerical perturbations. This complicates property control and can result in the generation of unstable molecular structures.

Recent studies have begun exploring Bayesian Flow Network (BFN) [18] as a solution to these issues in 3D molecular generation [46, 37]; however, these typically treat BFN as independent generative frameworks distinct from diffusion models, leaving gradient-based conditional generation (guidance) underexplored both theoretically and practically. Therefore, systematic theoretical clarification and practical expansion of gradient guidance mechanisms within BFN, particularly in comparison with well-established diffusion-based guidance strategies, remain essential research directions. A detailed discussion on this point is provided in the Appendix J.

**Necessity of Posterior Sampling in Guidance.**   In gradient-based generative frameworks such as diffusion models, conditional generation typically leverages a posterior conditioned on labels (attributes) $\mathbf{l}$, known as the conditional score function $\nabla_{\mathbf{x}_t} p(\mathbf{l} \mid \mathbf{x}_t)$, which is learned via a dedicated neural network [11, 45, 22]. While effective in image domains, where intermediate noisy states remain semantically informative, this approach is unsuitable for 3D molecular generation because noisy intermediate structures lack chemical validity and making reliable attribute prediction difficult, as shown in Figure 2. Recent studies have proposed posterior sampling methods utilizing predicted final states ($\mathbf{x}_0$) to address this challenge [7, 23]; however, these methods focus primarily

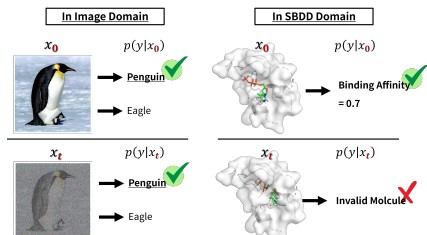

Figure 2: A comparative illustration of attribute conditioning using $p(\mathbf{l} \mid \mathbf{x}_t)$ versus $p(\mathbf{l} \mid \mathbf{x}_0)$ in the image and SBDD domains.

on general conditional molecular generation tasks, not specifically on structure-based drug design (SBDD). Furthermore, existing methods unnaturally discretize categorical atom types into continuous variables and neglect inherent prediction uncertainties of the final state. Thus, further research and methodological refinements are required to effectively employ $\mathbf{x}_0$-based guidance for SBDD tasks. A detailed discussion on this subject is provided in the Appendix J.

**Need for Selectivity dataset and various evaluation metrics**. Previous research in SBDD has predominantly relied on binding affinity measurements, typically using AutoDock Vina [12], to evaluate generated 3D molecules, introducing potential evaluation biases. To enhance reliability, incorporating diverse docking algorithms for affinity evaluation is essential.

Another key consideration is Synthetic Accessibility (SA) scores [14], combining structural complexity and fragment contributions into a single numerical value between 0 and 1, have frequently been employed to assess synthetic feasibility. However, high SA scores often do not guarantee practical synthetic routes, highlighting a critical gap in accurately evaluating real-world synthesizability.

Moreover, selectivity (ensuring that molecules bind specifically to target proteins without off-target interactions) is equally important as affinity [47, 26]. Poor selectivity can cause unwanted side effects, and while recent studies have proposed diffusion-based guidance methods [17, 6], these rely heavily on pretrained classifiers predicting true binding molecules. The widely used CrossDocked2020 [15] dataset, initially intended for general docking research, is unsuitable for selectivity evaluations without substantial additional docking computations. Furthermore, unclear criteria for identifying true binders and significant data imbalance hinder obtaining reliable guidance signals. Consequently, establishing biologically relevant benchmark datasets and developing effective controllable generation strategies for selectivity is urgently required. A detailed discussion on this subject is provided in the Appendix J.

The **Contributions** of this research addressing the aforementioned points are as follows:

- We introduce a novel approach that integrates BFN with diffusion models from a guidance perspective to circumvent the limitations of diffusion guidance in SBDD. This approach explicitly formulates a gradient-based guidance mechanism within a Bayesian update process and rigorously establishes its theoretical foundations.
- We provide a comprehensive analysis of how injecting guidance into generative models affects posterior sampling–based predictions and uncertainty in SBDD. This study offers new insights into the interplay between guided generation and uncertainty quantification.
- We revisit the limitations of existing evaluation metrics in SBDD and propose a new set of essential metrics for evaluating practical molecular generative models, including binding affinity, Synthetic Accessibility (SA), and selectivity.
- Through extensive and comprehensive experiments, we demonstrate that the proposed framework CBYG outperforms existing baseline models by a substantial margin. Notably, it achieves state-of-the-art performance under both conventional and newly introduced evaluation criteria.

## 3 Background

A comprehensive overview of related work pertinent to this study is provided in the Appendix C.

### 3.1 Bayesian Flow Network in 3D Molecule Modeling

Bayesian Flow Network (BFN) [18] synthesize data through iterative Bayesian refinements in a unified parameter space, distinct from diffusion models which operate directly in data space. Unlike conventional methods that separately estimate distributions for continuous atomic coordinates and categorical atom types, BFN jointly model both modalities within a single, tractable distribution (typically Gaussian). Formally, the data distribution $p_\phi(\mathbf{m})$ is approximated by progressively refining parameters $\boldsymbol{\theta}$ via Bayesian updates:

$$p_\phi(\mathbf{m}) = \int p_\phi(\mathbf{m} \mid \boldsymbol{\theta}_n) p(\boldsymbol{\theta}_0) \prod_{i=1}^{n} p_{\mathtt{U}}(\boldsymbol{\theta}_i \mid \boldsymbol{\theta}_{i-1}; \alpha_i) d\boldsymbol{\theta}_{1:n} \tag{1}$$

Each iterative refinement involves three transition kernels:

**Sender distribution** $p_{\mathtt{S}}(\mathbf{y}_i \mid \mathbf{m}; \alpha_i)$: injects controlled noise into the molecular representation, enabling gradual data transformations.

**Output distribution** $p_{\mathtt{O}}(\hat{\mathbf{m}} \mid \boldsymbol{\theta}_{i-1}; t_i)$: uses a neural network $\Psi$ to reconstruct molecular features based on previous parameter states, ensuring coherence despite noise.

**Receiver distribution** $p_{\mathtt{R}}(\mathbf{y}_i \mid \boldsymbol{\theta}_{i-1}; t_i, \alpha_i)$: predicts noisy observations by marginalizing over the output distribution.

Bayesian updates are computed by incorporating noisy observations $\mathbf{y}_i$ into parameters $\boldsymbol{\theta}_i$:

$$p_{\mathtt{U}}(\boldsymbol{\theta}_i \mid \boldsymbol{\theta}_{i-1}; \alpha_i) = \mathbb{E}_{p_{\mathtt{O}}(\mathbf{m} \mid \theta_{i-1}; t_i)} \left[ \mathbb{E}_{p_{\mathtt{S}}(\mathbf{y}_i \mid \mathbf{m}; \alpha_i)}[\delta(\boldsymbol{\theta}_i - h(\boldsymbol{\theta}_{i-1}, \mathbf{y}_i, \alpha_i))] \right] \tag{2}$$

BFN ultimately optimize the KL divergence between sender and receiver distributions, progressively refining parameters toward accurate data approximations. Therefore, this framework uniformly handle discrete and continuous data through a modality-agnostic framework operating entirely on

distribution parameters, thus avoiding discontinuities found in discrete diffusion models. After training, sample generation involves iteratively refining parameters $\boldsymbol{\theta}_i$: 1) Draw intermediate sample $\mathbf{m}'_i \sim p_0(\cdot \mid \boldsymbol{\theta}_{i-1}, t_i)$. 2) Obtain noisy observation $\mathbf{y}_i \sim p_\mathtt{S}(\cdot \mid \mathbf{m}'_i, \alpha_i)$. 3) Update parameters via Bayesian inference: $\boldsymbol{\theta}_i = h(\boldsymbol{\theta}_{i-1}, \mathbf{y}_i, \alpha_i)$.

This iterative Bayesian refinement yields progressively accurate parameter representations, enabling coherent and robust data generation. For a detailed explanation of this part, we recommend referring to the original paper [18] that proposed this model, as well as the Appendix A provided in this paper.

# 4 Unified Framework of Bayesian Flow Network and Score Based Diffusion

## 4.1 Overview

Building on the previously outlined theoretical foundations of score-based models [22], particularly the concept of guidance, we now discuss how this framework can be integrated into the inference process of BFN across different data modalities.

## 4.2 BFN for Continuous Variables from Score Gradient

When modeling continuous variables (e.g., atomic coordinates) using a Gaussian distribution $\mathcal{N}\left(\mathbf{X} \mid \boldsymbol{\theta}^\mathbf{x}, \rho^{-1}\mathbf{I}\right)$, the parameters $\boldsymbol{\theta}$ that BFN seeks to optimize can be defined as the mean and covariance matrices of the distribution: $\boldsymbol{\theta} = \{\boldsymbol{\theta}^\mathbf{x}, \rho\}$. The bayesian update function of update distribution (Equation (2)) whose fundamental objective is to find the optimal $\theta$ that best describes the data distribution, is defined as below:

$$h\left(\left\{\boldsymbol{\theta}^\mathbf{x}_{i-1}, \rho_{i-1}\right\}, \mathbf{y}, \alpha_i\right) = \left\{\boldsymbol{\theta}^\mathbf{x}_i, \rho_i\right\}, \quad \text{where} \quad \rho_i = \rho_{i-1} + \alpha_i, \ \boldsymbol{\theta}^\mathbf{x}_i = \frac{\boldsymbol{\theta}^\mathbf{x}_{i-1}\rho_{i-1} + \mathbf{y}\alpha_i}{\rho_i} \quad (3)$$

**Proposition 4.1.** *According to Graves [18], when the sender distribution for sampling $\boldsymbol{\theta}^\mathbf{x}_i$ is defined as $p_\mathtt{S}(\mathbf{y} \mid \mathbf{X}; \alpha\mathbf{I}) = \mathcal{N}\left(\mathbf{X}, \alpha^{-1}\mathbf{I}\right)$, the updated $\boldsymbol{\theta}^\mathbf{x}_i$ resulting from the update function can be expressed in the following normal distribution form:*

$$\boldsymbol{\theta}^\mathbf{x}_i \sim \mathcal{N}\left(\frac{\boldsymbol{\theta}^\mathbf{x}_{i-1}\rho_{i-1} + \alpha\mathbf{x}}{\rho_i}, \frac{\alpha_i}{\rho_i^2}\mathbf{I}\right) \quad (4)$$

*By applying the reparameterization trick to the normal distribution above, it can be reformulated as the following linear Gaussian model: $\boldsymbol{\theta}^\mathbf{x}_i = \frac{\alpha\mathbf{X} + \boldsymbol{\theta}^\mathbf{x}_{i-1} \cdot \rho_{i-1}}{\rho_i} + \frac{\sqrt{\alpha_i}}{\rho_i} \cdot \mathbf{z}$ for $\mathbf{z} \sim \mathcal{N}(\mathbf{0}, \mathbf{I})$. The transformation from above linear gaussian model to the following equivalent formulation*

$$\mathbf{x} \sim \mathcal{N}\left(\frac{1}{\alpha}\left(\rho_i\boldsymbol{\theta}^\mathbf{x}_i - \boldsymbol{\theta}^\mathbf{x}_{i-1} \cdot \rho_{i-1}\right), \frac{1}{\alpha}\mathbf{I}\right) \quad (5)$$

*is obtained by rearranging the terms to explicitly express $\mathbf{x}$ in terms of the updated parameters.*

The resulting distribution allows us to approximate the mean of $\mathbf{x}$ as follows: $\frac{1}{\alpha}\left(\rho_i\boldsymbol{\theta}^\mathbf{x}_i - \boldsymbol{\theta}^\mathbf{x}_{i-1} \cdot \rho_{i-1}\right)$. Furthermore, this can be alternatively interpreted using Tweedie's formula from a different perspective.

**Definition 4.1**(Tweedie's Formula [13, 28]). *Let $z$ be a Gaussian random variable following the distribution $\mathbf{x} \sim \mathcal{N}(\mathbf{x}; \mu_\mathbf{x}, \Sigma_\mathbf{x})$. Then, Tweedie's formula states that:*

$$\mathbb{E}[\mu_\mathbf{x}|\mathbf{x}] = \mathbf{x} + \Sigma_\mathbf{x}\nabla_\mathbf{x}\log p(\mathbf{x}) \quad (6)$$

*where $p(\mathbf{X})$ denotes the marginal distribution of $\mathbf{x}$: $f(\mathbf{x}) = \int_{-\infty}^{\infty} \phi_\sigma(\mathbf{x} - \mu)g(\mu)d\mu$, here $\phi_\sigma(\mu) = \left(2\pi\sigma^2\right)^{-1/2} \exp\left\{-z^2/\sigma^2\right\}$*

Based on the principles outlined in Definition 5.1, the corresponding relationship for the normal distribution in Equation (5) can be formally established as follows: $\frac{1}{\alpha}\left(\rho_i\boldsymbol{\theta}^\mathbf{x}_i - \boldsymbol{\theta}^\mathbf{x}_{i-1} \cdot \rho_{i-1}\right) = \mathbf{x} + \frac{1}{\alpha} \cdot \nabla_\mathbf{x}\log p(\mathbf{x})$. The derived formula enables us to reinterpret the Bayesian update function in Equation (4) from the perspective of the score function.

$$\begin{aligned}
h\left(\boldsymbol{\theta}^\mathbf{x}_{i-1}, \mathbf{y}_i, \alpha_i\right) &= \frac{\boldsymbol{\theta}^\mathbf{x}_{i-1}\rho_{i-1} + \mathbf{y}\alpha_i}{\rho_i} = \frac{\alpha\mathbf{x} + \boldsymbol{\theta}^\mathbf{x}_{i-1} \cdot \rho_{i-1}}{\rho_i} + \frac{\sqrt{\alpha_i}}{\rho_i} \cdot \boldsymbol{\epsilon} \\
&= \frac{\alpha}{\rho_i} \cdot \mathbf{x} + \frac{\rho_{i-1}}{\rho_i} \cdot \boldsymbol{\theta}^\mathbf{x}_{i-1} + \frac{1}{\rho_i}\nabla\mathbf{x}\log p(\mathbf{x}) \quad (7)
\end{aligned}$$

Using the reparameterized Bayesian update function derived above, we can rewrite the Bayesian update distribution $p_U$ from Equation (2) as follow:

$$p_U \left( \boldsymbol{\theta}_i^{\mathbf{x}} \mid \boldsymbol{\theta}_{i-1}^{\mathbf{x}}; \alpha_i \right) = \mathop{\mathbb{E}}_{p_0(\mathbf{m}|\theta_{i-1};t_i)} \left[ \mathop{\mathbb{E}}_{p_S(\mathbf{y}_i|\mathbf{m};\alpha_i)} \left[ \zeta \left( \theta_i \mid \boldsymbol{\theta}_{i-1}, \mathbf{y}_i, \alpha_i \right) \right] \right]$$

$$, \text{where} \quad \zeta \left( \boldsymbol{\theta}_i^{\mathbf{x}} \mid \boldsymbol{\theta}_{i-1}^{\mathbf{x}}, \mathbf{y}_i, \alpha_i \right) : \boldsymbol{\theta}_i^{\mathbf{x}} \leftarrow \frac{\alpha}{\rho_i} \cdot \mathbf{x} + \frac{\rho_{i-1}}{\rho_i} \cdot \boldsymbol{\theta}_{i-1}^{\mathbf{x}} + \frac{1}{\rho_i} \nabla_{\mathbf{x}} \log p(\mathbf{x}) \tag{8}$$

where we define a compact notation for convenience: $\delta \left( \boldsymbol{\theta}_i^{\mathbf{x}} - h \left( \boldsymbol{\theta}_{i-1}^{\mathbf{x}}, \mathbf{y}_i, \alpha_i \right) \right) = \zeta \left( \boldsymbol{\theta}_i^{\mathbf{x}} \mid \boldsymbol{\theta}_{i-1}^{\mathbf{x}}, \mathbf{y}_i, \alpha_i \right)$ Interestingly, the above derivation demonstrates that we can update the parameter $\boldsymbol{\theta}$ by means of the gradient taken with respect to $\mathbf{x}$.

### 4.3 BFN for Discrete Variables from Score Gradient

Similar to the modeling of continuous variables, and recalling that the fundamental objective of BFN is to identify parameters that effectively encapsulate the data, this section explores whether the Bayesian update process for discrete types can be reinterpreted in gradient form. According to Graves, the Bayesian update function for discrete variables is defined as follows: $h \left( \boldsymbol{\theta}_{i-1}, \mathbf{y}, \alpha \right) = \boldsymbol{\theta}_i = \frac{e^{\mathbf{y}} \boldsymbol{\theta}_{i-1}}{\sum_{k=1}^{K} e^{\mathbf{y}_k (\boldsymbol{\theta}_{i-1})_k}}$. Furthermore, the sender distribution, which governs the sampling of the perturbed state $\mathbf{y}$ of the discrete variable, is given by:

$$p_S \left( \mathbf{y} \mid \mathbf{x}; \alpha \right) = \mathcal{N} \left( \mathbf{y} \mid \alpha \left( K \mathbf{e}_{\mathbf{x}} - \mathbf{1} \right), \alpha K \cdot \mathbf{I} \right) \tag{9}$$

where $\mathbf{1}$ is a vector of ones, $\mathbf{I}$ is the identity matrix, and $\mathbf{e}_i \in \mathbb{R}^K$ is a vector defined as the projection from the class index $i$ to a length K one-hot vector (details provided by [18]). It is important to note that $\mathbf{e}_{\mathbf{x}}$ is a one-hot vector only when it serves as the initial input.

**Proposition 4.2.** *Similar to Proposition 4.1, we can reformulate the relationship between $\mathbf{e}_{\mathbf{x}}$ and $\mathbf{y}$ using the reparameterization trick, as expressed follow: $\mathbf{e}_{\mathbf{x}} \sim \mathcal{N} \left( \frac{1}{\alpha K} \cdot \mathbf{y} + \frac{1}{K}, \frac{1}{\alpha K} \mathbf{I} \right)$. The essence of the Bayesian update function for discrete variables lies in applying a softmax operation to the previous timestep parameter $\boldsymbol{\theta}$ and the perturbed variable $\mathbf{y}$. Notably, this allows us to express $\mathbf{y}$ in a gradient form, enabling a formal characterization of the relationship between the update function and the score gradient.*

Similarly, applying Tweedie's formula to the normal distribution from which $\mathbf{e}_{\mathbf{x}}$ is sampled, as derived above, yields the following relationship for the computed mean: $\frac{1}{\alpha K} \cdot \mathbf{y} + \frac{1}{K} = \mathbf{e}_{\mathbf{x}} + \frac{1}{\alpha K} \nabla_{\mathbf{e}_{\mathbf{x}}} \log p(\mathbf{e}_{\mathbf{x}})$. Utilizing this result, the update function for discrete variables can be extended into the following gradient form:

$$h \left( \boldsymbol{\theta}_{i-1}, \mathbf{y}, \alpha \right) = \frac{e^{\mathbf{y}} \boldsymbol{\theta}_{i-1}}{\sum_{k=1}^{K} e^{\mathbf{y}_k} (\boldsymbol{\theta}_{i-1})_k} = \text{Softmax}(e^{\mathbf{y}} \cdot \boldsymbol{\theta}_{i-1}) \tag{10}$$

$$, \text{where} \quad \mathbf{y} = \alpha \left( K \cdot \mathbf{e}_{\mathbf{x}} - \mathbf{1} \right) + \sqrt{\alpha K} \cdot \epsilon = \alpha \left( K \cdot \mathbf{e}_{\mathbf{x}} - \mathbf{1} \right) + \nabla_{\mathbf{e}_{\mathbf{x}}} \log p(\mathbf{e}_{\mathbf{x}})$$

Similarly, as with the case of continuous variables, the Bayesian update distribution for discrete variables can be expressed using Eq. 10 as follows (for notational convenience, $\delta \left( \boldsymbol{\theta}_i - h \left( \boldsymbol{\theta}_{i-1}, \mathbf{y}_i, \alpha_i \right) \right)$ is denoted as $\zeta_{\mathbf{x}} \left( \theta_i \mid \boldsymbol{\theta}_{i-1}, \mathbf{y}_i, \alpha_i \right)$ in the discrete variable case):

$$p_U \left( \boldsymbol{\theta}_i^{\mathbf{v}} \mid \boldsymbol{\theta}_{i-1}^{\mathbf{v}}; \alpha_i \right) = \mathop{\mathbb{E}}_{p_0(\mathbf{m}|\boldsymbol{\theta}_{i-1}^{\mathbf{v}};t_i)} \left[ \mathop{\mathbb{E}}_{p_S(\mathbf{y}_i|\mathbf{m};\alpha_i)} \left[ \zeta_{\mathbf{v}} \left( \boldsymbol{\theta}_i^{\mathbf{v}} \mid \boldsymbol{\theta}_{i-1}^{\mathbf{v}}, \mathbf{y}_i, \alpha_i \right) \right] \right] \tag{11}$$

$$, \text{where} \quad \zeta_{\mathbf{v}} \left( \boldsymbol{\theta}_i^{\mathbf{v}} \mid \boldsymbol{\theta}_{i-1}^{\mathbf{v}}, \mathbf{y}_i, \alpha_i \right) : \boldsymbol{\theta}_i^{\mathbf{v}} \leftarrow \text{Softmax} \left( e^{\alpha(K \cdot \mathbf{e}_{\mathbf{x}} - \mathbf{1}) + \nabla_{\mathbf{e}_{\mathbf{x}}} \log p(\mathbf{e}_{\mathbf{x}})} \cdot \boldsymbol{\theta}_{i-1} \right)$$

Through this, we have demonstrated that the Bayesian update process for both continuous and categorical variables can be effectively reformulated into a gradient-based representation. In addition, we provide a detailed theoretical discussion and analysis of the relationship between our extended BFN and diffusion models in the Appendix B. In the following section, we will incorporate this concept specifically into the context of molecular generation for SBDD, introducing a strategy to generate molecules with desired target properties.

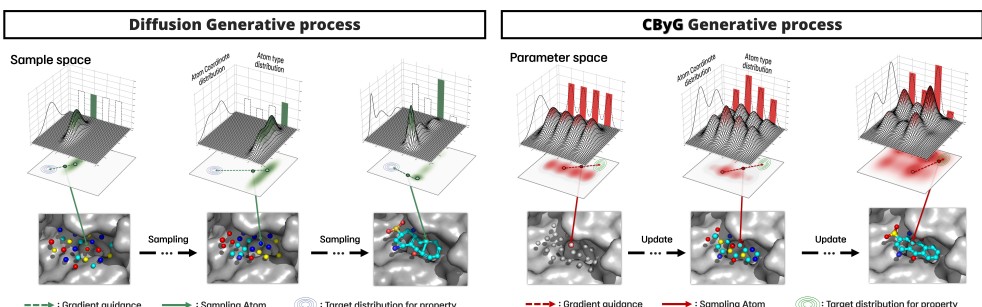

Figure 3: Schematic illustration comparing the guidance propagation mechanisms of diffusion-based models and CBYG in target protein-aware molecular generation. The figure shows how guidance propagation occurs in the molecular distribution $p(\mathbf{x})$ across the 3D molecular space for each model type. Further interpretation of this schematic illustration is provided in the Appendix E.

# 5 Guided Bayesian Flow Networks for 3D Molecule Generation in Structure-Based Drug Design

## 5.1 Notation

The primary objective of this study is to advance SBDD by developing generative models capable of producing molecules specifically tailored to bind to a given protein target. A target protein is represented as $\mathbf{P} = \{(\mathbf{x}_\mathrm{P}, \mathbf{v}_\mathrm{P})\}$, and our goal is to generate ligand molecules denoted by $\mathbf{m} = \{(\mathbf{x}_\mathrm{M}, \mathbf{v}_\mathrm{M})\}$.

Each atom in molecules and proteins is described by its 3D position $\mathbf{x} \in \mathbb{R}^3$ and its chemical type $\mathbf{v} \in \mathbb{R}^K$, where $K$ represents the total number of possible atom types. Molecules and proteins can thus be effectively represented as matrices: $\mathbf{m} = [\mathbf{x}_\mathrm{M}, \mathbf{v}_\mathrm{M}]$ and $\mathbf{p} = [\mathbf{x}_\mathrm{P}, \mathbf{v}_\mathrm{P}]$, with dimensions $\mathbf{x}_\mathrm{M} \in \mathbb{R}^{N_\mathrm{M} \times 3}$, $\mathbf{v}_\mathrm{M} \in \mathbb{R}^{N_\mathrm{M} \times K}$, $\mathbf{x}_\mathrm{P} \in \mathbb{R}^{N_\mathrm{P} \times 3}$, and $\mathbf{v}_\mathrm{P} \in \mathbb{R}^{N_\mathrm{P} \times K}$. For brevity, we denote molecules as $\mathbf{M} = [\mathbf{x}, \mathbf{v}]$. Here, $N_\mathrm{M}$ and $N_\mathrm{P}$ denote the number of atoms in the molecule and the protein, respectively.

In the following section, we introduce our proposed model, CBYG. A general schematic comparison between CBYG and diffusion models is provided in Figure 3.

## 5.2 Bayesian Update Distribution in SBDD

To enable property-controlled generation without retraining, we leverage the gradient-based Bayesian update formulation (Section 4) explicitly conditioned on target properties $\mathbf{l}$: $\boldsymbol{\theta}_i \leftarrow \mathbf{h}(\boldsymbol{\theta}_{i-1}, \mathbf{y}_i, \alpha_i, \mathbf{l})$. Instead of embedding properties directly into the output distribution (which would require retraining), we integrate a pretrained BFN with a BNN (Bayesian Neural Network) based external property predictor. This conditional sampling approach closely aligns with methods used in conditional diffusion models and guided generative frameworks [45, 4]. Formally, extending Equation (1), the conditional molecular distribution is defined as follows:

$$
\begin{aligned}
p_\phi(\mathbf{m}|\mathbf{p}, \mathbf{l}) &= \int p_\phi(\mathbf{m}|\boldsymbol{\theta}_n, \mathbf{p}, \mathbf{l}) p(\boldsymbol{\theta}_0) \prod_{i=1}^{n} p_\mathtt{U}(\boldsymbol{\theta}_i|\boldsymbol{\theta}_{i-1}, \mathbf{p}, \mathbf{l}; \alpha_i) d\boldsymbol{\theta}_{1:n} \\
&= \left[ \int p_\phi(\mathbf{x}|\boldsymbol{\theta}_n^\mathbf{x}, \mathbf{p}, \mathbf{l}) p(\boldsymbol{\theta}_0^\mathbf{x}) \prod_{i=1}^{n} p_\mathtt{U}(\boldsymbol{\theta}_i^\mathbf{x}|\boldsymbol{\theta}_{i-1}^\mathbf{x}, \mathbf{p}, \mathbf{l}; \alpha_i) d\boldsymbol{\theta}_{1:n}^\mathbf{x} \right] \\
&\quad \cdot \left[ \int p_\phi(\mathbf{v}|\boldsymbol{\theta}_n^\mathbf{v}, \mathbf{p}, \mathbf{l}) p(\boldsymbol{\theta}_0^\mathbf{v}) \prod_{i=1}^{n} p_\mathtt{U}(\boldsymbol{\theta}_i^\mathbf{v}|\boldsymbol{\theta}_{i-1}^\mathbf{v}, \mathbf{p}, \mathbf{l}; \alpha_i) d\boldsymbol{\theta}_{1:n}^\mathbf{v} \right].
\end{aligned}
\tag{12}
$$

The Bayesian update distribution $p_\mathtt{U}$, incorporating explicit conditioning on protein $\mathbf{p}$ and property $\mathbf{l}$, is given by:

$$
\begin{aligned}
p_\mathtt{U}(\boldsymbol{\theta}_i^\mathbf{x}|\boldsymbol{\theta}_{i-1}^\mathbf{x}, \mathbf{p}, \mathbf{l}; \alpha_i) &= \mathbb{E}_{p_0(\mathbf{m}|\boldsymbol{\theta}_{i-1}, \mathbf{p}; t_i)} \left[ \mathbb{E}_{p_\mathtt{S}(\mathbf{y}_i|\mathbf{x}; \alpha_i)} \left[ \zeta_\mathbf{x}(\boldsymbol{\theta}_i^\mathbf{x}|\boldsymbol{\theta}_{i-1}^\mathbf{x}, \mathbf{y}_i, \mathbf{x}, \alpha_i, \mathbf{p}, \mathbf{l}) \right] \right] \\
p_\mathtt{U}(\boldsymbol{\theta}_i^\mathbf{v}|\boldsymbol{\theta}_{i-1}^\mathbf{v}, \mathbf{p}, \mathbf{l}; \alpha_i) &= \mathbb{E}_{p_0(\mathbf{m}|\boldsymbol{\theta}_{i-1}^\mathbf{v}, \mathbf{p}; t_i)} \left[ \mathbb{E}_{p_\mathtt{S}(\mathbf{y}_i|\mathbf{v}; \alpha_i)} \left[ \zeta_\mathbf{v}(\boldsymbol{\theta}_i^\mathbf{v}|\boldsymbol{\theta}_{i-1}^\mathbf{v}, \mathbf{y}_i, \mathbf{v}, \alpha_i, \mathbf{p}, \mathbf{l}) \right] \right].
\end{aligned}
\tag{13}
$$

The sender distribution for atom positions is defined as: $p_\mathtt{S}(\mathbf{y} \mid \mathbf{x}; \alpha\mathbf{I}) = \mathcal{N}(\mathbf{x}, \alpha^{-1}\mathbf{I})$, where $\alpha$ controls the noise intensity around clean coordinates $\mathbf{x}$. The corresponding output distribution

is modeled via a neural network $\Psi$: $p_0(\mathbf{m} \mid \boldsymbol{\theta}_{i-1}, \mathbf{p}; t) = \Psi(\boldsymbol{\theta}_{i-1}, \mathbf{p}, t_i)$, which predicts atom coordinates and types at each time step $t_i$, conditioned on protein $\mathbf{p}$. For atom types, the sender distribution embeds discrete types, $\mathbf{v}$ into continuous space: $p_S(\mathbf{y} \mid \mathbf{v}; \alpha) = \mathcal{N}(\mathbf{y} \mid \alpha(K\mathbf{e_v} - \mathbf{1}), \alpha K \cdot \mathbf{I})$, with $\mathbf{e_v}$ as a one-hot vector, $K$ the number of categories, and $\alpha$ scaling the noise. Employing score-based guidance [45, 11], we embed property conditions into the gradient updates of $\boldsymbol{\theta}$ identified in Section 4, explicitly guiding molecule generation towards desired properties. Thus, our guided Bayesian updates for continuous and discrete variables are respectively defined as:

$$\zeta_{\mathbf{x}}\left(\boldsymbol{\theta}_i^{\mathbf{x}} \mid \boldsymbol{\theta}_{i-1}^{\mathbf{x}}, \mathbf{y}_i, \mathbf{x}, \alpha_i, \mathbf{p}, \mathbf{1}\right) : \boldsymbol{\theta}_i^{\mathbf{x}} \leftarrow \frac{\alpha_i}{\rho_i} \cdot \mathbf{x} + \frac{\rho_{i-1}}{\rho_i} \cdot \boldsymbol{\theta}_{i-1}^{\mathbf{x}} + \frac{1}{\rho_i}\nabla\mathbf{x}\log p(\mathbf{x} \mid \mathbf{1})$$

$$= \underbrace{\frac{\alpha_i}{\rho_i} \cdot \mathbf{y} + \frac{\rho_{i-1}}{\rho_i} \cdot \boldsymbol{\theta}_{i-1}^{\mathbf{x}}}_{\text{Unconditional Generation}} + \underbrace{\frac{1}{\rho_i}\nabla_{\mathbf{x}}\log p(\mathbf{1} \mid \mathbf{x})}_{\text{Controllable Guidance}} \quad (14)$$

$$\zeta_{\mathbf{v}}\left(\boldsymbol{\theta}_i^{\mathbf{v}} \mid \boldsymbol{\theta}_{i-1}^{\mathbf{v}}, \mathbf{y}_i, \mathbf{x}, \alpha_i, \mathbf{p}, \mathbf{1}\right) : \boldsymbol{\theta}_i^{\mathbf{v}} \leftarrow \text{Softmax}\left(e^{\alpha_i(K \cdot \mathbf{e_x} - \mathbf{1}) + \nabla_{\mathbf{e_x}}\log p(\mathbf{e_x}|\mathbf{1})} \cdot \boldsymbol{\theta}_{i-1}^{\mathbf{v}}\right)$$

$$= \text{Softmax}\left(\underbrace{e^{\mathbf{y}} \cdot \boldsymbol{\theta}_{i-1}^{\mathbf{v}}}_{\substack{\text{Unconditional} \\ \text{Generation}}} \cdot \underbrace{e^{\nabla_{\mathbf{e_x}}\log p(\mathbf{1}|\mathbf{e_x})}}_{\substack{\text{Controllable} \\ \text{Guidance}}}\right) \quad (15)$$

The $n$-step sampling process of CBYG begins with an initial parameter $\boldsymbol{\theta}_0$, accuracy parameters $\alpha_i$, and time steps $t_i$. This process sequentially generates $\boldsymbol{\theta}_1, \ldots, \theta_n$ to produce the final sample. First, using $\boldsymbol{\theta}_{i-1}$ and $t_{i-1}$, a sample $\mathbf{x}$ is drawn from the output distribution $p_0(\cdot \mid \boldsymbol{\theta}_{i-1}, \mathbf{p}; t_i)$. Next, based on $\mathbf{x}$ and $\alpha_i$, a noisy sample $y$ is drawn from the sender distribution $p_S(\cdot \mid \mathbf{x}_{i-1}, \mathbf{p}; \alpha_i)$. The parameter is then updated via $p_U(\cdot \mid \boldsymbol{\theta}_{i-1}, \mathbf{y}, \mathbf{p}, \mathbf{1}; \alpha_i)$. Finally, the sample is generated from $p_0(\cdot \mid \boldsymbol{\theta}_n, \mathbf{p}; t_n)$ using the refined parameter $\theta_n$. Detailed descriptions of the Bayesian neural network-based property predictor for quantifying predictive uncertainty (guidance reliability), the sampling procedure of CBYG that incorporates this uncertainty, and the SE(3)-equivariance of the proposed guidance method are provided in Appendix D.

## 6 Experiment

To thoroughly evaluate the effectiveness and practicality of our proposed framework, we define 4 key research questions essential to the domain of structure-based drug design: 1) Are conventional metrics sufficient for reliably evaluating the predicted binding affinity of generated molecules? 2) Is the widely-used Synthetic Accessibility (SA) score an absolute indicator for practical synthetic feasibility? 3) Does the guidance mechanism incorporated into our Bayesian Flow Network (BFN) framework significantly enhance controllable generation performance? 4) Can our generative framework effectively handle molecular selectivity, a key property required for generating novel drug candidates? In the subsequent sections, we systematically address each of these research questions through rigorous empirical analyses and extensive evaluations.

### 6.1 Experimental Setup

We use the CrossDocked dataset [15] for training and testing, which originally contains 22.5 million protein-ligand pairs, and after the RMSD-based filtering and 30% sequence identity split by Luo et al. [31], results in 100,000 training pairs and 100 test proteins. For each test protein, we sample 100 molecules for evaluation. Baseline models used for comparison are described in the Appendix F.

### 6.2 Comprehensive Evaluation Including Binding Affinity of Generated Molecules

Previous evaluations of generative models for target-specific molecular generation have typically relied on binding affinity metrics derived primarily from AutoDock Vina [12], introducing inherent biases due to dependence on a single algorithm. To address this limitation, we incorporate additional docking tools, namely SMINA [30], which captures broader physicochemical interactions, and GNINA [32], employing a deep learning-based scoring function, to enhance the reliability and comprehensiveness of the evaluation. Furthermore, shifting away from the narrow focus on binding affinity alone, we introduce the PoseBusters [5] benchmark, which assesses molecular validity and

Table 1: Summary of binding affinity and molecular properties of reference molecules and molecules generated by CBYG and baselines. (↑) / (↓) denotes whether a larger / smaller number is preferred. Top 2 results are bolded and underlined, respectively.

| Methods | | SMINA (↓) | | GNINA (↓) | | Vina (↓) | | High Affinity(↑) | | SA(↑) | | PB-Valid(↑) | | Diversity(↑) | |
|---|---|---|---|---|---|---|---|---|---|---|---|---|---|---|---|---|
| | | Score. | Dock. | Score. | Dock. | Score. | Dock. | Avg. | Med. | Avg. | Med. | Avg. | Med. | Avg. | Med. |
| | Reference | -6.37 | -7.92 | -7.06 | -7.61 | -6.36 | -7.45 | - | - | 0.73 | 0.74 | 95.0% | 95.0% | - | - |
| Gen. | AR[31] | -5.04 | -7.18 | -5.96 | -6.31 | -5.75 | -6.75 | 37.9% | 31.0% | 0.63 | 0.63 | 88.3% | 90.0% | 0.70 | 0.70 |
| | Pocket2Mol [34] | -4.38 | -7.85 | -5.50 | -6.30 | -5.14 | -7.15 | 48.4% | 51.0% | 0.74 | 0.75 | 72.5% | 86.5% | 0.69 | 0.71 |
| | TargetDiff [20] | -3.66 | -8.51 | -5.50 | -6.44 | -5.47 | -7.80 | 58.1% | 59.1% | 0.58 | 0.58 | 67.0% | 70.0% | 0.72 | 0.71 |
| | DecompDiff [21] | -4.48 | -8.33 | -5.89 | -6.42 | -5.67 | -8.39 | 64.4% | 71.0% | 0.61 | 0.60 | 46.5% | 45.9% | 0.68 | 0.68 |
| | MolCRAFT [37] | _-6.03_ | _-8.56_ | -6.02 | _-7.26_ | -6.61 | -7.59 | 63.4% | - 70.0% | 0.68 | 0.67 | 74.0% | 82.5% | _0.70_ | **0.73** |
| Gen. + Opt. | RGA [16] | 22.61 | -7.25 | 19.99 | -5.28 | 20.58 | -8.01 | 64.4% | 89.3% | 0.71 | 0.73 | _91.3%_ | _94.1%_ | 0.41 | 0.41 |
| | DecompOpt [51] | -3.54 | -8.10 | -5.26 | -6.30 | -5.87 | _-8.98_ | 73.5% | _93.3%_ | 0.65 | 0.65 | 68.7% | 77.7% | 0.60 | 0.61 |
| | TacoGFN [42] | 39.80 | -7.46 | 32.90 | -6.50 | 32.63 | -7.74 | 58.4% | 63.6% | _0.79_ | _0.80_ | 89.3% | 90.1% | 0.56 | 0.56 |
| | ALIDiff [19] | -5.68 | -8.47 | _-6.85_ | -5.61 | _-7.07_ | -8.90 | 73.4% | 81.4% | 0.57 | 0.56 | 44.3% | 40.0% | **0.73** | _0.71_ |
| | TargetOpt | -5.71 | -8.39 | -6.12 | -7.01 | -6.96 | -8.87 | _76.6%_ | 82.3% | 0.71 | 0.69 | 78.8% | 81.3% | 0.60 | 0.59 |
| | **CBYG** | **-7.74** | **-9.61** | **-7.63** | **-8.33** | **-8.60** | **-9.16** | **93.6%** | **100.%** | **0.84** | **0.87** | **94.9%** | **96.0%** | 0.61 | 0.62 |

stability across 17 metrics, including chemical consistency and ligand stability. Detailed descriptions of the evaluation metrics and experimental setup are provided in the Appendix G.

**Results.** As shown in Table.1, Our proposed model substantially outperforms baseline methods across 12 key metrics related to binding affinity and molecular properties, confirming effective gradient-based guidance during generation. While baseline models exhibit significant differences between pre- (Score. in Table.1) and post- (Dock. in Table.1) docking scores, our model maintains consistently high affinity scores even before docking, indicating enhanced capability to implicitly identify favorable binding poses and explicitly capture protein-ligand interactions. The robustness of our model's superior binding affinity is further supported by its consistent outperformance across all three docking tools, unlike baseline methods whose relative rankings fluctuate across these tools. Although our method does not achieve state-of-the-art performance in molecular diversity, this limitation naturally arises from conditional generation, which inherently targets narrower distributions to meet specific binding criteria. Futhermore, We assessed energetic stability of top-ranked molecules from CBYG and baseline models using the PoseCheck [24] benchmark. Figure. 5 shows consistently low energies, indicating stable binding poses. Additional results are provided in the appendix. Detailed interpretations and results, including comprehensive score comparisons before and after docking and ablation studies on guidance scale variations within the CBYG framework, are provided in the Appendix H.

## 6.3 Evaluation of Realistic Synthetic Feasibility for Generated Molecules

To achieve a more precise assessment of the realistic synthesizability of generated molecules beyond conventional Synthetic Accessibility (SA) scores (as discussed in Section 2), we introduced the AiZynthFinder benchmark, which utilizes Monte Carlo Tree Search (MCTS) to systematically identify viable retrosynthetic pathways. This benchmark quantitatively evaluates practical synthetic feasibility through six distinct metrics (Detailed descriptions and values provided in the Appendix G).

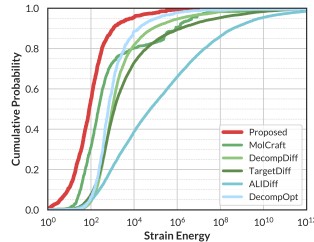

Figure 5: Cumulative distribution function of strain energy

**Results.** Our experimental results (Table. 3) demonstrate that despite exhibiting high binding affinity, our proposed model achieves near state-of-the-art performance in AiZynthFinder evaluations, indicating its ability to generate molecules that are both efficacious and realistically synthesizable. Interestingly, we observed no clear correlation between SA scores and actual retrosynthetic feasibility (as measured by the Solved metric), underscoring the need for broader and more comprehensive

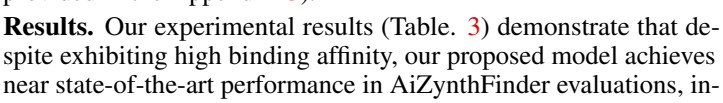

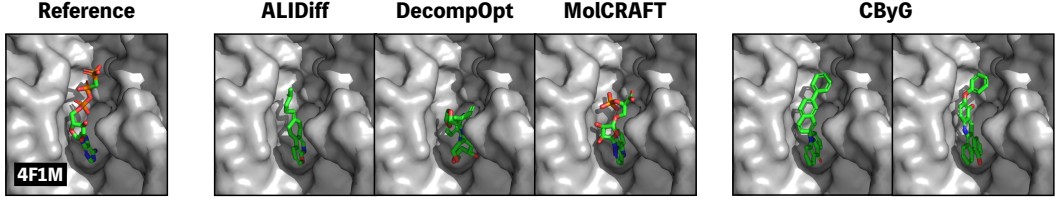

Figure 4: Visualizations of reference molecules and generated ligands for protein pockets (PDB ID: 4F1M) generated by CBYG, ALIDiff, DecompOpt, and MolCRAFT.

Table 2: Evaluation results of SA for molecules generated by baseline models and the proposed model using the AiZynthFinder benchmark.

| Metric | Solved (↑) | Routes (↑) | Solved Routes (↑) | Top Score (↑) |
|---|---|---|---|---|
| Reference | 0.410 | 294.1 | 11.43 | 0.826 |
| AR | 0.194 | 227.2 | 3.616 | 0.744 |
| Pocket2Mol | 0.373 | 150.6 | 7.743 | 0.797 |
| TargetDiff | 0.082 | 233.5 | 1.800 | 0.695 |
| DecompDiff | 0.038 | 202.7 | 1.165 | 0.703 |
| MolCRAFT | 0.196 | 242.2 | 4.554 | 0.745 |
| ALIDiff | 0.079 | 321.8 | 1.521 | 0.660 |
| RGA | **0.487** | 157.9 | 3.292 | 0.822 |
| DecompOpt | 0.080 | 201.9 | 1.205 | 0.713 |
| TacoGFN | 0.072 | 255.4 | 0.530 | 0.743 |
| **CBYG** | **0.487** | **334.4** | **18.90** | **0.853** |

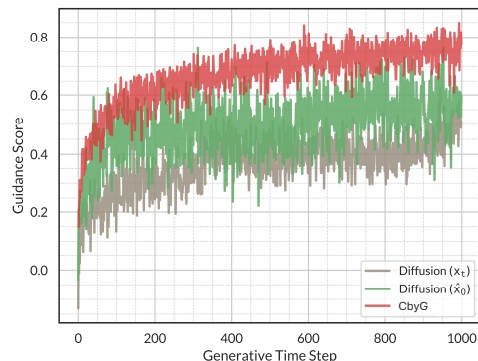

Figure 6: Guidance score dynamics of three model types throughout the generation process

evaluations of synthetic accessibility in future SBDD research. Notably, models with high SA scores exhibited practical retrosynthesis success rates of less than 50%, emphasizing the necessity of placing greater emphasis on realistic synthetic feasibility within the SBDD field. Further detailed discussion is provided in the Appendix H.

## 6.4 Assessment of Guidance Effectiveness Across Model Types

To validate the effectiveness of gradient-based guidance injection within CBYG framework, we compared it empirically against a diffusion-based model. Specifically, we analyzed the dynamics of guidance scores ($p(\mathtt{l} \mid \mathbf{m})$), representing molecular property predictions at each timestep during controllable generation (Diffusion model implementation details provided in Appendix E).

**Results.** As shown in Figure. 6, our CBYG guidance approach consistently achieves higher and more stable guidance scores compared to diffusion-based methods. While these method utilizing predicted final states ($\mathbf{x}_0$) can yield higher absolute scores, they exhibit increased variance due to unstable point-estimations from intermediate noisy states. Unlike diffusion models operating directly in sample space, CBYG conducts continuous gradient calculations within parameter space (including categorical variables) thereby enabling more stable and effective controllable generation. Further detailed analyses and discussions on this comparative experiment are provided in Appendix H.

## 6.5 Evaluation of Selectivity Control Capability

In this experiment, we evaluated whether our proposed controllable generation strategy effectively enhances molecular selectivity, a critical factor in practical drug discovery. We constructed a biologically relevant selectivity test set from prior pharmacological studies [27, 10] involving binding data of 38 kinase inhibitors across a panel of 317 kinases (Refer to Appendix I for detailed methods and dataset construction). Our results demonstrate that the proposed BFN-based guidance method significantly outperforms diffusion-based approaches in selectivity metrics (Succ. Rate and Score), with greater improvements upon guidance injection. These findings strongly support the practical applicability of BFN-based guidance in real-world drug development scenarios. Further detailed metrics and analyses are provided in the Appendix H.

Table 3: Evaluation of Selectivity Optimization Capability between Diffusion-based Guidance Methods and the CBYG Model. "CBYG w/o G" denotes the CByG model without guidance injection.

| Metric | Succ. Rate (↑) | Δ Score (↓) |
|---|---|---|
| TargetDiff | 58.2 % | -0.78 |
| TargetOpt | 68.6 % | |
| CBYG w/o G | 62.6 % | -1.39 |
| CBYG | 78.3 % | |

## 7 Conclusions

In this paper, we introduced CBYG, a generative framework designed to produce 3D molecules satisfying multiple critical properties, thereby enhancing practical applicability in structure-based drug design (SBDD). We discussed inherent limitations of conventional diffusion-based guidance methods and theoretically integrated a gradient-based guidance mechanism within the BFN framework to address these issues. Empirical evaluations, guided by the key objectives highlighted in Section 2, demonstrated that our proposed model significantly outperforms comparative baselines across various evaluation metrics. Nevertheless, the finding that approximately half of the molecules generated

(even by advanced models including CBᵧG) remain synthetically infeasible offers a critical insight and underscores a significant area for further investigation within this research domain.

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

# A  Detail description of Bayesian Flow Network

## A.1  Key Distribution

A Bayesian Flow Network (BFN) maintains four distributions at each step: the input distribution $p_{\mathtt{I}}(\mathbf{m} \mid \boldsymbol{\theta})$, the output distribution $p_{\mathtt{0}}(\mathbf{m} \mid \boldsymbol{\theta}, t)$, the sender distribution $p_{\mathtt{S}}(\mathbf{y} \mid \mathbf{m}; \alpha)$, and the receiver distribution $p_{\mathtt{R}}(\mathbf{y} \mid \boldsymbol{\theta}, t, \alpha)$. Semantically, the input distribution represents the model's current belief (prior or posterior) over the data; the output distribution is the network's predicted distribution (allowing context) given the input parameters; the sender distribution is a noisy perturbation of the true data; and the receiver distribution is the model's predicted distribution over such noisy messages (averaging over the output distribution). Mathematically, all four are factorized over the data dimensions for tractability.

**Input distribution** $p_{\mathtt{I}}(\mathbf{m} \mid \boldsymbol{\theta})$. This is a factorized distribution over the data $\mathbf{m} = (\mathbf{m}^{(1)}, \ldots, \mathbf{m}^{(D)})$ with parameters $\boldsymbol{\theta} = (\boldsymbol{\theta}^{(1)}, \ldots, \boldsymbol{\theta}^{(D)})$. For example, each $\boldsymbol{\theta}^{(d)}$ might parametrize a univariate Gaussian or categorical for $\mathbf{m}^{(d)}$. Initially $p_{\mathtt{I}}$ is a simple prior (e.g. $\mathcal{N}(0,1)$ or a uniform categorical). During sampling, $p_{\mathtt{I}}$ is updated via Bayes's rule when new observations arrive, so its parameters $\boldsymbol{\theta}$ become progressively more informative about $\mathbf{m}$.

**Output distribution** $p_{\mathtt{0}}(\mathbf{m} \mid \boldsymbol{\theta}, t)$. Given input parameters $\boldsymbol{\theta}$ and the current (discrete or continuous) process time $t$, a neural network computes an output vector that parameterizes $p_{\mathtt{0}}$. However, unlike $p_{\mathtt{I}}$ it integrates context across dimensions via the network: the parameters of $p_{\mathtt{0}}$ depend jointly on all of $\boldsymbol{\theta}$ and $t$, allowing modeling of correlations among data dimensions . In effect, $p_{\mathtt{0}}$ represents the empirical sample distribution accumulated up to step $t$, providing a comprehensive prediction that integrates the gathered evidence $\theta$ together with contextual information, and it will be used to predict the clean data distribution.

**Sender distribution** $p_{\mathtt{S}}(\mathbf{y} \mid \mathbf{m}; \alpha)$. This is the distribution over a noisy observation $\mathbf{y}$ given the true data $\mathbf{m}$, with accuracy (noise level) $\alpha_i$. It is also factorized across dimensions: $p_{\mathtt{S}}(\mathbf{y}_i \mid \mathbf{m}; \alpha_i) = \prod_{d=1}^{D} p_{\mathtt{S}}\left(\mathbf{y}_i^{(d)} \mid \mathbf{m}^{(d)}; \alpha_i\right)$. The parameter $\alpha_i$ controls informativeness: at $i = 0$ the sender's sample is pure noise (uninformative about $\mathbf{m}$), and as $i \to \infty$ the sample concentrates on $\mathbf{y} = \mathbf{m}$. In practice, $\alpha$ increases through the transmission steps, so that each sender sample carries more refined information about the true data. Intuitively, $p_{\mathtt{S}}$ defines the "message" drawn from data: Alice adds controlled Gaussian or categorical noise to $x$ to form the sender distribution and then draws a sample $y \sim p_{\mathtt{S}}(\cdot \mid x; \alpha)$ (For the Alice-and-Bob example, we recommend consulting the original BFN paper [18]).

**Receiver distribution** $p_{\mathtt{R}}(\mathbf{y} \mid \boldsymbol{\theta}, t, \alpha_i)$. This is the model's predictive distribution over possible noisy observations $\mathbf{y}$ given the output distribution. Formally, it is obtained by marginalizing out the unknown true data $\mathbf{m}$: $p_{\mathtt{R}}(\mathbf{y}_i \mid \boldsymbol{\theta}_{i-1}; t_i, \alpha_i) = \mathbb{E}_{\hat{\mathbf{m}} \sim p_{\mathtt{0}}(\hat{\mathbf{m}}|\theta_{i-1}; t_i)} \, p_{\mathtt{S}}(\mathbf{y} \mid \hat{\mathbf{m}}; \alpha_i)$. In other words, for every candidate $\mathbf{m}$, we consider the sender distribution $p_{\mathtt{S}}(\mathbf{y} \mid \mathbf{m}; \alpha)$ that would have been used if $\mathbf{m}$ were the truth, and weight these by the network's probability $p_{\mathtt{0}}(\mathbf{m} \mid \boldsymbol{\theta}, t)$. The receiver distribution thus captures both the "known unknown" due to the sender noise (entropy of $p_S$) and the "unknown unknown" from the output distribution's uncertainty.

**Bayesian update distribution** $p_{\mathtt{U}}(\theta_i \mid \theta_{i-1}; \alpha_i)$. This distribution specifies how the parameter vector of the input distribution evolves after assimilating a noisy observation. Concretely, let the closed-form update rule be $\boldsymbol{\theta}_i = h(\boldsymbol{\theta}_{i-1}, \mathbf{y}_i, \alpha_i)$ where $\mathbf{y}_i \sim p_{\mathtt{S}}(\mathbf{y}_i \mid \mathbf{m}; \alpha_i)$. Conditioning on the current parameters $\boldsymbol{\theta}_i$ and the accuracy level $\alpha_i$, the update distribution is defined as the push-forward of the sender distribution:

$$p_{\mathtt{U}}\left(\boldsymbol{\theta}_i \mid \boldsymbol{\theta}_{i-1}; \alpha_i\right) = \mathbb{E}_{p_{\mathtt{0}}(\mathbf{m}|\theta_{i-1}; t_i)} \left[ \mathbb{E}_{p_{\mathtt{S}}(\mathbf{y}_i|\mathbf{m}; \alpha_i)} \left[ \delta\left(\boldsymbol{\theta}_i - h\left(\boldsymbol{\theta}_{i-1}, \mathbf{y}_i, \alpha_i\right)\right) \right] \right] \tag{16}$$

Because $h(\cdot)$ is applied component-wise under the factorisation of $p_{\mathtt{S}}$, $p_{\mathtt{U}}$ factorises across parameter dimensions, preserving tractability. The role of $p_{\mathtt{U}}$ is to enact a one-step Bayesian posterior update on $\boldsymbol{\theta}$: at early iterations, when $\alpha_i$ is small, $p_{\mathtt{U}}$ is broad and admits substantial parameter movement, whereas at later iterations, larger $\alpha_i$ makes the update sharply concentrated, refining $\boldsymbol{\theta}$ toward the values most consistent with the clean data. Thus $p_{\mathtt{U}}$ provides the formal bridge between each noisy message and the progressively more informative input distribution maintained by the network.

## A.2 Modality-Agnostic Representation and Progressive Refinement

Because BFN operate on distribution parameters rather than raw data, the same framework applies uniformly to discrete, discretized, and continuous modalities. For example, discrete data are represented by categorical distribution parameters (probabilities), which lie on a probability simplex and thus provide continuous inputs to the network. In fact, the parameters of a categorical distribution are real-valued probabilities, so the inputs to the network are continuous even when the data is discrete. Likewise, continuous data use Gaussian or other continuous distributions. In all cases the model processes continuous-valued parameters, avoiding discontinuities common in discrete diffusion models. This modality-agnostic formulation means that the same Bayes-and-network machinery can generate text, images, or other data with only minimal adaptation.

## A.3 Generative Sampling Process

After training, sample generation proceeds by iteratively refining the distribution parameters via Bayesian updates. Starting from an initial parameter $\boldsymbol{\theta}_0$ (e.g., a prior at initial noise level $t_0$), the model performs $N$ iterations of the following procedure for $i = 1, \ldots, N$:

1. **Sample from output distribution:** An intermediate sample $\mathbf{m}_i'$ is drawn from the output distribution $\mathbf{m}_i' \sim p_0(\cdot \mid \boldsymbol{\theta}_{i-1}, t_i)$ given the current parameter $\boldsymbol{\theta}_{i-1}$ and scheduled noise level $t_i$.

2. **Sample from sender distribution:** A noisy observation $\mathbf{y}_i$ is then drawn from the sender distribution, conditional on $\mathbf{m}_i'$, via $\mathbf{y}_i' \sim p_{\mathrm{S}}(\cdot \mid \mathbf{m}_i', \alpha_i)$, where $\alpha_i$ is the accuracy (inverse noise variance) prescribed for step $i$.

3. **Bayesian parameter update:** The distribution parameter is updated by incorporating the observation $\mathbf{y}_i$ through the Bayesian update function: $\boldsymbol{\theta}_i = h(\boldsymbol{\theta}_{i-1}, \mathbf{y}_i, \alpha_i)$. Here $h(\boldsymbol{\theta}_{i-1}, \mathbf{y}_i, \alpha_i)$ computes the posterior parameter after observing $y_i$ with precision $\alpha_i$, given the prior $\boldsymbol{\theta}_{i-1}$.

Repeating the above steps yields a final parameter $\boldsymbol{\theta}_N$ after $N$ iterations. This $\boldsymbol{\theta}_N$ characterizes a highly concentrated distribution (approximately a Dirac delta distribution in the limit of large $N$), from which the final data sample can be obtained by drawing $\mathbf{m} \sim p_0(\cdot \mid \theta_N, t_N)$. Importantly, the receiver distribution $p_{\mathrm{R}}$ is not explicitly used during generation – its effect is implicitly achieved by the two-step sampling ($p_0$ followed by $p_{\mathrm{S}}$) at each iteration. This ensures that sampling relies solely on the forward update pattern $\mathbf{m}_i' \to \mathbf{y}_i \to \boldsymbol{\theta}_i$ described above, in line with the canonical BFN [18] formulation. A simplified schematic illustration of this is provided in Figure 7.

# B Score-Driven Sampling in DDPM vs. BFN: A Rigorous Comparison

## B.1 Continuous Variable Modeling

**Diffusion based sampling**

$$\mathbf{x}_{t-1} = \frac{1}{\sqrt{\alpha_t}}\mathbf{x}_t - \frac{1 - \alpha_t}{\sqrt{1 - \bar{\alpha}_t}\sqrt{\alpha_t}}\boldsymbol{\epsilon}_0 + \sigma_t \mathbf{z}$$

$$= \frac{\sqrt{\alpha_t}(1 - \bar{\alpha}_{t-1})}{1 - \bar{\alpha}_t}\mathbf{x}_t + \frac{\sqrt{\bar{\alpha}_{t-1}}(1 - \alpha_t)}{1 - \bar{\alpha}_t}\mathbf{x}_0 + \sigma_t \mathbf{z}$$

$$= \frac{1}{\sqrt{\alpha_t}}\mathbf{x}_t + \frac{1 - \alpha_t}{\sqrt{\alpha_t}}\nabla_{\mathbf{x}_t}\log p(\mathbf{x}_t) + \sigma_t \mathbf{z}$$

$$\text{guidance} \to \frac{1}{\sqrt{\alpha_t}}\mathbf{x}_t + \frac{1 - \alpha_t}{\sqrt{\alpha_t}}\nabla_{\mathbf{x}_t}\log p(\mathbf{x}_t \mid y) + \sigma_t \mathbf{z} \tag{17}$$

$$= \frac{1}{\sqrt{\alpha_t}}\mathbf{x}_t + \frac{1 - \alpha_t}{\sqrt{\alpha_t}}\nabla_{\mathbf{x}_t}\log p(\mathbf{x}_t) + \frac{1 - \alpha_t}{\sqrt{\alpha_t}}\nabla_{\mathbf{x}_t}\log p(y \mid \mathbf{x}_t) + \sigma_t \mathbf{z} \tag{18}$$

$$\simeq \frac{1}{\sqrt{\alpha_t}}\mathbf{x}_t + \frac{1 - \alpha_t}{\sqrt{\alpha_t}}\nabla_{\mathbf{x}_t}\log p(\mathbf{x}_t) + \frac{1 - \alpha_t}{\sqrt{\alpha_t}}\nabla_{\mathbf{x}_t}\log p(y \mid \mathbf{x}_0) + \sigma_t \mathbf{z} \tag{19}$$

**BFN based sampling**

$$\boldsymbol{\theta}_i = \frac{\rho_{i-1}}{\rho_i}\boldsymbol{\theta}_{i-1} + \frac{\alpha_i}{\rho_i}\mathbf{y} \tag{20}$$

$$= \frac{\rho_{i-1}}{\rho_i}\boldsymbol{\theta}_{i-1} + \frac{\alpha}{\rho_i}\mathbf{x}_0 + \frac{1}{\rho_i}\nabla_{\mathbf{x}_0}\log p(\mathbf{x}_0) \tag{21}$$

$$\text{guidance} \rightarrow \frac{\rho_{i-1}}{\rho_i}\boldsymbol{\theta}_{i-1} + \frac{\alpha}{\rho_i}\mathbf{x}_0 + \frac{1}{\rho_i}\nabla_{\mathbf{x}_0}\log p(\mathbf{x}_0 \mid y) \tag{22}$$

$$= \frac{\rho_{i-1}}{\rho_i}\boldsymbol{\theta}_{i-1} + \frac{\alpha}{\rho_i}\mathbf{x}_0 + \frac{1}{\rho_i}\nabla_{\mathbf{x}_0}\log p(\mathbf{x}_0) + \frac{1}{\rho_i}\nabla_{\mathbf{x}_0}\log p(y \mid \mathbf{x}_0) \tag{23}$$

$$= \frac{\rho_{i-1}}{\rho_i}\boldsymbol{\theta}_{i-1} + \frac{\alpha_i}{\rho_i}\mathbf{y} + \frac{1}{\rho_i}\nabla_{\mathbf{x}_0}\log p(y \mid \mathbf{x}_0) \tag{24}$$

## B.2 Categorical Variable Modeling

**Diffusion based sampling**

$$\mathbf{x}_{t-1} \sim \mathcal{C}\left(\mathbf{x}_{t-1} \mid \boldsymbol{\theta}_{\text{post}}(\mathbf{x}_t, \hat{\mathbf{x}}_0)\right) \tag{25}$$

$$\text{, where } \boldsymbol{\theta}_{\text{post}}(\mathbf{x}_t, \hat{\mathbf{x}}_0) = \frac{\left[\alpha_t \mathbf{x}_t + \frac{1-\alpha_t}{K}\right] \odot \left[\bar{\alpha}_{t-1}\hat{\mathbf{x}}_0 + \frac{1-\bar{\alpha}_{t-1}}{K}\right]}{\sum_{k=1}^{K}\left(\left[\alpha_t \mathbf{x}_t + \frac{1-\alpha_t}{K}\right] \odot \left[\bar{\alpha}_{t-1}\hat{\mathbf{x}}_0 + \frac{1-\bar{\alpha}_{t-1}}{K}\right]\right)_k} \tag{26}$$

Here, $\odot$ denotes element-wise multiplication, ensuring the probabilities are normalized over all categories. And, $\mathcal{C}$ denotes categorical distribution to model discrete atom types $\mathbf{v}$.

**BFN based Sampling**

$$\boldsymbol{\theta}_i = \text{Softmax}(e^{\mathbf{y}} \cdot \boldsymbol{\theta}_{i-1}) \tag{27}$$

$$= \text{Softmax}(e^{\alpha(K \cdot \mathbf{e_x} - \mathbf{1}) + \nabla_{\mathbf{e_x}}\log p(\mathbf{e_x})} \cdot \boldsymbol{\theta}_{i-1}) \tag{28}$$

$$\text{guidance} \rightarrow \text{Softmax}(e^{\alpha(K \cdot \mathbf{e_x} - \mathbf{1}) + \nabla_{\mathbf{e_x}}\log p(\mathbf{e_x}|\mathbf{1})} \cdot \boldsymbol{\theta}_{i-1}) \tag{29}$$

$$= \text{Softmax}\left(e^{\mathbf{y}} \cdot \boldsymbol{\theta}_{i-1}^{\mathbf{v}} \cdot e^{\nabla_{\mathbf{e_x}}\log p(\mathbf{1}|\mathbf{e_x})}\right) \tag{30}$$

## B.3 Theoretical Distinctions between Diffusion Models and Bayesian Flow Networks

Bayesian Flow Networks (BFN) differ fundamentally from Diffusion Models (DMs) in their mechanism of uncertainty injection and the domain of parameter updates. Unlike diffusion-based generative processes, which explicitly manipulate the data sample $x$ and inject Gaussian noise at each step to maintain stochasticity, BFN iteratively update distribution parameters $\theta$ through Bayesian inference, integrating noisy observations $y_i$ drawn from a sender distribution $p_S(y \mid x; \alpha_i)$.

In contrast to diffusion methods, BFN have no explicit forward noise injection process. Instead, the generation is achieved by a deterministic parameter update conditional upon the observed

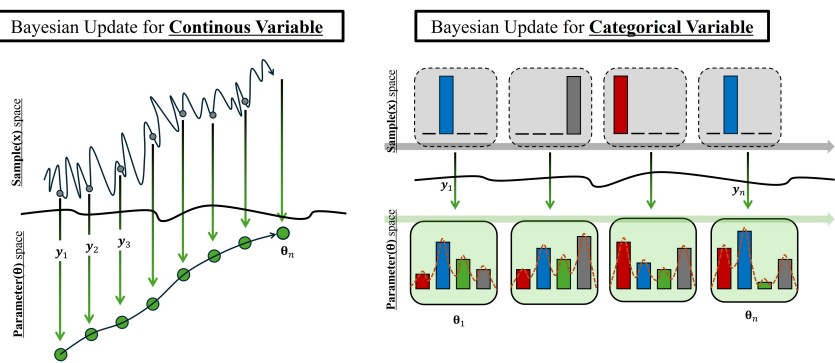

Figure 7: Schematic illustration of Bayesian updates for each variable type

sample $y_i$, implicitly encoding uncertainty through the random sampling of $y_i$. Specifically, for continuous variables, the sender distribution $p_{\mathrm{s}}(y_i \mid x; \alpha_i)$ is typically Gaussian, with precision (inverse variance) controlled by $\alpha_i$. A small value of $\alpha_i$ implies high noise, rendering the observed value $y_i$ nearly independent of the true data $x$, whereas a large $\alpha_i$ indicates a highly informative (low-noise) observation. Formally, the BFN update can be expressed as:

$$\theta_i = \left(1 - \frac{\alpha_i}{\rho_i}\right)\theta_{i-1} + \frac{\alpha_i}{\rho_i}y_i, \quad \text{where} \quad \rho_i = \rho_{i-1} + \alpha_i. \tag{31}$$

This update rule can be directly interpreted through the lens of Kalman filtering for scalar Gaussian models. In this analogy, each update of $\theta_i$ is drawn toward the observed sample $y_i$ proportionally to the precision parameter $\alpha_i$, thus progressively reducing uncertainty as $\rho_i$ increases. Given the sampled $y_i$, the Bayesian update function remains strictly deterministic, and no additional random noise is explicitly introduced beyond the implicit stochasticity already present in $y_i$.

This principle of implicit uncertainty introduction also extends naturally to categorical variables. Suppose the parameter vector $\theta_i$ represents a categorical probability distribution over $K$ classes, i.e., $\theta_i = (p_i^{(1)}, p_i^{(2)}, \ldots, p_i^{(K)})$. In this case, the sender distribution can be viewed as emitting symbols $y_i$ according to a confusion matrix parameterized by $\alpha_i$. Specifically, we have:

$$p_{\mathrm{s}}(y_i = c \mid x = k; \alpha_i) \approx \begin{cases} \frac{1}{K}, & \text{when } \alpha_i = 0 \quad \text{(uninformative observation)}, \\ \mathbf{1}_{c=k}, & \text{as } \alpha_i \to \infty \quad \text{(perfectly informative)}. \end{cases} \tag{32}$$

Upon observing the symbol $y_i = c$, the Bayesian update for the parameter $\theta_i$ follows:

$$\theta_{i+1}(j) \propto \theta_i(j) \cdot \Pr_{\rho_i}[Y = c \mid X = j]. \tag{33}$$

Although $y_i$ is originally sampled as a continuous vector from a Gaussian distribution centered around a one-hot encoding of the true class, the Bayesian update interpretation treats $y_i$ as if it were decoded into a discrete class index $c = \arg\max_k y_k$. For rigorous derivation and proof of this decoding interpretation, we refer the reader to Graves et al. [18]. The resulting posterior update normalizes probabilities over all classes $j$, illustrating explicitly how the sampled observation $y_i$ reweights prior probabilities $\theta_i(j)$ based on the likelihood of observing $y_i$ under each class hypothesis $X = j$. Thus, the sampled variable $y_i$ serves directly as an informative mediator driving the parameter dynamics.

Unlike the arbitrary Gaussian noise $z$ used in diffusion processes, the sampled variable $y_i$ in BFN is inherently linked to the unknown true data $x$ through the sender distribution $p_{\mathrm{s}}(y_i \mid x)$. As the inference procedure progresses (i.e., as $i$ increases and noise decreases), $y_i$ provides increasingly precise information, causing the parameter $\theta_i$ to converge toward a distribution sharply concentrated around the true data. This iterative Bayesian updating mechanism effectively transfers information from the sample space into the parameter space.

In summary, two critical distinctions between BFN and diffusion-based generative methods emerge from our theoretical analysis:

1. **Implicit versus Explicit Noise Injection:** BFN deterministically update parameters given noisy observations, implicitly capturing uncertainty. In contrast, diffusion models explicitly add random noise to samples at each step to maintain stochasticity.

2. **Parameter-space versus Sample-space Updates:** Diffusion models perform both training and inference entirely within the sample space. Conversely, BFN operate fundamentally within the parameter space, integrating information from the sample space through occasional noisy observations, thus creating a natural integration of information between the two spaces.

## C   Related Work

### C.1   Molecule Generation in Structure-based Drug Design

With the rapid accumulation of protein structural data, generative methods for molecule design have become increasingly important in structure-based drug discovery. Initial approaches, such as [44], employed sequence-based generative models to produce SMILES representations informed by protein binding sites. However, the advent of powerful geometric and 3D modeling methods has shifted the paradigm toward directly constructing molecules within three-dimensional spaces. For

example, [38] represented molecules using voxelized atomic density grids and utilized Variational Autoencoders (VAEs) for molecular synthesis. Meanwhile, [31], [34], and [36] introduced sequential, autoregressive approaches for placing atoms or functional groups step-by-step into target binding pockets. Building upon these autoregressive techniques, fragment-based strategies, as seen in FLAG [49] and DrugGPS [48], integrated chemically meaningful fragments, thereby improving the structural realism of generated ligands.

In parallel, diffusion-based generative methods have emerged, achieving remarkable success across various generative tasks such as image and text synthesis. Adaptations of these models in molecular contexts, exemplified by recent studies [20, 39, 25, 21], iteratively refine atom identities and coordinates, leveraging symmetry-preserving architectures such as SE(3)-equivariant neural networks to ensure chemical validity and structural accuracy.

Despite substantial advancements, current generative frameworks frequently encounter difficulties in simultaneously optimizing multiple pharmacologically relevant properties, including binding affinity, synthetic accessibility, and low toxicity. In practical drug development scenarios, the simultaneous control and optimization of these attributes are typically mandatory requirements rather than optional criteria [9]. Thus, developing generative strategies capable of effectively incorporating multiple property constraints remains a critical challenge in the field.

### C.2 Optimization based Molecule Generation in Structure-based Drug Design

Recent research in molecular generative modeling has moved beyond approximating training data distributions toward explicit optimization strategies aimed at producing molecules with desirable properties, such as high target protein binding affinity and synthetic accessibility (SA).

For example, RGA [16] employs genetic algorithms specifically tailored to structure-based drug design (SBDD), explicitly incorporating target protein structures into molecular optimization. DecompOpt [51] combines a pre-trained, structure-aware equivariant diffusion model to initially identify suitable molecular substructures complementary to target binding pockets, followed by a docking-based greedy iterative optimization to enhance binding affinity. TacoGFN [43] leverages generative flow networks (G-FlowNets) to identify pharmacophoric interactions with target proteins, subsequently utilizing reinforcement learning to optimize binding affinity and synthetic accessibility. Additionally, ALIDiff [19] introduces a preference-based optimization method that aligns pre-trained generative models to specified molecular properties through fine-tuning.

Despite their demonstrated effectiveness, these approaches share an intrinsic limitation: since optimization methods are tightly integrated into their training processes, adapting the models to new property requirements inevitably necessitates retraining. In contrast, our proposed approach distinguishes itself by leveraging a pre-trained generative model, enabling effective multi-property optimization directly through modifications at sampling time, thus avoiding costly retraining and enhancing flexibility in targeting diverse pharmacologically relevant properties.

### C.3 Bayesian Flow Network

Recently, Bayesian Flow Network (BFN) have gained attention as effective models for protein sequence modeling [2] and molecular structure generation [46, 37], demonstrating promising capabilities particularly in the generation of realistic three-dimensional molecular structures. Despite these advancements, the development of controllable generation methods within BFN, aimed at optimizing diverse molecular properties required for viable drug candidates, remains largely unexplored. Thus, establishing a theoretical link between generative models based on maximum likelihood estimation, such as diffusion models and BFN, and subsequently formulating gradient-guidance strategies within the BFN framework, represents an essential step forward. Such advancements could significantly inform future directions in the field of structure-based drug design.

## D  Implement Detail

### D.1  Predictor with Bayesian Neural Network for Uncertainty

To achieve property-driven generation without retraining the generative model, we introduce an external Bayesian neural network (BNN) predictor modeling the conditional distribution $p(\mathtt{l} \mid \mathbf{m}, \mathbf{p})$ as a Gaussian distribution $\mathcal{N}\big(\mathtt{l}; \boldsymbol{\mu}(\mathbf{m}, \mathbf{p}), \boldsymbol{\sigma}(\mathbf{m}, \mathbf{p})^2\big)$ with mean $\boldsymbol{\mu}_\vartheta(\mathbf{m}, \mathbf{p})$ and variance $\boldsymbol{\sigma}_\vartheta(\mathbf{m}, \mathbf{p})^2$. Unlike point estimates, this BNN provides uncertainty-aware predictions for properties (e.g., binding affinity), enabling informed guidance. During molecule generation, the gradient $\nabla_\mathbf{m} \log p(\mathtt{l} \mid \mathbf{m}, \mathbf{p})$,

which incorporates predictive uncertainty, guides the generative process. This uncertainty integration allows the model to appropriately moderate its guidance, preventing overly confident predictions toward regions unsupported by the predictor.

We approximate the overall predicted label distribution as Gaussian with predictive mean and variance computed from these samples. In particular, the mean is $\hat{\boldsymbol{\mu}}_\vartheta(\mathbf{x}, \mathbf{p}) = M^{-1} \sum_{i=1}^{M} \boldsymbol{\mu}_{\vartheta,i}(\mathbf{x}, \mathbf{p})$, and the predictive variance $\hat{\boldsymbol{\sigma}}_\vartheta^2(\mathbf{x}, \mathbf{p}) = M^{-1} \sum_i \left( \boldsymbol{\sigma}_{\vartheta,i}^2(\mathbf{x}) + \boldsymbol{\mu}_i^2(\mathbf{x}) \right) - \boldsymbol{\mu}_{\vartheta,i}^2(\mathbf{x})$ combines the average of the BNN's output variances with the variance of its output means. Using the law of total variance, we decompose the predictive uncertainty into aleatoric and epistemic components:

$$
\begin{aligned}
\boldsymbol{\sigma}_\vartheta^2(\mathbf{m}, \mathbf{p}) &= M^{-1} \sum_i \boldsymbol{\sigma}_{\vartheta,i}^2(\mathbf{m}, \mathbf{p}) + M^{-1} \sum_i \boldsymbol{\mu}_i^2(\mathbf{m}, \mathbf{p}) - \boldsymbol{\mu}_{\vartheta,i}^2(\mathbf{m}, \mathbf{p}) \\
&= \mathbb{E}\left[\boldsymbol{\sigma}_\vartheta^2(\mathbf{m}, \mathbf{p})\right] + \mathbb{E}\left[\boldsymbol{\mu}^2(\mathbf{m}, \mathbf{p})\right] - \mathbb{E}\left[\boldsymbol{\mu}_\vartheta(\mathbf{m}, \mathbf{p})\right]^2 \\
&= \underbrace{\mathbb{E}\left[\boldsymbol{\sigma}_\vartheta^2(\mathbf{m}, \mathbf{p})\right]}_{\text{Aleatoric Uncertainty}} + \underbrace{\text{Var}\left[\boldsymbol{\mu}(\mathbf{m}, \mathbf{p})\right]}_{\text{Epistemic Uncertainty}}
\end{aligned}
\tag{34}
$$

where $\vartheta$ represents the BNN's parameters (random due to the weight posterior). The first term $\sigma_{\text{aleatoric}}^2$ is the expected predictive variance (reflecting inherent noise or irreducible uncertainty in the property given $(\mathbf{x}, \mathbf{p})$), while the second term $\sigma_{\text{epistemic}}^2$ is the variance of the predicted means (reflecting uncertainty in the model parameters due to limited training data). Our CBYG can thus modulate the influence of the guidance signal in proportion to the predictor's confidence, improving robust controllability.

We train the property predictor on an external dataset of protein–ligand complexes, allowing it to learn a mapping from 3D structures to property values independent of the generative model. In particular, we use the CrossDocked2020 dataset [15] to train the predictor. Each training sample provides a protein structure $\mathbf{p}$, a ligand $\mathbf{m}$, and a ground-truth label $\mathbf{l}$ (e.g., an experimental or docking-derived affinity score). We optimize the BNN by maximizing the likelihood of the true labels under its predicted Gaussian distribution. The negative log-likelihood (NLL) loss for a single sample is given by (For notational simplicity, we omit $\mathbf{p}$):

$$
\mathcal{L}_{\text{NLL}}\left(\mathbf{y}_n, \mathbf{x}_n\right) = \frac{\log \boldsymbol{\sigma}_\vartheta^2\left(\mathbf{x}_n\right)}{2} + \frac{\left(\boldsymbol{\mu}_\vartheta\left(\mathbf{x}_n\right) - \mathbf{y}_n\right)^2}{2\boldsymbol{\sigma}_\vartheta^2\left(\mathbf{x}_n\right)},
\tag{35}
$$

where we omit the constant $\frac{1}{2}\log(2\pi)$ for brevity. In addition to the standard NLL, we also employ a $\beta$-weighted NLL variant (denoted $\beta$-NLL) to improve training stability in the presence of heteroscedastic uncertainty [41], as follow:

$$
\mathcal{L}_{\beta-\text{NLL}}\left(\mathbf{y}_n, \mathbf{x}_n\right) = \text{stop}\left(\sigma^{2\beta}\right) \mathcal{L}_{\text{NLL}}\left(\mathbf{y}_n, \mathbf{x}_n\right),
\tag{36}
$$

for some hyperparameter $\beta > 0$. Setting $\beta = 0$ recovers the original NLL loss, while a positive $\beta$ increases the relative penalty for large predicted variances.

### D.2 Implement detail

The generative backbone architecture employed in our proposed CByG framework directly inherits the architecture and pretrained weights from MolCRAFT [37]. For the property predictor backbone, we adapted the graph transformer architecture from TargetDiff [20] by removing the equivariant head, resulting in a novel SE(3)-invariant graph transformer. This design choice naturally aligns with the inherent symmetry of protein-ligand complexes, where binding affinity and synthetic accessibility scores remain invariant to rotations and translations.

Specifically, the property predictor comprises 16 attention heads, with each attention block consisting of three SE(3)-invariant layers featuring a hidden dimension of 64. Key and value embeddings are generated through a two-layer MLP. Layer normalization is uniformly applied throughout the network, and the Swish activation function is adopted. The learning rate is exponentially decayed by a factor of 0.6, with a lower bound set at $1 \times 10^{-6}$. This decay is triggered if no improvement is observed in the validation loss for ten consecutive evaluations, which occur every 1000 training steps. Finally, the property predictor's scoring function is defined as Score $= \left(\frac{\text{DS}}{-20} \times \text{SA}\right)$, where DS means docking score and SA means synthetic accessibility score.

### D.3 Sampling

---

**Algorithm 1** Sampling Procedure CBYG

---

1: **Input:**
2: Protein pocket $\mathbf{p}$,
3: Pre-trained output network:
4: $\quad p_0(\mathbf{m} \mid \boldsymbol{\theta}_{i-1}, \mathbf{p}; t) \leftarrow \Psi_{\texttt{output}}(\boldsymbol{\theta}_{i-1}, \mathbf{p}, t_i)$,
5: Property predictor network providing property predictions and uncertainty:
6: $\quad p_1(\mathbf{l} \mid \mathbf{m}, \mathbf{p}) \leftarrow \mathcal{N}\left(\mathbf{l}; \boldsymbol{\mu}_\vartheta(\mathbf{m}, \mathbf{p}), \boldsymbol{\sigma}_\vartheta(\mathbf{m}, \mathbf{p})^2\right)$,
7: Pre-defined precision schedule for coordinate and type $(\alpha_{\mathbf{x},i}, \alpha_{\mathbf{v},i})$ according to [18],
8: Guidance scale for coordinate and type $\leftarrow \lambda_{\mathbf{x}}, \lambda_{\mathbf{v}}$
9: $\boldsymbol{\theta}_0^{\mathbf{x}}, \boldsymbol{\theta}_0^{\mathbf{v}} \leftarrow \mathbf{0}, \left[\frac{1}{K}\right]_{\mathbb{N}_M \times K}$
10: **for** $i = 1$ to $N$ **do**
11: $\quad t \leftarrow \frac{i-1}{n}$
12: $\quad \mathbf{m} : [\hat{\mathbf{x}}, \hat{\mathbf{v}}] \sim p_0(\mathbf{m} \mid \boldsymbol{\theta}_{i-1}, \mathbf{p}; t)$
13: $\quad \mathbf{y}_{\mathbf{x},i} \sim p_{\mathbf{s}}(\mathbf{y}_i \mid \hat{\mathbf{x}}; \alpha_{\mathbf{x},i})$
14: $\quad \mathbf{y}_{\mathbf{v},i} \sim p_{\mathbf{s}}(\mathbf{y}_i \mid \hat{\mathbf{v}}; \alpha_{\mathbf{v},i})$
15: $\quad \boldsymbol{\mu}_\vartheta, \boldsymbol{\sigma}_\vartheta^2 \leftarrow p_1(\mathbf{l} \mid \mathbf{m}, \mathbf{p})$ $\qquad\qquad\qquad$ ▷ Mean and uncertainty from property predictor
16: $\quad \boldsymbol{\theta}_i^{\mathbf{x}} \leftarrow \frac{\alpha_i}{\rho_i} \mathbf{y}_{\mathbf{x},i} + \frac{\rho_{i-1}}{\rho_i} \boldsymbol{\theta}_{i-1}^{\mathbf{x}} + \boldsymbol{\sigma}_\vartheta^2 \cdot \lambda_{\mathbf{x}} \cdot \frac{1}{\rho_i} \nabla_{\mathbf{x}} \log p_1(\mathbf{l} \mid [\hat{\mathbf{x}}, \hat{\mathbf{v}}], \mathbf{p})$
17: $\quad \boldsymbol{\theta}_i^{\mathbf{v}} \leftarrow \mathrm{Softmax}\left(e^{\mathbf{y}_{\mathbf{v},i}} \cdot \boldsymbol{\theta}_{i-1}^{\mathbf{v}} \cdot e^{\mathbf{h}}\right)$
$\qquad\qquad$ where $\mathbf{h} = \boldsymbol{\sigma}_\vartheta^2 \cdot \lambda_{\mathbf{v}} \cdot \nabla_{\mathbf{e}_{\mathbf{v}}} \log p_1(\mathbf{l} \mid [\hat{\mathbf{x}}, \hat{\mathbf{v}}], \mathbf{p}), \mathbf{e}_{\mathbf{v}} = \mathrm{GumbelSoftmax}(\hat{\mathbf{v}})$
18: **end for**
19: $\mathbf{m} \sim p_0(\mathbf{m} \mid \boldsymbol{\theta}_N, \mathbf{p}; t)$
20: **return** $[\hat{\mathbf{x}}, \hat{\mathbf{v}}]$

---

### D.4 SE(3)-Equivariance

Since our proposed guidance injection strategy is integrated into an SE(3)-equivariant generative model, the designed guidance itself must inherently preserve the SE(3)-equivariance property. As described by [40], aligning the protein-ligand complex to center the pocket at the origin removes translation equivariance, requiring only O(3)-equivariance. The following provides a formal proof verifying this property.

**Proposition D.1.** *Suppose the property preditor $\Psi_{\texttt{prop}}([\mathbf{x}_M, \mathbf{v}_M], [\mathbf{x}_P, \mathbf{v}_P]) \leftarrow p_1(\mathbf{l} \mid \mathbf{m}, \mathbf{p})$ is invariant such that $\Psi_{\texttt{prop}}([\mathbf{x}_M, \mathbf{v}_M], [\mathbf{x}_P, \mathbf{v}_P]) = \Psi_{\texttt{prop}}([T_g(\mathbf{x}_M), \mathbf{v}_M], [T_g(\mathbf{x}_P), \mathbf{v}_P])$. Denoting $T_g$ as the group of O(3)-transformation, $T_g(\mathbf{x}) = \mathbf{Rx}$, where $\mathbf{R} \in \mathbb{R}^{3 \times 3}$ is the rotation matrix, and $\mathbf{b} \in \mathbb{R}^3$ is the translation vector. Then, gradient guidance function is orthogonal equivariant such that $\nabla_{\mathbf{x}_M} \Psi_{\texttt{prop}}(T_g(\mathbf{x}_M), T_g(\mathbf{x}_P)) = T_g(\nabla_{\mathbf{x}_M} \Psi_{\texttt{prop}}(\mathbf{x}_M, \mathbf{x}_P))$. Variables corresponding to type $\mathbf{v}$ are omitted from the notation, as they remain unaffected by O(3) transformations.*

*Proof.* Given the invariance of the property predictor $\Psi_{\texttt{prop}}(\mathbf{x}_M, \mathbf{x}_P)$ under O(3) transformations, it follows that $\Psi_{\texttt{prop}}(\mathbf{Rx}_M, \mathbf{Rx}_P) = \Psi_{\texttt{prop}}(\mathbf{x}_M, \mathbf{x}_P)$. Differentiating both sides of the equation with respect to $\mathbf{x}_M$ yields:

$$\Psi_{\texttt{prop}}(\mathbf{x}_M, \mathbf{x}_P) = \Psi_{\texttt{prop}}(\mathbf{Rx}_M, \mathbf{Rx}_P) \tag{37}$$

$$\nabla_{\mathbf{x}_M} \Psi_{\texttt{prop}}(\mathbf{x}_M, \mathbf{x}_P) = \nabla_{\mathbf{x}_M} \Psi_{\texttt{prop}}(\mathbf{Rx}_M, \mathbf{Rx}_P)$$

$$= \left(\frac{\partial(\mathbf{Rx}_M)}{\partial \mathbf{x}_M}\right)^\top \nabla_{\mathbf{x}_M} \Psi_{\texttt{prop}}(\mathbf{Rx}_M, \mathbf{Rx}_P)$$

$$= \mathbf{R}^\top \nabla_{\mathbf{x}_M} \Psi_{\texttt{prop}}(\mathbf{Rx}_M, \mathbf{Rx}_P) \tag{38}$$

$$\mathbf{R} \nabla_{\mathbf{x}_M} \Psi_{\texttt{prop}}(\mathbf{x}_M, \mathbf{x}_P) = \mathbf{RR}^\top \nabla_{\mathbf{x}_M} \Psi_{\texttt{prop}}(\mathbf{Rx}_M, \mathbf{Rx}_P)$$

$$= \nabla_{\mathbf{x}_M} \Psi_{\texttt{prop}}(\mathbf{Rx}_M, \mathbf{Rx}_P) \tag{39}$$

Therefore, $\nabla_{\mathbf{x}_M} \Psi_{\texttt{prop}}(\cdot)$ exhibits equivariance under the transformation $T_g(\cdot)$, completing the proof. $\qquad\square$

## E *TargetOpt* Implement Detail

To evaluate the effectiveness of our proposed guidance injection within the Bayesian Flow Network framework, we implemented a comparable guidance method within a diffusion-based generative

model. Specifically, we adopted the TargetDiff [20] architecture, enhancing it with gradient-based guidance strategies. Two distinct gradient guidance approaches were considered: (1) gradient computation based on property prediction at arbitrary intermediate time steps $\mathbf{x}_t$, and (2) gradient computation via posterior sampling, leveraging property prediction at the fully denoised state $\hat{\mathbf{x}}_0$.

For fair comparison, the model utilized for property prediction at $\hat{\mathbf{x}}_0$ was identical to that employed in our CBʏG model. Below, we present the unconditional denoising procedures used by TargetDiff for each variable type; additional details can be found in the original paper [20].

$$q\left(\mathbf{x}_{t-1} \mid \mathbf{x}_t, \mathbf{x}_0\right) = \mathcal{N}\left(\mathbf{x}_{t-1}; \tilde{\boldsymbol{\mu}}_t\left(\mathbf{x}_t, \mathbf{x}_0\right), \tilde{\beta}_t \mathbf{I}\right), \quad q\left(\mathbf{v}_{t-1} \mid \mathbf{v}_t, \mathbf{v}_0\right) = \mathcal{C}\left(\mathbf{v}_{t-1} \mid \tilde{\boldsymbol{c}}_t\left(\mathbf{v}_t, \mathbf{v}_0\right)\right) \tag{40}$$

, where $\tilde{\boldsymbol{\mu}}_t\left(\mathbf{x}_t, \mathbf{x}_0\right) = \frac{\sqrt{\bar{\alpha}_{t-1}}\beta_t}{1-\bar{\alpha}_t}\mathbf{x}_0 + \frac{\sqrt{\alpha_t}(1-\bar{\alpha}_{t-1})}{1-\bar{\alpha}_t}\mathbf{x}_t$, $\tilde{\beta}_t = \frac{1-\bar{\alpha}_{t-1}}{1-\bar{\alpha}_t}\beta_t$, and $\tilde{\boldsymbol{c}}_t\left(\mathbf{v}_t, \mathbf{v}_0\right) = \boldsymbol{c}^* / \sum_{k=1}^K c_k^*$ and $\boldsymbol{c}^*\left(\mathbf{v}_t, \mathbf{v}_0\right) = [\alpha_t \mathbf{v}_t + (1-\alpha_t)/K] \odot [\bar{\alpha}_{t-1}\mathbf{v}_0 + (1-\bar{\alpha}_{t-1})/K]$.

Under the scenario of property prediction at arbitrary intermediate time steps $\mathbf{x}_t$, the update procedure for atomic coordinate and type can be modified as follows.

$$\tilde{\boldsymbol{\mu}}_t\left(\mathbf{x}_t, \mathbf{x}_0\right) = \frac{\sqrt{\bar{\alpha}_{t-1}}\beta_t}{1-\bar{\alpha}_t}\mathbf{x}_0 + \frac{\sqrt{\alpha_t}\left(1-\bar{\alpha}_{t-1}\right)}{1-\bar{\alpha}_t}\mathbf{x}_t + \frac{1-\alpha_t}{\sqrt{\alpha_t}}\nabla_{\mathbf{x}_t}\log p\left(1 \mid \boldsymbol{x}_t, \mathbf{p}\right) \tag{41}$$

$$\tilde{\boldsymbol{c}}_t\left(\mathbf{v}_t, \mathbf{v}_0\right) = \left(\frac{\boldsymbol{c}^*}{\sum_{k=1}^K c_k^*} + \boldsymbol{\delta}\right) \cdot e^{\nabla_{\mathbf{v}_t} p(1|\mathbf{v}_t, \mathbf{p})} \tag{42}$$

Additionally, when employing posterior sampling for gradient computation, the gradient form can be reformulated analogous to DPS as follows.

$$\nabla_{\mathbf{x}_t}\log p\left(1 \mid \mathbf{x}_t\right) \simeq \nabla \log \mathbb{E}_{\mathbf{x}_0 \sim p(\mathbf{x}_0|\mathbf{x}_t)}\left[p\left(y \mid \hat{\mathbf{x}}_0\right)\right] \tag{43}$$

$$\nabla_{\mathbf{v}_t}\log p\left(1 \mid \mathbf{v}_t\right) \simeq \nabla \log \mathbb{E}_{\mathbf{v}_0 \sim p(\mathbf{v}_0|\mathbf{v}_t)}\left[p\left(y \mid \hat{\mathbf{v}}_0\right)\right] \tag{44}$$

# F   Baseline Model for Evaluation

For a fair comparison, we categorize baseline models into two groups: "Only Generation Type" and "Optimization based Generation Type". Movevoer, we selected state-of-the-art generative models exhibiting competitive performance as representative models for the "Only Generation Type" category.

## Only Generation Type

- **AR** [31] leverages Markov chain Monte Carlo techniques to sequentially infer molecular structures from spatial atomic density representations.
- **Pocket2Mol** [34] incrementally synthesizes molecules through sequential prediction of atoms and associated bonds using an E(3)-equivariant architecture, selectively expanding frontier atoms to significantly improve sampling efficiency.
- **TargetDiff** [20] enhances dual-modality diffusion methodologies, distinctly processing continuous and discrete modalities through parallel diffusion pipelines, demonstrating improved outcomes over purely continuous formulations such as DiffSBDD.
- **DecompDiff** [21] adopts molecular decomposition techniques, separating the molecular structure into functional arms and connective scaffolds, thereby integrating chemically-informed priors within diffusion-based generative mechanisms.
- **MolCRAFT** [37] exploits Bayesian Flow Networks coupled with sophisticated sampling schemes, showing significant enhancements relative to contemporary diffusion-based methodologies. *We directly utilized the publicly available molecules provided through their respective GitHub repositories for our experiments.*

## Optimization based Generation Type

- **RGA** [16] extends the evolutionary optimization approach of AutoGrow4 by integrating a reinforcement learning-based policy conditioned on target binding pockets, effectively constraining exploratory randomness during molecular search procedures. *Since the molecules generated by these models for CrossDocked2020 benchmark are not publicly available on GitHub, we trained these models ourselves to produce molecules for evaluation.*

- **DecompOpt** [51] employs conditional generative modeling of chemically meaningful fragments aligned with receptor sites, iteratively optimizing molecular generation via guided resampling within a structured diffusion latent space, informed by fragment-based oracle rankings. *Since the molecules generated by these models for CrossDocked2020 benchmark are not publicly available on GitHub, we trained these models ourselves to produce molecules for evaluation.*

- **TacoGFN** [42] leverages a G-FlowNet-based autoregressive approach, incrementally assembling molecular structures fragment-by-fragment while identifying key pharmacophoric interactions with target proteins. It integrates a reward-based optimization mechanism, simultaneously promoting advantageous properties such as binding affinity and synthetic accessibility (SA score) throughout the generative process. *We directly utilized the publicly available molecules provided through their respective GitHub repositories for our experiments.*

- **ALIDiff** [19] is an SE(3)-equivariant diffusion generative model that incorporates recent reinforcement learning from human feedback techniques, leveraging Energy Preference Optimization to effectively generate molecules exhibiting superior properties such as enhanced binding affinity and other desirable attributes. *We directly utilized the publicly available molecules provided through their respective GitHub repositories for our experiments.*

- **TargetOpt** is a diffusion-based generative model developed specifically in this study to enable gradient-based guidance propagation, serving as a comparative baseline for evaluating the effectiveness of guidance mechanisms between BFN and diffusion-based frameworks. Although several studies have adopted similar optimization strategies, we exclude these approaches from consideration, as they either exclusively optimize for binding affinity or neglect guidance on categorical atom types.

## G    Additional Experiment Metrics

### G.1    Experimental Setup of Section 6.2

We employ multiple metrics to comprehensively evaluate the binding affinity and intrinsic properties of the generated molecules. Specifically, we utilize SMINA, GNINA, and AutoDock Vina to independently calculate two distinct forms of binding affinity: (1) the intrinsic docking scores of the generated molecules themselves (denoted as Score.), and (2) the affinity scores obtained via re-docking procedures (denoted as Dock.). Additionally, we introduce the High Affinity metric, defined as the percentage of generated molecules exhibiting superior binding affinity compared to a given reference ligand for each target protein. To quantify the intrinsic molecular properties, we measure the Synthetic Accessibility (SA) and Diversity metrics. Finally, to capture the overall stability and validity of generated molecules (both intrinsically and in complex with the target protein) we utilize the PB-valid score from the PoseBusters benchmark. The metric PB-valid denotes the proportion of generated molecules considered valid under the PoseBusters benchmark, assuming that violation of any of its 17 evaluation criteria renders the molecule invalid.

### G.2    Experimental Setup of Section 6.3

We quantitatively evaluate the synthetic feasibility of generated molecules using six key metrics provided by AiZynthFinder (Solved, Routes, Solved Routes, and Top Score). The reported values in Table 2 are averages computed across all generated molecules per model. Specifically, Solved indicates whether AiZynthFinder successfully identified at least one valid retrosynthetic route for a given molecule. A higher number of nodes indicates a more extensive exploration of possible reaction pathways by the algorithm, which may reflect the inherent complexity of the target molecule, diversity in applicable reaction templates, or increased search depth. While a large node count can imply a thorough and comprehensive search, it might also signal inefficiencies if numerous unproductive pathways were evaluated. Thus, this metric provides valuable insights into the balance between computational effort and search comprehensiveness.

Routes counts the total number of distinct retrosynthetic pathways (complete reaction sequences from purchasable precursors to the target molecule) identified by AiZynthFinder. This metric quantifies the diversity of retrosynthetic solutions identified by the algorithm. A higher number of identified routes suggests multiple viable synthetic strategies for the target molecule, providing chemists with alternative synthetic options. Nevertheless, not all identified routes may be equally practical or feasible; thus, this metric should be interpreted alongside complementary measures such as the number of solved routes or the top-scoring pathways.

Solved Routes is the subset of these routes comprising exclusively purchasable precursors listed in commercially available databases (e.g., ZINC), thus representing practically realizable synthetic pathways. This metric enables rapid assessment of synthetic feasibility given a predefined inventory of building blocks. Specifically, if at least one retrosynthetic pathway is successfully identified, the target molecule is considered synthesizable, making this a straightforward, high-level indicator of synthetic achievability. However, as it does not capture pathway quality or diversity, it should be interpreted in conjunction with complementary metrics.

Lastly, Top Score reflects the highest-ranked synthetic route as evaluated by AiZynthFinder's scoring function, which aggregates criteria such as precursor availability, reaction step count, and reaction feasibility (e.g., average template frequency). This metric quantitatively represents the quality of the highest-ranked synthetic route, assisting chemists in prioritizing retrosynthetic pathways for consideration. A higher score reflects routes with greater feasibility, efficiency, and desirability. This measure is particularly useful for comparing alternative pathways or selecting the most promising candidates for subsequent experimental validation.

## H  Additional Experiment result

### H.1  Experimental Analysis of Section 6.2

Our proposed model consistently outperforms baseline methods across 12 evaluation metrics covering binding affinity and intrinsic molecular properties, as shown Table 1. In particular, our model significantly outperforms baseline methods in the 'Score' metric, which measures the binding affinity of generated molecules prior to any docking procedure. This result indicates that our model inherently generates molecules possessing high binding affinity, even without additional docking optimization. Similar superior performance is observed in the Synthetic Accessibility (SA) and PoseBusters validity (PB-Valid) metrics, indicating that gradients derived from binding affinity and SA scores are effectively propagated during the guidance-based sampling process.

Furthermore, we observe that different baseline models excel depending on the affinity evaluation tool used; specifically, ALIDiff outperforms DecompDiff when evaluated by SMINA, whereas DecompDiff achieves better results when GNINA is used. Considering the variability in performance across different evaluation metrics, the consistently strong performance of our proposed model across all three docking tools (Vina, SMINA, and GNINA) highlights its robustness and reliability in generating molecules with high binding affinity, as well as its generalizable efficacy across diverse evaluation standards. Additionally, our model uniquely exhibits minimal differences in binding affinity between pre-docking and post-docking evaluations. This minimal discrepancy indicates our model's ability to intrinsically predict stable and energetically favorable binding poses, explicitly capturing meaningful protein-ligand interactions.

Table 4 demonstrates the effectiveness of jointly applying gradient-based guidance to both atomic coordinates and atom types. As clearly indicated by the results, simultaneous guidance across both modalities consistently outperforms methods employing guidance on a single modality alone. This outcome aligns with fundamental chemical principles, as molecular properties and the corresponding energy landscape inherently depend upon intricate interactions between atomic types and their spatial configurations.

In addition to the results presented in Figure 5, we conducted supplementary experiments using the PoseCheck benchmark to further assess the generated molecules' structural stability and validity. Crucially, molecules evaluated in this additional experiment were directly sampled from generative models, without employing docking. This methodological choice ensures that the evaluation reflects the inherent capability of the generative models, rather than improvements arising from docking-based optimizations or adjustments by docking tools. Their reliance on external docking for generating final 3D conformations makes it unsuitable to accurately evaluate these models from the perspective of generating intrinsically stable 3D molecular structures. Consequently, models such as RGA and TacoGFN, which initially generate molecules as SMILES strings or 2D graphs and subsequently rely on docking software to derive the final 3D conformations, were excluded from this comparative analysis.

Experimental results (Table 5) indicate that the CByG model outperforms baseline models in terms of the "Clash" and "Strain Energy" metrics, whereas DecompOpt achieves the best performance in the "Intermolecular Interaction" metric. Considering that DecompOpt explicitly optimizes molecules by fixing protein-interacting fragments, this result aligns naturally with its optimization strategy.

Table 4: Summary of binding affinity and molecular properties of reference molecules and molecules generated by CBYG and baselines. (↑)/(↓) denotes whether a larger/smaller number is preferred. Top 2 results are bolded and underlined, respectively.

| Methods | SMINA (↓) | | GNINA (↓) | | Vina (↓) | | SA (↑) | | PB-Valid (↑) | |
|---|---|---|---|---|---|---|---|---|---|---|
| | Score. | Dock. | Score. | Dock. | Score. | Dock. | Avg. | Med. | Avg. | Med. |
| Reference | -6.37 | -7.92 | -7.06 | -7.61 | -6.36 | -7.45 | 0.73 | 0.74 | 95.0% | 95.0% |
| **CBYG** | **-7.74** | **-9.61** | **-7.63** | **-8.33** | **-8.60** | **-9.16** | **0.84** | **0.87** | 94.9% | **96.0%** |
| CBYG w/o pos guidance | -7.05 | -8.60 | -6.88 | -7.42 | -7.74 | -8.12 | 0.78 | 0.79 | 90.4% | 91.1% |
| CBYG w/o type guidance | -6.92 | -8.45 | -6.70 | -7.20 | -7.60 | -7.95 | 0.76 | 0.77 | 88.3% | 89.0% |
| CBYG w/o uncertainty | -7.25 | -8.81 | -7.10 | -7.64 | -7.90 | -8.31 | 0.75 | 0.76 | 87.1% | 87.7% |
| **CBYG** ($\lambda_\mathbf{x} = 40, \lambda_\mathbf{v} = 40$) | **-7.74** | **-9.61** | **-7.63** | **-8.33** | **-8.60** | **-9.16** | **0.84** | **0.87** | 94.9% | **96.0%** |
| CBYG ($\lambda_\mathbf{x} = 30, \lambda_\mathbf{v} = 30$) | -7.13 | -8.64 | -6.95 | -7.47 | -7.79 | -8.17 | 0.79 | 0.81 | 91.0% | 91.4% |
| CBYG ($\lambda_\mathbf{x} = 50, \lambda_\mathbf{v} = 50$) | -7.33 | -8.94 | -7.22 | -7.79 | -8.07 | -8.49 | 0.74 | 0.76 | 85.6% | 86.2% |
| CBYG ($\lambda_\mathbf{x} = 30, \lambda_\mathbf{v} = 40$) | -7.28 | -8.89 | -7.15 | -7.68 | -7.98 | -8.42 | 0.77 | 0.78 | 89.2% | 89.7% |
| CBYG ($\lambda_\mathbf{x} = 40, \lambda_\mathbf{v} = 30$) | -7.10 | -8.59 | -6.92 | -7.42 | -7.76 | -8.10 | 0.80 | 0.82 | 91.8% | 92.3% |

Table 5: Posebusters results for all methods.

| | **CBYG** | TargetDiff | DecompDiff | MolCraft | DecompOpt | ALIDiff |
|---|---|---|---|---|---|---|
| Avg. Clash (↓) | **3.71** | 10.54 | 13.66 | 5.72 | 17.05 | 8.71 |
| Avg. Strain Energy (↓) | $\mathbf{5.85 \times 10^7}$ | $1.41 \times 10^{14}$ | $1.44 \times 10^9$ | $7.57 \times 10^{11}$ | $1.18 \times 10^{11}$ | $6.22 \times 10^{16}$ |
| Avg. Interaction (↑) | 15.65 | 17.15 | 18.26 | 15.92 | **18.77** | 17.74 |

Table 6: PB-Valid results for all metrics.

| Metric | Valid Score |
|---|---|
| Mol Pred Loaded & Sanitization & Aromatic Ring Flatness | 1.000 |
| All Atoms Connected | 0.985 |
| Bond Length & Bond Angle | 0.995 |
| Internal Steric Clash | 0.993 |
| Double Bond Flatness | 0.998 |
| Internal Energy | 0.970 |
| Protein-Ligand Maximum Distance & Minimum Distance to Organic Cofactors & Minimum Distance to Waters | 1.000 |
| Minimum Distance to Protein | 0.997 |
| Minimum Distance to Inorganic Cofactors | 0.994 |
| Volume Overlap with Protein & Volume Overlap with Organic Cofactors | 1.000 |
| Volume Overlap with Inorganic Cofactors & Volume Overlap with Waters | 1.000 |

## H.2 Experimental Analysis of Section 6.3

In this experiment, both our proposed model and the RGA model exhibited overall high performance. Here, it is important to consider molecular complexity in relation to binding affinity. Typically, molecules with greater complexity tend to possess higher binding affinity due to increased opportunities for intermolecular interactions with target proteins; conversely, simpler molecules usually exhibit lower affinity. Given this, the performance of the RGA model on the AiZynthFinder benchmark aligns logically with its relatively lower binding affinity scores reported in Table 1. Applying the same perspective to our proposed model, it is particularly noteworthy that our model not only achieves top-tier performance in binding affinity but also exhibits near state-of-the-art results on the AiZynthFinder benchmark. This indicates the ability of our approach to generate molecules that are simultaneously effective in terms of biological efficacy and practical retrosynthetic feasibility.

Interestingly, we observe that several models show no clear correlation between their performance on the AiZynthFinder benchmark and the SA score reported in Table 1. This highlights the necessity for evaluating synthetic feasibility in SBDD research using multiple diverse criteria beyond just the SA score. A notable concern is that despite high SA scores (approaching 0.8 for RGA and surpassing 0.8 for our proposed model) the fraction of molecules classified as synthetically feasible under the AiZynthFinder 'Solved' metric remains below 50%. This outcome suggests that synthetic feasibility deserves greater attention in future SBDD research, underscoring the need for broader consideration and deeper analysis of retrosynthetic practicality.

## H.3 Experimental Analysis of Section 6.4

As demonstrated in Figure 6, guidance scores obtained using the BFN-based approach consistently surpass those derived from diffusion-based methods across the entire generative trajectory. Furthermore, guidance scores from diffusion models utilizing predictions of the final clean state ($\hat{\mathbf{x}}_0$) exhibit marked instability, underscoring the robustness of the BFN-guided generation procedure.

Notably, diffusion-based methods (green and gray plots) yield higher absolute guidance scores when employing predictions of the final molecular states; however, these scores simultaneously exhibit increased variance as the generative process advances. This behavior highlights a fundamental trade-off within diffusion-based guidance strategies in 3D molecular generation: gradients derived from predicted clean states facilitate higher guidance scores but necessitate point estimation toward the final state in the sample space, inherently introducing instability into intermediate guidance steps. In contrast, BFN operates in a continuous parameter space rather than directly in sample space, enabling stable and continuous gradient propagation even for categorical variables. Consequently, BFN-based guidance using predicted final states provides inherently more stable gradient trajectories, making it particularly advantageous for robust and controllable 3D molecular generation.

### H.4 Experimental Analysis of Section 6.5

To evaluate the selectivity control capabilities of the proposed CBYG model, we conducted experiments using the selectivity benchmark set specifically constructed for this study. We primarily assessed and compared selectivity performance before and after applying guidance in two generative model categories: diffusion-based models and Bayesian Flow Network-based models. Notably, optimization-based generation models were excluded from this comparison due to their intrinsic requirement for retraining to optimize for different molecular properties, highlighting the versatility of our proposed model in addressing diverse property optimization objectives.

Molecule generation was directed toward enhancing binding affinity to designated on-target proteins, while guidance was explicitly designed to minimize binding affinity to specified off-target proteins. In Table 3, the "Succ.Rate" represents the proportion of generated molecules demonstrating superior affinity for the on-target protein relative to the off-target, whereas the "$\Delta$ Score" quantifies the differential affinity between on-target and off-target interactions.

Experimental results revealed that even without explicit selectivity-guidance, both model categories produced molecules with superior affinity toward the on-target protein in more than half of the generated cases. This outcome can be attributed to the inherent advantage of structure-based generative models, which explicitly encode and leverage the structural context of the target proteins during molecule generation. Nevertheless, Bayesian Flow Network-based models consistently demonstrated superior performance compared to diffusion-based models, and this advantage was markedly amplified when selectivity guidance was employed. These findings collectively underscore the efficacy and versatility of the proposed CBYG framework in achieving controlled generation not only for conventional metrics such as synthetic accessibility (SA score) but also for critical properties such as selectivity.

### H.5 Comparative Visualization of Generated Ligands

## I Selectivity Dataset

We constructed a selectivity-focused dataset based on kinase inhibitor selectivity data. Initially, we identified the selectivity profiles of 285 proteins across 38 kinase inhibitors, as reported in a study on the quantitative analysis of kinase inhibitor selectivity [27, 10]. Subsequently, for each inhibitor, we categorized the proteins into on-target and off-target groups and extracted their corresponding Entrez Gene Symbols. These gene symbols were then used to systematically gather protein-related data from the UniProt [8] database via REST APIs. The UniProt web crawling process was structured in three stages. First, the Entrez Gene Symbols were URL-encoded and filtered by the human taxonomy ID (9606) to retrieve corresponding UniProt IDs in JSON format. Second, protein sequences were obtained by querying the UniProt FASTA API, with FASTA headers subsequently removed. Third, ATP binding site information—including binding site positions and sequences—was extracted from the UniProt feature sections. The collected data, comprising UniProt IDs, sequences, and binding site details, were integrated using the initial set of 285 Entrez Gene Symbols as a reference. Proteins lacking ATP binding site data (six in total) were excluded, yielding a final dataset of 279 proteins prepared for further analysis. Protein structures were predicted using AlphaFold3 [1], and model structure files were retrieved in CIF format. These files were converted to PDB format using the Bio.PDB module of Biopython. The ATP binding site information was then employed to define and extract protein pocket structures, specifically targeting the binding site residues and the surrounding region within a 5Å radius. Structural similarity among the protein pockets was evaluated using TM-score and RMSD metrics. The TM-score quantifies topological similarity between protein structures, with values ranging from (0, 1]; scores above 0.5 typically indicate identical protein folds,

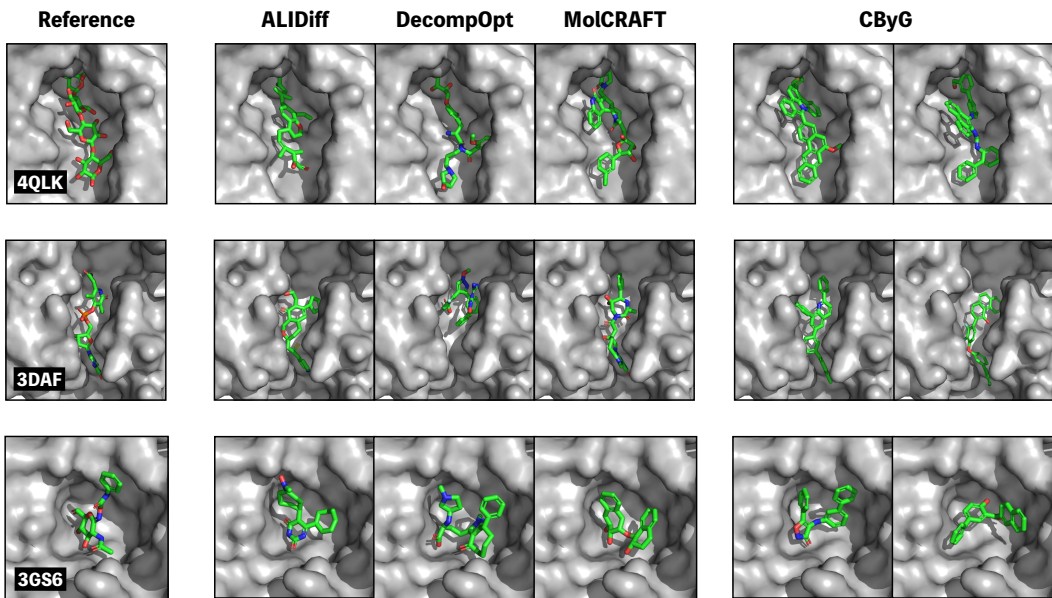

| Reference | ALIDiff | DecompOpt | MolCRAFT | CByG |

Figure 8: Visualization of generated ligands for protein pockets, with a reference molecule (left) and corresponding outputs from ALIDiff, DecompOpt, MolCRAFT, and CBYG.

while scores below 0.17 suggest unrelated structures. We classified and organized the extracted protein structures into directories based on a TM-score threshold of 0.4 and an RMSD of 1Å, reflecting structurally similar protein pockets suitable for downstream selectivity analyses. This process ultimately facilitated the construction of on-target (primary) and off-target pairs.

## J Rethinking

### J.1 Addressing Fundamental Challenges of Diffusion-based Guidance in 3D Molecular Generation

A core objective in Structure-based Drug Design (SBDD) is generating molecules that bind specifically to target proteins while simultaneously satisfying desired properties. Diffusion-based generative models are particularly suited to this task, as they can incorporate external predictors for property-guided sampling. Specifically, these models leverage guidance derived from predictors to direct the generative process toward property-specific regions of molecular space. However, as briefly mentioned in previous sections, 3D molecular structures inherently comprise hybrid data types, consisting of continuous variables (e.g., Cartesian coordinates) typically modeled by Gaussian distributions, and categorical variables (e.g., atom types such as oxygen or nitrogen) typically represented by categorical distributions. This hybrid nature presents fundamental challenges for conventional gradient-based guidance approaches.

First, since coordinates and atom types are sampled from fundamentally distinct distributions, guidance gradients tend to propagate independently across these data types. Consequently, the guidance mechanism often fails to accurately capture the critical chemical interdependencies between atomic coordinates and categorical atom identities, thereby undermining the chemical coherence of the generated molecules.

Second, categorical variables in diffusion models rely on discrete sampling processes involving an argmax operation at each reverse sampling step. Due to the discrete nature of argmax operations, direct application of gradient-based guidance becomes infeasible, as minor guidance gradients typically do not influence the argmax outcome unless excessively amplified. Yet, increasing the guidance scale excessively can cause the distribution during reverse sampling to become dominated by guidance gradients, resulting in unstable and unrealistic molecular structures. Alternatively, attempting to circumvent this issue by artificially converting categorical distributions into continuous or discretized variables introduces unnatural assumptions and significantly increases the complexity of model design.

Lastly, injecting guidance gradients directly into the denoising process, which operates in the molecular sample space, risks destabilizing intermediate molecular configurations. This instability arises due to the numerical sensitivity of 3D coordinates, potentially causing molecules to lose chemical validity and structural coherence during intermediate generation stages. Thus, this approach substantially hinders effective controllability over molecular properties, and leads to unstable molecular outcomes.

In the context of 3D molecules (with continuous coordinates and discrete atom types), diffusion-based generative guidance methods face several fundamental limitations.

First, guidance gradients often fail to capture interdependencies between modalities (e.g. coordinate updates and atom-type assignments may be misaligned if treated separately) as evidenced by the need for separate latent spaces or noise schedules for different variable types in prior diffusion approaches. Second, guidance in categorical diffusion is unstable and often ineffective: choosing atom types via an argmax during the denoising process introduces a discontinuous, non-differentiable operation that disrupts gradient-based optimization. Third, the denoising of spatial coordinates is structurally fragile – adding noise to atomic positions can break chemical bonds or distort interatomic distances beyond physical limits, leaving intermediate states chemically invalid and uninformative. Given these challenges, BFN offer a promising alternative for property-guided molecular generation. BFN operate in a fully differentiable parameter space and provide a unified probabilistic treatment of continuous and categorical modalities, thereby inherently modeling cross-modal dependencies and avoiding the need for modality-specific hacks. In contrast to diffusion, BFN do not require per-step argmax sampling for atom types; instead, they maintain a probability simplex representation for categorical variables, preserving gradient information throughout the generative process. This unified and differentiable approach enables stable gradient-based guidance on molecular properties, making BFN a robust paradigm for 3D molecule generation under complex hybrid objectives.

## J.2 Rethinking Posterior Guidance: From Intermediate States to Predicted Final Structures

In generative frameworks such as diffusion models, conditional generation typically involves controlling the generative process by leveraging gradient guidance from a posterior conditioned on labels (attributes) l. According to the theoretical foundations of the reverse process, the introduced posterior term can be represented as $\nabla_{\mathbf{x}_t} p(\mathbf{1} \mid \mathbf{x}_t)$, commonly known as the conditional score function, which is usually learned using a dedicated neural network. Here, it is crucial to reassess whether the intermediate state $x_t$, employed as input for the conditional score function, is a sensible variable for attribute prediction [11, 45, 22]. In domains like image generation, where score-based diffusion models were initially introduced, intrinsic structural characteristics of the data enable meaningful predictions of labels even from intermediate noisy states. Thus, utilizing the conditional score function in the form $p(\mathbf{1} \mid \mathbf{x}_t)$ has proven reasonable in these contexts.

However, unlike image data, intermediate states of 3D molecular structures with added noise lose their chemical validity, rendering derived molecular properties essentially meaningless. Consequently, employing a posterior conditioned directly on the intermediate noisy state $\mathbf{x}_t$, i.e., $p(\mathbf{1} \mid \mathbf{x}_t)$, is fundamentally unreasonable as guidance for molecule generation tasks. To overcome this limitation, recent studies have adopted a posterior sampling strategy, originally proposed in inverse problems within the image generation domain [7, 23]. Specifically, these methods predict the final, noise-free molecular structure $x_0$ and leverage this prediction to guide gradient-based generation, i.e., $p(\mathbf{1} \mid \hat{\mathbf{x}}_0)$. Further efforts have also extended such posterior sampling approaches to the conditional generation of 3D molecules. However, existing methods primarily focus on general conditional molecular generation rather than the specialized task of structure-based drug design (SBDD), which involves molecular binding to specific proteins. Therefore, substantial modifications and further methodological advancements are necessary for applying these approaches to the SBDD task. Notably, existing frameworks discretize categorical variables representing atom types into continuous representations, which is inherently unnatural given the data's discrete characteristics.

In summary, predicting the posterior for the final molecular structure $x_0$ and subsequently using it for calculating guidance gradients is more principled in the context of SBDD tasks. We propose that this principle is broadly applicable across generative modeling frameworks, including not only diffusion models but also BFN.

## J.3 Limitations of Current Evaluation Methods

In previous research on structure-based drug design (SBDD), evaluating the binding affinity between generated molecules and their target proteins has been a common practice to assess model performance. Most studies traditionally relied heavily on AutoDock Vina to measure binding affinity.

Although AutoDock Vina provides three distinct scoring metrics, these metrics inherently depend on the same underlying computational algorithm, potentially introducing bias due to reliance on a single scoring method. To enhance the generalizability and reliability of affinity assessments, incorporating multiple docking algorithms in the evaluation process is necessary. Accordingly, in Section 6.2, we present a detailed experimental setup employing several docking tools, including AutoDock Vina, to enable a more comprehensive and robust evaluation of generative model performance.

Furthermore, previous SBDD research has commonly utilized the Synthetic Accessibility (SA) score to evaluate the synthetic feasibility of generated molecules. The SA score quantitatively integrates chemical fragment contributions and structural complexity penalties into a single metric ranging between 0 and 1, with higher scores indicating greater synthetic accessibility. However, molecules possessing very high SA scores (e.g., greater than 0.9) frequently lack viable retrosynthetic pathways, making their actual synthesis infeasible. Regardless of a molecule's theoretical efficacy, its practical value is severely limited without an achievable synthetic route. Therefore, rigorously assessing realistic synthetic accessibility is critical, although research addressing this aspect has been relatively limited. Recognizing the importance of this issue, we introduce the AiZynthFinder benchmark, an evaluation method for retrosynthetic analysis based on practically available chemical building blocks.

From the viewpoint of practical drug development, selectivity is equally important as binding affinity for identifying promising drug candidates. Selectivity refers to the ability of a candidate molecule to specifically bind to its intended target protein without significant interactions with off-target proteins. Molecules lacking sufficient selectivity may interact with unintended proteins, potentially causing side effects or adverse reactions, thereby reducing or negating the desired pharmacological effects. Recently, selectivity has received increased attention in the field of 3D molecular generation, and several diffusion-based guidance strategies have been proposed to address this requirement.

However, existing selectivity-focused strategies typically require prior training of classifiers that distinguish between positive (binding) and negative (non-binding) protein-ligand pairs. Furthermore, the CrossDocked2020 dataset, commonly used in docking studies, was not originally constructed for selectivity evaluations. Thus, leveraging this dataset for selectivity assessments necessitates extensive additional docking computations. Moreover, the absence of clear criteria for identifying true binding molecules and the significantly greater number of false binding molecules relative to true binding molecules pose substantial challenges for obtaining generalizable guidance signals. Consequently, deriving selectivity metrics based solely on the CrossDocked2020 dataset inherently risks bias due to these intrinsic dataset limitations. Most critically, the CrossDocked dataset may not adequately reflect biologically meaningful selectivity, limiting its practical utility for reliable selectivity assessment. Therefore, establishing rigorous, standardized benchmark datasets capable of objectively evaluating selectivity is essential. Additionally, there is an urgent need to develop novel, efficient controllable generation strategies capable of effectively ensuring molecular selectivity.

# K   Justification for Applying Tweedie's Formula

**Justification for Applying Tweedie's Formula in Continuous Variables**   In our framework, the sender distribution for continuous variables is explicitly modeled as a Gaussian:

$$p_{\mathtt{S}}(y \mid x; \alpha) = \mathcal{N}(y \mid x, \alpha^{-1}I)$$

Given a prior over $x$ (possibly Gaussian), the Bayesian update for the parameter $\theta$ given a noisy observation $y$ from the sender is:

$$p(x \mid y) \propto p_{\mathtt{S}}(y \mid x; \alpha) \cdot p(x)$$

If we assume the prior $p(x) = \mathcal{N}(x \mid \mu_0, \Sigma_0)$, then the posterior $p(x \mid y)$ is also Gaussian, due to conjugacy:

$$p(x \mid y) = \mathcal{N}\left(x \mid \mu_{\text{post}}, \Sigma_{\text{post}}\right)$$

where

$$\Sigma_{\text{post}}^{-1} = \Sigma_0^{-1} + \alpha I, \quad \mu_{\text{post}} = \Sigma_{\text{post}}\left(\Sigma_0^{-1}\mu_0 + \alpha y\right)$$

Thus, the Bayesian update function for the mean parameter $\theta$ becomes a linear function of the previous mean and the new observation $y$:

$$\theta_{i+1} = \frac{\rho_{i-1}}{\rho_i}\theta_i + \frac{\alpha_i}{\rho_i}y_i$$

with $\rho_i = \rho_{i-1} + \alpha_i$, which matches the classic Kalman filter update for Gaussian models.

**Conclusion:**  Since the posterior is exactly Gaussian, Tweedie's formula

$$\mathbb{E}[x \mid y] = y + \alpha^{-1}\nabla_y \log p(y)$$

is strictly valid, and our use of Tweedie's formula to obtain a gradient-based update is mathematically justified.

**Justification for Applying Tweedie's Formula in Categorical Variables**   For categorical variables, the latent class indicator $e_x$ is originally a one-hot vector, i.e., $e_x \in \{0,1\}^K$ with $\sum_{k=1}^{K} e_x^{(k)} = 1$. However, in our framework, we reparameterize $e_x$ to a continuous relaxation (e.g., using the Gumbel-softmax trick), and the sender distribution is modeled as a multivariate Gaussian over this relaxed variable:

$$p_{\mathsf{S}}(y \mid e_x; \alpha) = \mathcal{N}\left(y \mid \alpha(Ke_x - \mathbf{1}), \alpha K I\right)$$

where $y$ is a continuous vector and $e_x$ is now allowed to be a point in the probability simplex.

Given this sender structure, the "pseudo-posterior" $p(e_x \mid y)$ (formally, the conditional distribution in the continuous relaxation) is also a multivariate Gaussian, due to the properties of conjugacy between Gaussians.

Thus, the Bayesian update for the categorical parameter $\theta$ (in the reparameterized space) can be written as a function of $y$ and the previous parameter:

$$e_{x,i+1} = \frac{\rho_{i-1}}{\rho_i}e_{x,i} + \frac{\alpha_i}{\rho_i}y_i$$

with $\rho_i = \rho_{i-1} + \alpha_i$, which is the direct analogue of the update for the continuous case.

**Applicability of Tweedie's Formula:**   Since both $y$ and $e_x$ are continuous and (conditionally) Gaussian under the sender, Tweedie's formula is applicable:

$$\mathbb{E}[e_x \mid y] = y + (\alpha K)^{-1}\nabla_y \log p(y)$$

and, by chain rule,

$$\nabla_{e_x} \log p(e_x) = J_{e_x \to y}^{\top}\nabla_y \log p(y)$$

where $J_{e_x \to y}$ is the Jacobian matrix of the sender's mapping.

**Conclusion:**   By reparameterizing the categorical variable into a continuous, sender-Gaussianized latent variable, all required conditions for Tweedie's formula (Gaussianity and differentiability) are satisfied in the update step. This mathematically justifies the use of Tweedie's formula for gradient-based guidance in categorical settings, just as in the continuous case.

