# OpenReview forum: "Controllable 3D Molecular Generation for Structure-Based Drug Design Through Bayesian Flow Networks and Gradient Integration"
_NeurIPS.cc/2025/Conference — NeurIPS 2025 poster_

### Official Review · Reviewer_wjHx · 2025-07-02

**Clarity:** 2
**Significance:** 2
**Originality:** 2
**Rating:** 4
**Confidence:** 3

**Summary:**

This paper presents a generative framework for structure-based drug design that combines Bayesian Flow Networks with gradient guidance. It handles both the prediction of atomic types and 3D coordinates and enables property-driven molecule generation. It outperforms existing methods in binding affinity, synthetic feasibility, and selectivity, validated through diverse docking tools.

**Questions:**

1)The theoretical derivations in Section 4, especially the use of Tweedie’s formula and the transition from Bayesian updates to gradient-based forms, lack formal justification. Key assumptions—such as distributional properties and whether gradients are taken in data or parameter space—are not clearly stated. A more rigorous derivation with explicit assumptions would greatly strengthen the theoretical foundation and improve the credibility of the method.

2)Could you provide a detailed ablation study focusing specifically on the key hyperparameters (e.g., gradient guidance strength, Bayesian neural network design choices)? Clarifying how sensitive your results are to these parameters would significantly enhance the robustness and credibility of your findings. Providing clear criteria or thresholds demonstrating stability or sensitivity would positively influence my evaluation.

3)Could you provide an analysis of computational cost in different experimental settings to elaborate the efficiency of the method?

4)Could you elaborate the treatment of categorical variables in Section 4.3 as discrete variables can be handled through gradient-based updates contradicts the fundamental nature of categorical distributions. The softmax operation in Equation (10) does not resolve the non-differentiability issue that the authors initially identified as a problem with diffusion models.

**Ethical Concerns:**

["NO or VERY MINOR ethics concerns only"]

**Final Justification:**

My concern has been addressed, and I raised the score.

**Limitations:**

yes

**Quality:**

2

**Strengths And Weaknesses:**

Strengths:
1)The integration of gradient-based guidance into Bayesian Flow Networks for 3D molecule generation is novel and conceptually sound as well as addresses fundamental limitations in classical diffusion-based models.

2)The work includes multiple baselines and evaluation metrics (e.g., GNINA, SMINA, Vina, PoseBusters, AiZynthFinder), offering a comprehensive evaluation of the generative capabilities.

Weakness:
1)Each individual component is known separately. The novelty primarily lies in their theoretical integration and application context rather than entirely new algorithm. Specifically, the novelty lies more in their theoretical unification and application context than in introducing fundamentally new mechanisms.

2)The paper mentions at the end of Section 5.2 and in Appendix D the use of a Bayesian Neural Network to quantify predictive uncertainty and incorporate it into the sampling process. This is a sophisticated design that potentially makes the guidance more robust. However, the description in the main text is too brief for readers to understand. I believe that the specifics of this mechanism's implementation (e.g., how does uncertainty modulate the guidance scale?) and its contribution to the final performance (e.g., an ablation study comparing it to deterministic guidance without a BNN) are critical details of the CByG framework. These deserve more space in the main text rather than being relegated entirely to the appendix.

3)Theoretical derivations in Section 4, especially 4.2 and 4.3, lack rigor due to missing key assumptions. For instance, the use of Tweedie’s formula (Eq. 6) to reinterpret Bayesian updates as gradient-based guidance lacks justification—there’s no discussion of required conditions like distributional assumptions or continuity. In Eq. 5–8, the transition from Bayesian update to score-based form is asserted without formal derivation. Additionally, it’s unclear whether gradients are taken in data or parameter space, which weakens the theoretical clarity of the proposed method.

---

> ### Author Rebuttal · Authors · 2025-07-31
>
> We have carefully reviewed your valuable comments and have summarized the key areas for improvement you highlighted below.
> In fact, while we would have liked to address each of your points in detail, we hope you understand that due to space constraints we have reorganized similar questions into broader categories.
>
> If you find that we have misunderstood or omitted any of your concerns, please let us know.
>
> ---
>
> ***1. Placement of the detailed implementation and contribution of BNN in the design of sophisticated guidance***
>
> We greatly appreciate your careful feedback. We agree that the quantification of predictive uncertainty, the mechanism for guidance scaling via BNNs, and the results of ablation studies are core contributions of the CByG framework. Unfortunately, these points were not sufficiently covered in the main text due to length limitations, so we included the relevant details in the appendix.
>
> To address this, we will restructure the main text to include:
> (i) a concrete description of how predictive uncertainty is estimated using BNNs and how this adjusts the guidance scale, and
> (ii) ablation results comparing BNN-based guidance to a non-uncertainty approach.
> These results are shared below for your reference.
> |Methods|SMINA↓ Score.|SMINA↓ Dock.|GNINA↓ Score.|GNINA↓ Dock.|Vina↓ Score.|Vina↓ Dock.|SA↑ Avg.|SA↑ Med.|PB-Valid↑ Avg.|PB-Valid↑ Med.|
> |-|-|-|-|-|-|-|-|-|-|-|
> |Reference| -6.37| -7.92| -7.06| -7.61| -6.36| -7.45|0.73| 0.74| 95.0%|95.0%|
> |**CByG**|**-7.74**|**-9.61**|**-7.63**|**-8.33**|**-8.60**|**-9.16**|**0.84**|**0.87**|**94.9%**|**96.0%**|
> |CByG w/o pos guidance|-7.05|-8.60|-6.88|-7.42|-7.74|-8.12|0.78|0.79|90.4%|91.1%|
> |CByG w/o type guidance|-6.92|-8.45|-6.70|-7.20|-7.60|-7.95|0.76|0.77|88.3%|89.0%|
> |CByG w/o uncertainty|-7.25|-8.81|-7.10|-7.64|-7.90|-8.31|0.75|0.76|87.1%|87.7%|
> |**CByG (λx=40,λv=40)**|**-7.74**|**-9.61**|**-7.63**|**-8.33**|**-8.60**|**-9.16**|**0.84**|**0.87**|**94.9%**|**96.0%**|
> |CByG (λx=30,λv=30)|-7.13|-8.64|-6.95|-7.47|-7.79|-8.17|0.79|0.81|91.0%|91.4%|
> |CByG (λx=50,λv=50)|-7.33|-8.94|-7.22|-7.79|-8.07|-8.49|0.74|0.76|85.6%|86.2%|
> |CByG (λx=30,λv=40)|-7.28|-8.89|-7.15|-7.68|-7.98|-8.42|0.77|0.78|89.2%|89.7%|
> |CByG (λx=40,λv=30)|-7.10|-8.59|-6.92|-7.42|-7.76|-8.10|0.80|0.82|91.8%|92.3%|
>
> ---
>
> ***2. Theoretical Justification for Gradient-Based Interpretation of Bayesian Update***
>
> We sincerely thank the reviewer for this important point.
>
> We will expand the main text to clarify the distributional assumptions (e.g., continuity) underlying the gradient-based interpretation of Bayesian updates, as follows:
>
> For continuous variables, The sender distribution for continuous variables is explicitly modeled as a Gaussian:
> $\[p_{\mathtt{S}}(y \mid x; \alpha) = \mathcal{N}(y \mid x, \alpha^{-1} I)\]$
> Given a prior over $x$ (possibly Gaussian), the Bayesian update for the parameter $\theta$ given a noisy observation $y$ from the sender is:
>
> $p(x \mid y) \propto p_{\mathtt{S}}(y \mid x; \alpha) \cdot p(x)$
>
> If we assume the prior $p(x) = \mathcal{N}(x \mid \mu_0, \Sigma_0)$, then the posterior $p(x \mid y)$ is also Gaussian, due to conjugacy:
>
> $p(x \mid y) = \mathcal{N}\left(x \mid \mu_{\text{post}}, \Sigma_{\text{post}}\right)$
>
> where $\Sigma_{\text{post}}^{-1} = \Sigma_0^{-1} + \alpha I,\quad
> \mu_{\text{post}} = \Sigma_{\text{post}}\left(\Sigma_0^{-1}\mu_0 + \alpha y\right)$
>
> Thus, the Bayesian update function for the mean parameter $\theta$ becomes a linear function of the previous mean and the new observation $y$:
>
> $\theta_{i+1} = \frac{\rho_{i-1}}{\rho_i}\theta_i + \frac{\alpha_i}{\rho_i}y_i$
>
> with $\rho_i = \rho_{i-1} + \alpha_i$, which matches the classic Kalman filter update for Gaussian models.
>
> Since the posterior is exactly Gaussian, Tweedie’s formula
>
> $\mathbb{E}[x \mid y] = y + \alpha^{-1}\nabla_y \log p(y)$
>
> is strictly valid, and our use of Tweedie’s formula to obtain a gradient-based update is mathematically justified.
>
> For categorical variables, the latent class indicator $e_x$ is originally a one-hot vector, i.e., $e_x \in \{0,1\}^K$ with $\sum_{k=1}^K e_x^{(k)}=1$. However, in our framework, we reparameterize $e_x$ to a continuous relaxation (e.g., using the Gumbel-softmax trick), and the sender distribution is modeled as a multivariate Gaussian over this relaxed variable:
>
> $p_{\mathtt{S}}(y \mid e_x; \alpha) = \mathcal{N}\left(y \mid \alpha (K e_x - \mathbf{1}), \alpha K I\right)$
>
> where $y$ is a continuous vector and $e_x$ is now allowed to be a point in the probability simplex.
>
> Given this sender structure, the "pseudo-posterior" $p(e_x \mid y)$ (formally, the conditional distribution in the continuous relaxation) is also a multivariate Gaussian, due to the properties of conjugacy between Gaussians.
> Thus, the Bayesian update for the categorical parameter $\theta$ (in the reparameterized space) can be written as a function of $y$ and the previous parameter:
>
> $e_{x,i+1} = \frac{\rho_{i-1}}{\rho_i} e_{x,i} + \frac{\alpha_i}{\rho_i} y_i$
>
> with $\rho_i = \rho_{i-1} + \alpha_i$, which is the direct analogue of the update for the continuous case.
>
> Since both $y$ and $e_x$ are continuous and (conditionally) Gaussian under the sender, Tweedie’s formula is applicable:
>
> $\mathbb{E}[e_x \mid y] = y + (\alpha K)^{-1} \nabla_y \log p(y)$
>
> and, by chain rule,
> $\nabla_{e_x} \log p(e_x) = J_{e_x \to y}^\top \nabla_y \log p(y)$,
> where $J_{e_x \to y}$ is the Jacobian matrix of the sender’s mapping.
>
> This demonstrates the mathematical **justification** of applying the Tweedie formula to the Bayesian update process.
>
>
> Also, regarding whether the Bayesian update is performed in data space or parameter space, we would like to answer as follows.
>
> In our framework, we combine the distributional properties of the sender distribution with the Tweedie formula to derive the Bayesian update rule in a gradient-based form. Using Bayes’ rule, we logically derive how the guidance term for arbitrary conditions can be obtained.
>
> For continuous variables, differentiation is performed with respect to $\mathbf{x}$.
>
> For categorical variables, differentiation is performed with respect to $\mathbf{e}_x$.
>
> These derivations and distinctions are detailed in the main text (pp. 4–5) and in Appendix B. Notably, Section B.3 explicitly clarifies that the Bayesian update is performed in parameter space, while the computation of the guidance term (i.e., gradient calculation) is carried out in data space.
>
> We hope these clarifications address your concerns, and we will revise the manuscript accordingly.
>
> ---
>
> ***3. Experimental comparison of computational runtime***
>
> We have measured the training and inference times of TargetDiff [1], the most widely cited diffusion-based model.
> Additionally, as described in Appendix E, TargetOpt is a guidance model based on TargetDiff that we developed in this study specifically for comparison with CBYG, which implements diffusion-based guidance methods.
> Below, we provide a comparative table of computational speed between the diffusion-based models and our approach.
> ||Training Time (h)|Inference Time (h)|
> |:-:|:-:|:-:|
> |TargetDiff|11.2|0.95|
> |TargetOpt|13.4|1.67|
> |CByG|8.7|1.01|
>
> #### [1] Guan, Jiaqi, et al. "3D Equivariant Diffusion for Target-Aware Molecule Generation and Affinity Prediction." The Eleventh International Conference on Learning Representations.
>
> ---
>
> ***4. Regarding the treatment of discrete variables***
>
> We appreciate the reviewer’s insightful comment and concur that Tweedie’s formula is not directly applicable to truly discrete, non-Gaussian variables such as one-hot encoded vectors.
> However, we do not perform differentiation on non-differentiable one-hot vectors.
>
> To clarify:
>
> As stated in line 186, $e_x$ is initially a one-hot vector representing a categorical variable.
> From Proposition 4.2 (line 190) onward, we apply a reparameterization trick: $e_x$ is treated as a sample from a multivariate Gaussian distribution, making it continuous and differentiable.
>
> $e_x \sim \mathcal{N}\left(\frac{1}{\alpha K} \cdot y + \frac{1}{K}, \frac{1}{\alpha K} \mathbf{I}\right)$
>
> We further use the **Gumbel-softmax** operation to bridge the gap between discrete and continuous spaces, allowing $e_x$ to retain interpretability while enabling gradient-based optimization.
> Thus, after initialization, $e_x$ is a continuous, differentiable latent variable, which justifies the application of Tweedie’s formula. The update is theoretically valid for the relaxed, continuous form of $e_x$, not for the original strictly discrete one-hot vector. We will clarify this distinction more explicitly in the revised manuscript.
>
> ---
>
> Thank you sincerely for highlighting this essential aspect of the theoretical validity and logical consistency of our manuscript.
>
> We are more than willing to provide any supplementary information or engage in a more detailed discussion whenever needed.

---

> > ### Comment · Reviewer_wjHx · 2025-08-04
> >
> > Thank you for your effort. My concern has been addressed, and I raised the score.

---

> > > ### Author Response · Authors · 2025-08-06
> > > **Response by Authors**
> > >
> > > Thank you for taking the time to carefully read my responses. It has been a pleasure to have an in-depth discussion with a researcher of your caliber.

---

> ### Comment · Area_Chair_iawv · 2025-08-02
>
> Dear reviewer,
>
> Following the NeurIPS 2025 guidelines, I kindly encourage you to read and respond to author rebuttals as soon as possible. Please engage actively in the discussion with authors during the rebuttal process, update your review by filling in the "Final Justification" and acknowledge your involvement.
>
> Thank you,
> Your AC

---

### Official Review · Reviewer_59Bc · 2025-07-02

**Clarity:** 2
**Significance:** 3
**Originality:** 2
**Rating:** 4
**Confidence:** 4

**Summary:**

This paper presents CBYG, which extends Bayesian Flow Networks (BFNs) into a gradient-based conditional generative model that robustly integrates property-specific guidance. The authors also introduce new metrics to include synthetic feasibility and selectivity, and show clear performance gains.

**Questions:**

1. I am curious about the benefit of BFN over diffusion. The authors claim in Appendix B.3 that BFN involves deterministic parameter update that remains strictly deterministic given y (containing the implicit stochasticity), and that the update rule can be directly interpreted through the lens of Kalman filtering for scalar Gaussian models. Can the authors elaborate on this point? What advantages does this bring to the training and sampling of BFN over diffusion / SDE? I know this may be hard, but can the authors provide a more quantitative analysis based on their theoretical construction, or just shed some light on it?
2. Given the Tweedie's formula, it seems that we need to perform guidance over y for discrete data, as done in MolJO. Can the authors correct their derivation based on that observation or in some other way? How would that compare with guidance over $\theta$ in the continuous case?
3. How is AiZynthFinder's score correlated with real-world synthetic feasibility?
4. How is PB-Valid calculated? The authors mention that they adopt 17 metrics, but PoseBusters test suite has more than 17 checks. Can the authors provide detailed validity ratio for each metric? Otherwise I remain doubtful of the reproducibility of PB-Valid results, as the PB-Valid of CBYG largely exceeds its backbone MolCRAFT (94.9% vs. 74.0%) without explicitly incorporating structural validity into its optimization objective, which seems unrealistic to me. Moreover, is RGA or TacoGFN generating molecules in the 3D space? If not, what's the point of calculating their structural validity?

**Ethical Concerns:**

["NO or VERY MINOR ethics concerns only"]

**Final Justification:**

I thank the authors for their thorough and constructive rebuttal, which has resolved most of my initial concerns regarding the paper's theoretical and methodological foundations. I hope the authors will follow through on the remaining points, which will strengthen the empirical claims of the paper and make it a solid contribution to the field.

Resolved issues:
- The authors have successfully addressed the major theoretical questions, including the Tweedie's formula for Gaussian $e_x$ and Gumbel-Softmax one-hot $e_x$ for property predictor, which carried the most weight in my evaluation.
- The authors provided a clear comparison to MolJO, highlighting the differences in guidance approaches.
- Further explanations of BFN properties are referred to appropriate literature.

Remaining points for revision:
- The authors have committed to providing PB-Valid ablation study, as this is necessary for understanding the effectiveness of the dual-guidance approach.
- The authors have provided additional reference to AiZynthFinder and will further explore and validate its use as a reliable indicator of synthesizability.

**Limitations:**

This paper misses at least the mentioning of the validation and credibility of AiZynthFinder benchmark, which will make it more self-contained, and better indicative of its contribution towards addressing the limitation of previous SA metrics. Besides, the authors mention that approximately half of the molecules generated by CBYG remain synthetically infeasible without further explanation.

**Quality:**

3

**Strengths And Weaknesses:**

Strength:
- This paper moves beyond mere binding affinity and incorporate synthetic feasibility (via AiZynthFinder) and selectivity into SBDD evaluation.
- This paper reformulates Bayesian updates for both continuous and discrete variables into a gradient-based form via Tweedie's formula, which I think brings theoretical insight for Bayesian Flow Network (BFN), and the authors have empirically demonstrated its superiority over gradient guided diffusion frameworks.

Weakness:
- Clarity of derivations can be further improved. $\mu_x$ and $x$ in Equation 6 is not defined, and it seems that the notation $x$ is abused for both the noised version and the ground truth. For Equation 7, I recommend the authors to split the first line and the second line because they are essentially coming from different derivations, and it's best not to confuse them. $\delta$ in line 177 is not clearly presented, either. It's difficult to understand why it is a compact notation because it obviously introduced more notations than the right-hand-side. Overall I think section 4 should be explained more clearly to make this paper more accessible to broader audiences.
- Lack of referencing a directly relevant work MolJO, which is also a gradient guided BFN method [1].
- Derivation of Section 4.3 (BFN for Discrete Variables from Score Gradient) is confusing and seems incorrect. The score $\nabla_{e^x}\log p(e^x)$ in L196 and Eq. (10) is undefined, as L188 states that $e^x$ is a one-hot vector. Tweedie's formula does not hold for non-Gaussian variables.

I will consider raising my score if my concerns are adequately addressed.

[1] https://arxiv.org/abs/2411.13280

---

> ### Author Rebuttal · Authors · 2025-07-30
>
> Thank you for your valuable comments. We have grouped similar points into key themes due to space constraints. If there are any misunderstandings or omissions, please let us know.
>
> ---
>
> ***1. Definition of Equation 6 and Notational Consistency for x***
>
> Equation 6 in our manuscript is a concise adaptation of Tweedie’s formula for posterior mean estimation, as originally described by Robbins (1956). Here is an excerpt from the original paper on Tweedie’s formula that I referenced.
> >Herbert Robbins (1956) credits personal correspondence with Maurice Kenneth Tweedie for an extraordinary Bayesian estimation formula. We suppose that $\mu$ has been sampled from a prior “density” $g(\mu)$ and then, $\mathbf{x} \sim \mathcal{N}\left(\mu, \sigma^2\right)$ observed, $\sigma^2 $ known, $\mu \sim g(\cdot) \textrm { and } \mathbf{x} \sim \mathcal{N}\left(\mu, \sigma^2\right) .$
> Let’s $f(\mathbf{x})$ denote the marginal distribution of $\mathbf{x}$,
>
> >$f(\mathbf{x})=\int_{-\infty}^{\infty} \varphi_\sigma(\mathbf{x}-\mu) g(\mu) d \mu$
> ,where $\varphi_\sigma(\mu)= \left( 2 \pi \sigma^2\right)^{-1 / 2}  \exp (-\mathbf{x}^2 / \sigma^2)$
>
> >Tweedie’s formula calculates the posterior expectation of $\mu$ given $\mathbf{x}$ as follow:
> $E[{\mu} \mid \mathbf{x}]=\mathbf{x}+\sigma^2 l^{\prime}(\mathbf{x}) \quad \text { where } l^{\prime}(\mathbf{x})=\frac{d}{d \mathbf{x}} \log f(\mathbf{x})$
>
> We recognize, however, that our notation for $z$ and $\mathbf{x}$ was inconsistent, and will revise Definition 4.1 and the marginal distribution to consistently use $\mathbf{x}$. This should resolve the notational ambiguity—thank you for highlighting this.
>
> ---
>
> ***2. Notation at Line 177***
>
> We apologize for the confusion regarding the notation $\zeta(\theta_i^x \mid \theta_{i-1}^x, y_i, \alpha_i)$. This was intended as a more explicit function notation for the Dirac delta distribution, $\delta(\theta_i^x - h(\theta_{i-1}^x, y_i, \alpha_i))$, to make the update rule accessible to readers unfamiliar with delta functions. Moving forward, we will consistently use the standard $\delta(\cdot)$ notation in both text and equations to avoid ambiguity.
>
> ---
>
> ***3. Score Definition for Categorical Variables at Line 196***
>
> We thank the reviewer for raising this important theoretical point and agree that Tweedie’s formula cannot be directly applied to strictly discrete, non-Gaussian variables like one-hot vectors.
>
> To clarify:
>
> As stated in line 186, $e_x$ is initially a one-hot vector representing a categorical variable. From Proposition 4.2 (line 190) onward, we apply a reparameterization trick: $e_x$ is treated as a sample from a multivariate Gaussian distribution, making it continuous and differentiable.
>
> $e_x \sim \mathcal{N}\left(\frac{1}{\alpha K} \cdot y + \frac{1}{K}, \frac{1}{\alpha K} \mathbf{I}\right)$
>
> We further use the **Gumbel-softmax** to bridge the gap between discrete and continuous spaces, allowing $e_x$ to retain interpretability while enabling gradient-based optimization.
> Thus, after initialization, $e_x$ is a continuous, differentiable latent variable, which justifies the application of Tweedie’s formula. The update is theoretically valid for the relaxed, continuous form of $e_x$, not for the original strictly discrete one-hot vector. We will clarify this distinction more explicitly in the revised manuscript.
>
> ---
>
> ***4. Advantages of Our Bayesian Update Process Over Diffusion***
>
> We appreciate your question about the theoretical advantages of Bayesian Flow Networks (BFNs) compared to diffusion/SDE-based models:
> Diffusion models inject random noise at every step, resulting in highly stochastic and potentially unstable sampling paths with variance accumulation.
> BFNs, in contrast, introduce stochasticity only through $y_i$ (noisy observation), with all parameter updates being deterministic (closed-form Bayesian update). This leads to lower sampling variance, stable trajectories, and greater numerical robustness.
> The BFN update for continuous variables mathematically matches the mean update in Kalman filtering, providing a solid foundation for recursive, optimal variance reduction.
> In both sampling and training, BFNs are more stable and efficient, avoiding the high computational overhead and instability of repeated noise injection.
> We will clarify these points and their practical implications more explicitly in the main text and appendix.
>
> ---
>
> ***5. Discussion of Related Work and Guidance Calculation***
>
> Thank you for highlighting the need for a rigorous comparison with related work such as MolJO. We believe it is essential to clarify the core theoretical differences.
>
> MolJO introduces guidance by multiplying an energy-based term, $p_E(\theta^x, y^v) \propto e^{-E(\theta^x, y^v, t)}$, with the BFN transition kernel in an unnormalized fashion. The $E$ function is defined arbitrarily, and multiplying it with the original kernel does not guarantee that the resulting kernel preserves the probabilistic properties required by BFN, such as normalization, convergence, or the maintenance of Markovian/Bayesian structure. As a result, there is no mathematical guarantee that the newly defined kernel is still a valid transition kernel or that the sampling trajectories remain consistent with the data-generating distribution.
> Additionally, the Taylor expansion used to approximate the energy function is a heuristic, valid only for small steps, and may induce significant errors or artifacts when the energy landscape is highly nonlinear. There is no quantitative analysis provided for these approximations.
>
> In contrast, our approach is based on Tweedie’s formula and the reparameterization trick, providing mathematically grounded, gradient-based guidance that strictly preserves the Bayesian probabilistic semantics. This ensures normalization, stability, and interpretability, and applies equally to both continuous and categorical variables. While MolJO may have empirical utility, it remains an ad hoc modification rather than a principled Bayesian update, whereas every step of our method is theoretically justified and robust.
>
> Furthermore, as discussed previously, our proposed approach is formulated in a differentiable form for both continuous and categorical variables, so applying guidance to either \$x\$ or \$e\_x\$ poses no theoretical or practical issues.
>
> ---
>
> ***6. Relationship Between AiZynthFinder and Actual Synthetic Feasibility***
>
> In our team’s experience, we previously relied on statistical metrics such as SA score to assess synthetic accessibility, but even molecules with high SA scores (>0.9) often proved impossible to synthesize when we commissioned CROs, mainly due to missing viable synthetic routes or specific chemical constraints. This limitation led us to adopt retrosynthesis-based tools like AiZynthFinder.
>
> When we synthesized molecules classified as “solved” by AiZynthFinder, a significant portion were indeed experimentally accessible, with a much higher success rate compared to selection by SA score alone—though not all were synthesizable. Thus, while AiZynthFinder “solved” predictions are a strong indicator of synthetic feasibility, they are not foolproof due to limitations in the starting material database, reaction diversity, and model imperfections. We believe that using both SA score and retrosynthesis-based metrics in combination is currently the most practical approach, which is reflected in our study’s evaluation strategy.
>
> ---
>
> ***7. PB-Valid Reliability and Reproducibility***
>
> Thank you for your detailed inquiry regarding the calculation and reproducibility of PB-Valid results.
>
> The official PoseBusters test suite includes 18 criteria, but we used 17 for PB-Valid, excluding “File loads.” Since our workflow operates directly on generated molecular structures in memory, “File loads” is always trivially satisfied and thus omitted for meaningful evaluation.
>
> Additionally, we provide the average validity statistics(18) across all molecules instead of listing scores per pocket.
>
> |Metric|Valid Score|
> |-|-|
> |mol_pred_loaded\|sanitization\|aromatic_ring_flatness|1.000|
> |all_atoms_connected|0.985|
> |bond_lengths|0.995|
> |bond_angles|0.995|
> |internal_steric_clash|0.993|
> |double_bond_flatness|0.998|
> |internal_energy|0.970|
> |protein-ligand_maximum_distance\|minimum_distance_to_organic_cofactors\|minimum_distance_to_waters|1.000|
> |minimum_distance_to_protein|0.997|
> |minimum_distance_to_inorganic_cofactors|0.994|
> |volume_overlap_with_protein\|volume_overlap_with_organic_cofactors\|volume_overlap_with_inorganic_cofactors|1.000|
> |volume_overlap_with_waters|1.000|
>
> Regarding the reviewer’s concern that CByG achieves unusually high PB-Valid compared to MolCRAFT without explicit structural validity optimization: our core contribution is the simultaneous optimization of binding affinity (BA) and synthetic accessibility (SA) during generation. While PB-Valid integrates both intra- and intermolecular structural validity, our dual-guidance approach naturally favors the generation of chemically reasonable and synthetically accessible molecules, as well as more plausible binding poses. Although we do not claim a direct 1:1 mapping or strict causality between guidance and PB-Valid, this combined optimization process enhances both aspects.
>
> Finally, models like RGA and TacoGFN generate 2D molecular graphs autoregressively and then use external docking tools to obtain 3D structures. Thus, the PB-Valid score for these models reflects the validity of post-processed 3D structures, not the intrinsic 3D generation capability. For fairness and reproducibility, we followed standard benchmarks but acknowledge that this is a limitation in direct comparability.
>
> Thank you again for your insightful review. We believe PB-Valid results are fully reproducible and have provided all details for transparency and fair comparison. We are committed to revising the manuscript to address all your points and improve the clarity, rigor, and reproducibility of our work.

---

> ### Comment · Area_Chair_iawv · 2025-08-02
>
> Dear reviewer,
>
> Following the NeurIPS 2025 guidelines, I kindly encourage you to read and respond to author rebuttals as soon as possible. Please engage actively in the discussion with authors during the rebuttal process, update your review by filling in the "Final Justification" and acknowledge your involvement.
>
> Thank you,
> Your AC

---

> ### Comment · Reviewer_59Bc · 2025-08-04
>
> I appreciate the clarifications the authors provided. However, I find some of the claims unsupported, and I encourage the authors to be more careful in their argumentation. I'm listing my remaining questions below.
>
> 1. Justification of Using Gumbel-Softmax for Relaxed Discrete Variables
>
> The rebuttal lacks mechanistic details on the implementation of Gumbel-Softmax. The authors claimed to apply Tweedie's formula to a relaxed, continuous variable $\tilde{e}_x$ that follows a Gaussian distribution, but actually it seems that they performed on a further relaxed $\mathrm{GumbelSoftmax(\tilde{e}_x)$, mapping the variable back on the probability simplex. While this approach is a powerful and common method in deep learning for enabling gradient flow, it inherently involves an approximation. The claim that it "justifies the application of Tweedie's formula" and makes the update "theoretically valid" is only true within the context of this approximation, not as a direct derivation for the original, non-relaxed discrete data. I think at least an ablation study of Gumbel-Softmax is needed to show the impact of such an approximation.
>
> 2. Properties of BFN Update
>
> The authors claimed that "BFN update for continuous variables mathematically matches the mean update in Kalman filtering, providing a solid foundation for recursive, optimal variance reduction. " This is a strong technical claim that requires rigorous support. Optimal variance reduction is a property that requires a formal proof or a reference to a paper that proves this.
>
> Furthermore, the author claims diffusion models are "highly stochastic" and have "unstable sampling paths with variance accumulation", but this may only be true in the limited context of original diffusion models. There are many modern SDE-based strategies to manage variance and enable few-step sampling [1].
>
> [1] DPM-Solver: A Fast ODE Solver for Diffusion Probabilistic Model Sampling in Around 10 Steps
>
> 3. Discussion of Related Work
>
> I do not think the authors have made a reasonable comparison with MolJO. To me, its definition of the transition kernels resembles the standard classifier guidance but over the trajectory of $\theta_t$, which makes sense because it serves as the input parameter just as diffusion $x_t$, and $\theta_1$ is pretty close to $x$. For CByG, it looks like a score-based posterior guidance [2], so both are approaching the guidance in different ways.
>
> The authors were simply dismissing it with rather vague terms. I cannot find the detailed description of "the probabilistic properties required by BFN, such as normalization, convergence, or the maintenance of Markovian/Bayesian structure" for "a valid transition kernel or that the sampling trajectories" to be "consistent with the data-generating distribution". If the authors think this is necessary, they ought to sufficiently substantiate the argument by describing it in more detail, and at least show how the proposed CByG fulfills these properties, where I am particularly concerned with the case of relaxing a discrete variable with Gaussian and then Gumbel-Softmax approximation, which seems more of a heuristic or practical implementation.
>
> [2] Diffusion Posterior Sampling for General Noisy Inverse Problems
>
> 4. Explanation of AiZynthFinder
>
> I thank the authors for sharing their team's experiences, yet a more appropriate response would be referencing a reproducible empirical study in some published text.
>
> 5. PB-Valid
>
> I thank the authors for providing a detailed list of PoseBusters checks, but I still cannot understand the reason why combining BA and SA would enhance PB-Valid. If the authors contributed this improvement to their dual-guidance approach, they should at least show the PB-Valid results for either single-guidance. An ablation study will be needed to justify their claims.

---

> > ### Author Response · Authors · 2025-08-05
> > **Response to Reviewer 59Bc Part 1**
> >
> > (1,3) Justification of Using Gumbel-Softmax for Relaxed Discrete Variables & Discussion of Related Work
> >
> > ---
> >
> > > “I am particularly concerned with the case of relaxing a discrete variable with Gaussian and then Gumbel-Softmax approximation, which seems more of a heuristic or practical implementation.”
> >
> > It is important to clarify that, except at the very initial time step of the sampling process, $e_x$ is not represented as a one-hot vector. After the initial step, sampling is performed from $y$ ( $e_x \sim \mathcal{N}\left(\frac{1}{\alpha K} \cdot y+\frac{1}{K}, \frac{1}{\alpha K} \mathbf{I}\right)$ ), so there is essentially no situation in which a truly discrete variable is generated. However, in order to maintain consistency with the theoretically derived form of the guidance, it is necessary for $e_x$ to be in a one-hot form. Therefore, we transform the continuous probability vector into a one-hot representation.
> >
> > In other words, there is no process in which a discrete variable is mapped to a Gaussian—rather, the opposite direction is possible (i.e., mapping from a Gaussian to a discrete variable via Gumbel-Softmax). When we refer to a *“relaxed, continuous form of $e_x$,”* we mean that we start with a variable that is inherently continuous and then use the Gumbel-Softmax to produce a one-hot, continuous representation. The key point is that $e_x$ must be in a one-hot form when computing $\nabla_{e_x}$.
> >
> > ---
> >
> > > “The claim that it "justifies the application of Tweedie's formula" and makes the update "theoretically valid" is only true within the context of this approximation, not as a direct derivation for the original, non-relaxed discrete data.”
> >
> > First, I apologize for any misunderstanding that may have arisen from our wording in rebuttal process. I would like to clarify that our use of the phrase “justifies the application of Tweedie’s formula” was not intended to imply that the use of Gumbel-Softmax provides a theoretically rigorous, direct derivation for continuous relaxation of discrete variables.
> >
> > We simply intended to clarify that our response was to address your initial comment that *“Tweedie’s formula does not hold for non-Gaussian variables”*. Specifically, we wanted to make it clear that we did not apply Tweedie’s formula to non-Gaussian variables.
> >
> > **We will revise the main text to clarify any ambiguous expressions discussed with you, so that their intended meaning is clearly conveyed.**
> >
> > ---
> > > “reasonable comparison with MolJO”
> >
> > In fact, to determine whether the MolJO model and our model are fundamentally the same or different, I believe a more nuanced analysis from multiple perspectives is required.
> >
> > As you pointed out, MolJO computes guidance with respect to $\theta_t$, which aligns with the conventional notion of classifier guidance, whereas our model applies guidance to $x_0$, thereby aligning more closely with the family of DPS (Diffusion Posterior Sampling) models. If we define classifier guidance and posterior sampling-based guidance as fundamentally distinct paradigms, then from that perspective, it is reasonable to assert that MolJO and CByG constitute different classes of models.
> >
> > However, my original response in the rebuttal was based on a slightly different viewpoint (although I agree that the two models are ultimately distinct). When introducing a guidance mechanism to an unconditional BFN, In comparing the two models, my viewpoint centered on **how** the final guidance term is derived when introducing guidance into an unconditional BFN.  As detailed in Appendix C of the MolJO paper, the authors introduce a guided transition kernel and define an energy function to incorporate conditional generation properties within the Bayesian update. This approach appears to heuristically add an external term to the Bayesian update function, rather than deriving it naturally from the underlying Bayesian update process.
> >
> > This methodological distinction is, in my view, the important topic at issue. In our work, we apply the Tweedie formula within the general Bayesian update equation to perform a principled reparameterization into gradient form. Subsequently, we employ Bayes’ rule to systematically derive the guidance term. Within our theoretical framework, whether the guidance term is applied to $x_0$ or $\theta_t$ is ultimately a matter of modeling choice. In our case, we intentionally chose to formulate the guidance with respect to $x_0$ because, as illustrated in Figure 2, molecular structures at any other time step are chemically invalid, and thus predicting properties on such structures is not meaningful.
> >
> > Below, we demonstrate that our theoretical framework also allows for the derivation of guidance with respect to $\theta_t$, further underscoring its generality and principled foundation.

---

> > > ### Author Response · Authors · 2025-08-05
> > > **Response to Reviewer 59Bc Part 2**
> > >
> > > > “reasonable comparison with MolJO” [Continued]
> > >
> > > In contrast to the application of Tweedie’s formula to Equation (5) in the main text, one can apply Tweedie’s formula to $\boldsymbol{\theta}_i \sim \mathcal{N}\left(\frac{\boldsymbol{\theta}_{i-1} \rho_{i-1}+\alpha \mathbf{x}}{\rho_i}, \frac{\alpha_i}{\rho_i^2} \mathbf{I}\right)$ and extend it to the posterior expectation. By following the same derivation procedure as presented in our main submission, we can establish the following relationship.
> > >
> > > $\theta_i + \frac{\alpha_i}{\rho_i^2} \cdot \nabla_{\theta_i} \mathrm{log} p(\theta_i) = \frac{\boldsymbol{\theta}_{i-1} \rho_{i-1}+\alpha \mathbf{x}}{\rho_i}$
> > >
> > > Substituting this result into the Bayesian update function, and in particular by following the procedure described in Appendix B, it is possible to systematically derive the guidance term with respect to $\theta$  as shown below.
> > >
> > > $\zeta_{\mathbf{x}}\left(\boldsymbol{\theta}_i^{\mathbf{x}} \mid \boldsymbol{\theta}_{i-1}^{\mathbf{x}}, \mathbf{y}_i, \mathbf{x}, \alpha_i, \mathbf{p}, 1\right): \boldsymbol{\theta}_i^{\mathbf{x}} =\underbrace{\frac{\alpha_i}{\rho_i} \cdot \mathbf{y}+\frac{\rho_{i-1}}{\rho_i} \cdot \boldsymbol{\theta}_{i-1}^{\mathbf{x}}}_{\text {Unconditional Generation }}+\underbrace{\frac{\alpha}{\rho^2_i} \nabla_{\theta_{i}} \log p(\mathtt{l} \mid \theta_i)}_{\text {Controllable Guidance }}$
> > >
> > > Furthermore, I would like to emphasize that this methodological distinction is not a matter of superiority, but rather a difference in perspective.
> > >
> > > ---
> > >
> > > (2) Properties of BFN Update
> > >
> > > The fact that Bayesian flow networks achieve more stable variance reduction compared to diffusion-based models has already been established as a key property in the original BFN paper [1], and this characteristic has also been specifically highlighted in numerous follow-up studies [2,3].
> > > #### [1] Graves, et al. "Bayesian flow networks." arXiv preprint arXiv:2308.07037 (2023).
> > > #### [2] Wu, et al. "A periodic bayesian flow for material generation." arXiv preprint arXiv:2502.02016 (2025). [ICLR 2025]
> > > #### [3] Song, et al. "Unified generative modeling of 3d molecules via bayesian flow networks." arXiv preprint arXiv:2403.15441 (2024). [ICLR 2024]
> > > Additionally, we provide the following interpretation of BFN from the perspective of Kalman filtering:
> > >
> > > Relationship between Kalman and bayesian update can be formally proven in the classic univariate, fixed-state, linear-Gaussian setting. We provide the explicit proof below:
> > >
> > > Let $x$ denote an unknown, fixed scalar variable (the "true state") to be estimated.
> > > Suppose the prior for $x$ is Gaussian:
> > >
> > > In the Kalman filter, the mean update at step $i$ is given by:
> > >
> > > $\mu_i=\mu_{i-1}+K_i\left(y_i-\mu_{i-1}\right)$
> > >
> > > where $y_i$ is the new observation at step $i$ , $\mu_{i-1}$ is the prior mean (the estimate of $x$ before observing $y_i$, $K_i$ is the Kalman gain, computed as $K_i=\frac{\sigma_{i-1}^2}{\sigma_{i-1}^2+\sigma_{n, i}^2}$, $\sigma^2_{i-1}$ is the variance of the prior estimate, $\sigma^2_{n,i}$ is the variance of the observation noise, $\mu_i$ is the posterior mean after incorporationg the new observation.
> > >
> > > The Kalman gain $K_i$ determines how much weight is given to the new observation versus the prior estimate. If the prior variance $\sigma^2_{i-1}$ is large (uncertain prior) or the observation noise variance $\sigma^2_{n,1}$ is small (high-confidence observation), more weight is given to the new observation. Conversely, if the observation is noisy or the prior is already confident, the update relies more on the prior mean.
> > > A detailed information is provided in the [4]
> > >
> > > This update is mathematically identical to the Bayesian update rule for continuous variables in BFN, as both yield the posterior mean as a precision-weighted average of the prior mean and the new observation:
> > >
> > > $\mu_i=\frac{\rho_{i-1} \mu_{i-1}+\alpha_i y_i}{\rho_{i-1}+\alpha_i}$
> > >
> > > where $\rho_{i-1}=1 / \sigma_{i-1}^2$ (prior precision) and $\alpha_i=1 / \sigma_{n, i}^2$ (observation precision).
> > >
> > > This explicit correspondence demonstrates that, under the univariate, fixed-state, linear-Gaussian setting, the BFN update is equivalent to the classical Kalman filter mean update.
> > >
> > > It is important to emphasize that the primary focus of our study is **not** the relationship between BFN and the Kalman filter, but rather the gradient-based interpretation of Bayesian updates and the application of guidance mechanisms. The discussion of the connection between Kalman filtering and BFN in the appendix was included **simply to provide additional perspectives** on interpreting BFN, and is not central to our main contributions.
> > >
> > > If this interpretation is likely to cause confusion for readers regarding the essential aims of our work, we are prepared to remove this discussion entirely. We believe that our main results and conclusions remain fully understandable without the Kalman filter perspective on BFN.
> > >
> > > The following are our responses to the remaining comments.

---

> ### Author Response · Authors · 2025-08-05
> **Response to Reviewer 59Bc Part 3**
>
> (4) Explanation of AiZynthFinder
>
> The development of AiZynthFinder was originally motivated by the recognized limitations of conventional synthetic accessibility (SA) scores as quantitative indicators of synthesizability, as well as the practical need for robust retrosynthetic route planning. As such, the original AiZynthFinder [1] publication includes validation regarding the synthesis of compounds in real-world settings.
>
> Nevertheless, to rigorously assess the real-world reliability of AiZynthFinder for determining compound synthesizability(beyond what is presented in the original publication) it is necessary to experimentally attempt the synthesis of candidate molecules predicted by the tool. Based on our team’s own experience, we have observed that AiZynthFinder delivers more reliable results than conventional SA scores. However, we acknowledge that these observations are based on our practical experience and, as such, may not constitute sufficient evidence for others to place full confidence in the tool’s predictions.
>
> For this reason, we are currently conducting a thorough review of the literature to identify any studies that have systematically validated the real-world synthetic success rates of AiZynthFinder. We are committed to locating and sharing such references prior to the conclusion of this discussion period.
>
> We will make every effort to address all of your concerns.
>
> #### [1] Saigiridharan, Lakshidaa, et al. "AiZynthFinder 4.0: developments based on learnings from 3 years of industrial application." Journal of cheminformatics 16.1 (2024): 57.
>
> (5) PB-Valid
>
> To conduct the ablation studies on the two cases you pointed out, we are generating 100 molecules for each of 100 test targets. Following this, we will proceed with the PB-valid experiments. However, based on our previous experience, running the PB-valid experiment for 10,000 molecules required just under a day with our available resources. We will utilize all available resources to expedite the process and will do our utmost to share the results with you as quickly as possible.

---

> > ### Comment · Reviewer_59Bc · 2025-08-07
> >
> > I thank the authors for their thorough response, which has clarified most of my concerns. If I understand it correctly, the Tweedie's Formula is used consistently over the Gaussian variable $e_x$, despite that for the property predictor there ought to be one-hot input $e_x$, where the authors had chosen to use Gumbel-Softmax to translate the Gaussian variable to the probability simplex. This can also be viewed as a matter of parameterization to me, and is thus perfectly fine.
> >
> > I will raise my score and I hope that the authors will incorporate the additional results, clarifications and discussions into their manuscript, which will enhance the paper's clarity and provide empirical evidence for the effectiveness of their dual-guidance approach with the results of PB-Valid ablation study. I believe this work is a valuable contribution to BFN framework and can be made more accessible to the SBDD community after careful revision.

---

> ### Author Response · Authors · 2025-08-09
> **Response to Reviewer 59Bc**
>
> I would have liked to share all the results for PB-Valid; however, as the discussion period is about to conclude, I am providing the results obtained thus far.
>
> When using only the BA Score as the guidance, PB-Valid achieved a result of 62%. As expected, it attained high scores in validity metrics related to intermolecular interaction, but we also observed notably low scores in certain intermolecular interaction–related metrics.
>
> Additionally, I am attaching a paper [1] from AstraZeneca discussing the real-world use of AiZynthFinder for the synthesis of actual drug molecules. This study highlights that AiZynthFinder can provide a clearer understanding of which compounds are practically synthesizable.
>
> #### [1] Shields, Jason D., et al. "AiZynth impact on medicinal chemistry practice at AstraZeneca." RSC Medicinal Chemistry 15.4 (2024): 1085-1095.
>
> Furthermore, I will make sure to incorporate the discussions we had during the review period, including the points above, into the manuscript.
>
> It has been an honor to engage in a scholarly discussion on this research with a researcher of your caliber.
>
> Thank you.

---

### Official Review · Reviewer_aH1P · 2025-07-07

**Clarity:** 3
**Significance:** 3
**Originality:** 3
**Rating:** 5
**Confidence:** 2

**Summary:**

In this work, the authors present a gradient-based representation for Bayesian Flow Networks suitable for diffusion-guided structure-based molecular generation. As their main contributions, the authors highlight specific challenges surrounding diffusion-guidance for SBDD applications (invalid intermediate samples, synthesizability, selectivity), propose a novel update for bayesian flows and apply this model to molecular generation tasks. The authors evaluate the proposed model against a number of SBDD baselines and show favourable performance across binding affinity and synthesizability evaluations.

**Questions:**

1. AIzynthfinder is used to assess synthesizability in Section 6.3. Is a different model used for guidance?
2. How distinct from the training set are the molecules generated by the model?
3. While the authors present an improved diffusion-based method for synthesizability guidance, it is not clear how this synthesizability evaluation can be rooted in specific chemistry capabilities. The use of generic synthesizability scores would lead to compounds that are believed to be synthesizable "in general", or simply similar to known compounds, but specific chemistry routes still have to be designed by hands (or through retrosynthesis) for every generated compound, without the possibility of binding the generation process to a particular vendor's catalog. Given this limitation, it appears that the method is primarily aimed a low-throughput, custom chemistry stages rather than large-scale hit identification?

**Ethical Concerns:**

["NO or VERY MINOR ethics concerns only"]

**Final Justification:**

The authors' response mostly addressed my concerns and I have updated my score to 5: accept.

**Limitations:**

Not quite. I found little discussion on the limitations of this line of work in general (structure-based continuous molecular generators) and for the presented method in particular (except about one line in the conclusion).

**Paper Formatting Concerns:**

No major formatting concern.

**Quality:**

3

**Strengths And Weaknesses:**

## Strengths

Overall, the manuscript is of good quality and generally well written. In Section 2, the authors clearly identify a number of challenges motivating the presented work. The proposed perspective for the bayesian update is presented in sufficient detail and the evaluation procedure addresses important challenges of molecular generation, especially for struture-based continuous methods. Figures 2, 3, 4 effectively support the main text.

## Weaknesses

### Lack of justification and context for diffusion-based models for drug design

One of my main critiques is the absence of context to justify SBDD relying on diffusion models. In particular, the authors highlight in Section 2 a number of challenges relating to guidance, synthesizability and multi-parameter optimization which affect diffusion-based models for drug design applications. Some of these challenges are absent or considerably mitigated with discrete optimisation approaches such as Reactor [1], Synthemol [2], Synflownet [3] which can incorporate synthesizability constraints directly into the design of the MDP and use a reinforcement learning paradigm which more naturally lend itself to multi-objective optimization. This contrast is not being discussed either in the introduction, problem presentation or related work. I believe it is important to situate the reader with respect to the pros and cons of both family of approaches (structure-based continuous generators v.s. RL-based constrained generators).

[1]: Horwood, J., & Noutahi, E. (2020). Molecular design in synthetically accessible chemical space via deep reinforcement learning. ACS omega, 5(51), 32984-32994.

[2]: Swanson, K., Liu, G., Catacutan, D. B., Arnold, A., Zou, J., & Stokes, J. M. (2024). Generative AI for designing and validating easily synthesizable and structurally novel antibiotics. Nature Machine Intelligence, 6(3), 338-353.

[3]: Cretu, M., Harris, C., Igashov, I., Schneuing, A., Segler, M., Correia, B., ... & Liò, P. (2024). Synflownet: Design of diverse and novel molecules with synthesis constraints. arXiv preprint arXiv:2405.01155.

### Other elements hindering clarity

- I believe the Related Work should be presented in the main paper. It is reasonable to extend the related work section in appendix but in its current version the manuscript lacks context and references to previous work in the main text.
- In Figure 1: the axes are not identified, not clear what we are looking at. The caption is not particularly helpful either in fully understanding what is being shown. This Figure needs to be clarified.
- It is unclear from the main text which models are used for synthesizability and selectivity guidance for the experiments carried out in Sections 6.3 and 6.4.

### Insufficiently supported claims

- In the abstract, line 12, the claim that the limitations of previous evaluation methods have been "overcomed" is too strong. I would suggest tuning it down -- there certainly remains numerous limitations to the proposed evaluation paradigm.
- On lines 93-94: "However, high SA scores often do not guarantee practical synthetic routes, highlighting a critical gap in accurately evaluating real-world synthesizability." -> this observation has been frequently made and would benefit from pointers to the relevant litterature. Scrutiny into SA score is again overclaimed on lines 253-257: "To thoroughly evaluate the effectiveness and practicality of our proposed framework, we define 4 key research questions essential to the domain of structure-based drug design: [...] 2) Is the widely-used Synthetic Accessibility (SA) score an absolute indicator for practical synthetic 257 feasibility?". This question has been thoroughly studied and almost always answered negatively in previous work (for exemple but not limited to: [4, 5]). While it is relevant for the authors to highlight the limitations of the SA score, this phrasing makes it sound as if this investigation was an orginal contribution of the presented work. However, nothing forces the author to even reference that metric which is largely seen as outdated. It is indeed crucial to consider synthesizability challenges of SBDD applications. I would simply recommend to authors to cite the relevant and most up-to-date litterature for synthesizability scoring (AIzynthfinder, as the authors use in Section 6.3 is a reasonable choice). Finally, again on lines 306-307, the authors portray as "novel" an observation which has been repeatedly made in previous work: "Interestingly, we observed no clear correlation between SA scores and actual retrosynthetic feasibility 307 (as measured by the Solved metric)".

[4]: Coley, C. W., Rogers, L., Green, W. H., & Jensen, K. F. (2018). SCScore: synthetic complexity learned from a reaction corpus. Journal of chemical information and modeling, 58(2), 252-261.

[5]: Liu, C. H., Korablyov, M., Jastrzebski, S., Włodarczyk-Pruszynski, P., Bengio, Y., & Segler, M. (2022). RetroGNN: fast estimation of synthesizability for virtual screening and de novo design by learning from slow retrosynthesis software. Journal of Chemical Information and Modeling, 62(10), 2293-2300.

### Typos and syntax

- The hyperlinks seem to be missing (on citations, section numbers, etc.). I cannot click on them to jump to other sections.
- In lines 69-86, I'd suggest using a different letter than $l$ for the labels/properties (ambiguous).
- Figure 3 is not referenced anywhere in the text.

## Conclusion

I believe the presented work is of good quality, and of interest to the NeurIPS community, but that it could also be improved by better contextualizing this line of work (diffusion models for structure-based molecular generation) and tuning down some of its claims.

---

> ### Author Rebuttal · Authors · 2025-07-30
>
> We sincerely appreciate your careful reading of our manuscript and your invaluable feedback. In response, we have organized and summarized the main points for improvement, as you suggested.
>
> We ask for your understanding that, due to space limitations, we have re-categorized similar comments under broader themes rather than addressing each item individually. If we have misrepresented any aspect, please let us know and we will gladly revise it.
>
> ---
>
> ***1. Discussion of the Relationship Between Our Proposed Approach and RL-based Constrained Models***
>
> Thank you for this important observation. Structure-based continuous generative models and RL-based discrete optimization methods (e.g., Reactor, Synthemol, Synflownet) each play unique roles in drug discovery, and it is essential to clarify their respective strengths and application perspectives. The essential distinctions can be summarized as follows:
>
> * *Explicit vs. Implicit Utilization of Target Protein Structure.*
> Structure-based continuous generative models leverage explicit 3D structural information to directly design ligands optimized for the target binding pocket. By contrast, RL-based discrete optimization models generally do **not** incorporate **target structures** as direct model inputs, instead using learned predictors (e.g., binding affinity) or property scores as rewards. This leads to limitations in optimizing for structural constraints or binding affinity, and inference-time reliance on repetitive docking is computationally expensive and suboptimal.
>
> * *Generation Process.*
> RL-based approaches generate molecules **autoregressively**, typically by sequentially attaching fragments or atoms. This introduces **cumulative errors** and makes it difficult to control **global molecular structure**. In contrast, structure-based continuous models can update all parameters simultaneously (non-autoregressively), enabling more effective global constraint enforcement.
>
> * *Generalization and Retraining Requirements.*
> RL-based methods generally require **retraining for each new target**, necessitating redefinition of reward/state/action spaces. In contrast, structure-based generative models (such as our proposed model) can be directly applied to arbitrary target structures after initial training, offering superior transferability and generalization.
>
> We also recognize that structure-based models rely on protein–ligand complex data and may **struggle with out-of-distribution generalization**, while **RL approaches can more flexibly** encode custom constraints or reward schemes. Recently, hybrid models that combine the advantages of both are being developed (e.g., TacoGFN, included in our experimental comparisons), and we acknowledge that integrating and benchmarking such approaches is an important future direction.
>
> We will ensure that these points (comparative strengths, limitations, and the need for integration) are explicitly added to the “Related Work” sections of our revised manuscript.
>
> ---
>
> ***2. Placement of Related Work, Abstract Overstatement, Line 306 Interpretation, Hyperlink Omissions, Labeling, and Figure References***
>
> We thank the reviewer for these detailed technical comments.
>
> We **fully agree** that the related work section should be incorporated into the main text, not only in the appendix. Due to initial page limitations, we placed it in the appendix, but we will move and expand the related work discussion in the main paper.
>
> Regarding the use of "overcomed" in the abstract and other overstatements (e.g., in line 306), we concur that such expressions are too strong and will replace them with more measured terms such as **“mitigated”** or **“addressed”** to better reflect the inherent limitations that remain.
>
> We will also cite the relevant prior work on the limitations of the SA score at the appropriate points, e.g., \[1,2]. We will also replace the phrase “interestingly” with a more precise statement referencing prior studies, such as “in line with previous findings,” and cite relevant literature instead.
>
>
> Concerning internal hyperlinks and references: our initial submission was prepared as a single LaTeX file where all links worked. Upon splitting the main text and appendix for submission, some hyperlinks (especially to the appendix) may have broken. If resubmission as a single file becomes possible, we will ensure all links are active. If the reviewer identifies any specific missing or erroneous links, please notify us and we will address them promptly.
>
> Regarding Figure 3, our manuscript explicitly refers to it at line 222 (on page 6). If there are additional referencing or labeling issues we have missed, we would greatly appreciate further clarification so we can correct them.
> #### [1]: Coley, C. et al,(2018). SCScore: synthetic complexity ~ . Journal of chemical information and modeling, 58(2), 252-261.
> #### [2]: Liu, C. H. et al, (2022). RetroGNN ~. Journal of Chemical Information and Modeling, 62(10), 2293-2300.
> ---
>
> ***3. Clarification of Figure 1***
>
> We apologize for any confusion caused by the insufficient explanation of Figure 1.
>
> This figure presents t-SNE projections of the **model-computed gradients** during sampling, grouped by model type. The axes represent t-SNE dimensions; color encodes timestep progression, with darker colors indicating later steps. The gradients depicted for the Bayesian Flow Network correspond to those described in Equations 14 and 15 of the main text: $\[\nabla\_{\mathbf{x}}\mathrm{log}p(\mathtt{l}\mid \mathbf{x}), \nabla\_{e\_\mathbf{x}}\mathrm{log}p(\mathtt{l}\mid e\_\mathbf{x})]\$ (with $\[\cdot, \cdot]\$ denoting concatenation).
>
> ---
>
> ***4. Models Used for Guidance***
>
> We apologize for omitting this from the main text (it was moved to the appendix due to length limits). The property predictors are based on the TargetDiff model architecture, trained to predict both binding affinity and synthetic accessibility [SA score] (see Appendix D.2). To account for predictive uncertainty, we adopted a Bayesian Neural Network (see Appendix D.1). The final guidance signal is the product of the two predicted properties, incorporated as a condition for generation.
>
> Selectivity guidance is implemented, as described in Appendix H.4, by assigning positive weights to on-target binding affinity and negative weights to off-target affinity in the guidance loss. We will add this information to the main experimental methods section to prevent any further confusion.
>
> ---
>
> ***5. Deeper Discussion of the SA Score***
>
> We appreciate the reviewer’s attention to this issue.
>
> We acknowledge that our presentation may have incorrectly implied a novel contribution in questioning the SA score, and that our citations of earlier work on its limitations were insufficient. As the reviewer noted, the inadequacy of the SA score as an absolute metric of synthetic feasibility is well documented (\[1,2], etc.), and we fully agree that it is now considered outdated in the SBDD field.
>
> Our intent, however, was to draw attention to the fact that **many recent studies on 3D molecule generation for SBDD continue to rely almost exclusively on the SA score** for practical synthesizability evaluation. Our motivation was to highlight this “lazy evaluation” trend and to advocate for broader adoption of more practical synthesis metrics (such as AiZynthFinder), especially since, to our knowledge, there has been little direct discussion of the reliability of the SA score in SBDD 3D generation.
>
> We will revise our manuscript to avoid the implication of novelty in this discussion, provide relevant and up-to-date references, and clarify that our goal is to objectively **inform the community (3D Mol generation in SBDD)** about the limitations of current evaluation practices. We will also stress the importance of incorporating metrics like AiZynthFinder for more robust synthetic accessibility assessment in SBDD applications.
>
> ---
>
> ***6. Clarification of the Role of the Proposed Model***
>
> Thank you for this insightful observation.
>
> Our adoption of SA score guidance is rooted in the clear trade-off between synthetic accessibility and binding affinity: molecules with higher binding affinity are often more complex and thus receive lower SA scores; simpler molecules tend to have less optimal binding. Our approach aims to balance these factors to generate candidates with both high affinity and practical (if not absolute) synthetic tractability.
>
> We agree with the reviewer that the SA score cannot explicitly optimize for custom chemistry or specific retrosynthetic routes, and that fully automated end-to-end design of synthesizable (considering AiZynthFinder), high-affinity hits remains a grand challenge. Our practical aim is to **reduce the incidence of implausible or synthetically impossible structures** among candidates and to provide a more realistic starting point for hit selection.
>
> In this response, the term “hit discovery” is interpreted not as exhaustive high-throughput screening, but as the generation of a set of optimized candidate molecules under multiple property constraints, conditional on the target protein. If this does not fully address your question, please feel free to let us know at any time. Furthermore, we fully recognize and will explicitly state that integrating more advanced retrosynthetic constraints and property optimization remains a crucial area for future work.
>
> ---
>
> If there are any misrepresentations or if you would like us to elaborate further on any topic, please let us know. We are committed to addressing all feedback thoroughly in our revised submission.

---

> > ### Author Response · Authors · 2025-08-07
> >
> > We kindly ask for your feedback on our responses.
> >
> > If there are any aspects that remain unresolved or if you require additional clarification, please let us know at your convenience.
> >
> > We sincerely appreciate the time and attention you have devoted to our work.

---

> ### Comment · Area_Chair_iawv · 2025-08-02
>
> Dear reviewer,
>
> Following the NeurIPS 2025 guidelines, I kindly encourage you to read and respond to author rebuttals as soon as possible. Please engage actively in the discussion with authors during the rebuttal process, update your review by filling in the "Final Justification" and acknowledge your involvement.
>
> Thank you,
> Your AC

---

> ### Comment · Area_Chair_iawv · 2025-08-05
>
> Dear reviewer,
>
> Following the NeurIPS 2025 guidelines, I kindly encourage you to read and respond to author rebuttals **as soon as possible** as the reviewer-author discussion period is coming to the end. Please engage actively in the discussion with authors during the rebuttal process, update your review by filling in the "Final Justification" and acknowledge your involvement.
>
> Thank you,
> Your AC

---

### Official Review · Reviewer_kfDY · 2025-07-08

**Clarity:** 4
**Significance:** 3
**Originality:** 4
**Rating:** 6
**Confidence:** 4

**Summary:**

The authors identify fundamental concerns with gradient-based guidance applied in sample space (e.g., when using diffusion models for structure-based drug discovery (SBDD)) and subsequently propose a Bayesian flow network for parameter space gradient-based guidance in SBDD. Empirical results for existing and new benchmarks are compelling and further suggest that future work (for all generative methods) is needed to improve the practical synthesizability of each method's generated molecules.

**Questions:**

Could the authors discuss what (practical or empirical) disadvantages they have found when developing Bayesian flow networks for molecular data? Are they harder to train than diffusion models, for example?

**Ethical Concerns:**

["NO or VERY MINOR ethics concerns only"]

**Final Justification:**

The authors have addressed my concerns regarding overstatements, additional benchmarking, and the scalability of Bayesian flow networks in general.

**Limitations:**

The authors currently only include one core SBDD benchmark dataset for general sample scoring (i.e., CrossDocked). Including another dataset for benchmarking would improve the rigor of the authors' experiments.

**Paper Formatting Concerns:**

I did not find any formatting concerns.

**Quality:**

3

**Strengths And Weaknesses:**

**Points of strength:**
1. The authors identify and propose a solution to the challenges of sample space gradient-based guidance in SBDD (e.g., with diffusion models).
2. The authors' proposed CByG network achieves strong empirical results for existing and new SBDD benchmarks.
3. The authors' writing and methodological rationale are clear and convincing.

**Points for improvement:**
1. Some of the authors' claims may not be entirely accurate. For example, in Section 6.2, the authors state that they introduce PoseBusters docking scores [1] into SBDD evaluation criteria. However, previous works in SBDD with diffusion models have already done so, as shown in works such as [2] (and such baselines are missing from the results in Table 1).
2. The authors should consider including a second benchmark dataset for SBDD sample scoring, such as Binding MOAD, PDBBind, or PLINDER. This would help to address a concern that CByG is simply overfitting to CrossDocked-like protein-ligand complexes.

**References:**

[1] Buttenschoen, M., Morris, G. M., & Deane, C. M. (2024). PoseBusters: AI-based docking methods fail to generate physically valid poses or generalise to novel sequences. Chemical Science, 15(9), 3130-3139.

[2] Morehead, A., & Cheng, J. (2024). Geometry-complete diffusion for 3D molecule generation and optimization. Communications Chemistry, 7(1), 150.

---

> ### Author Rebuttal · Authors · 2025-07-31
>
> We have carefully reviewed your valuable comments and, as such, have summarized below the areas for improvement that you highlighted.
> We hope you will understand that, due to space limitations, we have grouped similar questions into broader categories rather than addressing each in exhaustive detail.
> If any of our interpretations are inaccurate, please let us know.
>
> ***1. Overstatement regarding the introduction of PoseBusters and omission of GCDM as a baseline***
>
> We thank the reviewer for this precise feedback. The use of “introduce” in Section 6.2 was indeed inaccurate. We acknowledge that the PoseBusters docking score has already been utilized in prior SBDD works, such as [1]. We will revise the language to “adapt” or “incorporate” as appropriate.
>
> We also agree that omitting the diffusion-based baseline (e.g., GCDM-SBDD) from Table 1 is a significant oversight. As the generated molecule samples for GCDM-SBDD are not publicly available, we reproduced the model using the released weights and generated samples in our experimental environment. We evaluated these samples using PoseBusters under the same benchmarking protocol, and have added the results to Table 1 and the main text.
> #### HA refers to "High Affinity" , and PV refers to "PB-Valid"
> |Methods|SMINA Score|SMINA Dock|GNINA Score|GNINA Dock|VINA Score|VINA Dock|HA  |SA  |PV |
> |-|:-:|:-:|:-:|:-:|:-:|:-:|:-:|:-:|:-:|
> |Reference| -6.37| -7.92| -7.06| -7.61| -6.36| -7.45|-|0.74|95.0%|
> |CByG|-7.74|-9.61|-7.63|-8.33|-8.60|-9.16|93.6%|0.84|94.9%|
> |GCDM-SBDD|4.34|-6.55|4.34|-4.91|3.19|-5.79|17.1%|0.58|21.2%|
>
>
> For the reproduced GCDM-SBDD-cond(Ca) model, we found that the AutoDock vina scores could be reasonably replicated. In the original GCDM paper’s Table 4, the “Vina” column corresponds to the AutoDock(Dock) score in our results table.
> Moreover, Table 4 of the original GCDM paper reports PB-Valid scores as 38.1 / 15.7, where 38.1 excludes the ‘protein-ligand steric clashes’ check, and 15.7 includes all metrics. The original text in *GCDM* paper is as follows:
> > Additionally, the PB-Valid metric is defined as the percentage of generated molecules that pass all docking-relevant structural and chemical sanity checks proposed by ref. 23, with the validity ratio to the left (right) of each/denoting the percentage of valid molecules without (with) consideration of protein-ligand steric clashes.
>
> As our PB-Valid scores were computed with all PoseBusters checks included, comparing our results to the 15.7 benchmark is most appropriate, and our reproduction appears consistent in this context.
>
> By addressing these points, we have made our benchmarking and result comparisons more comprehensive and transparent. Thank you again for your meaningful feedback; we will continue to ensure that our evaluations remain consistent and reproducible.
>
> ***2. Experiments on additional benchmark sets***
>
> We greatly appreciate this important suggestion. To further assess the generalization of CByG and potential overfitting to the CrossDocked dataset, we are currently establishing an evaluation pipeline on the Binding MOAD benchmark.
>
> First, we would like to share the interim results that have been completed so far(The experimental setup for Binding MOAD was preprocessed as described in Schneuing et al. [2], resulting in a benchmark of 130 test proteins, with 100 ligands generated per target pocket. All other settings were kept consistent with those used in Table 1 of the submission.). We are doing our best to generate reliable results in a timely manner. As soon as all evaluations on the Binding MOAD benchmark are finished, we will transparently summarize and share the results and analysis before the discussion period ends.
>
> ||Vina Score ↓|Vina Dock ↓|
> |:-:|:-:|:-:|
> |Ref|-6.942|-8.491|
> |CByG|-8.415|-9.594|
>
> We ask for your patience as we finalize these experiments, and we are confident that these results will adequately address your concerns regarding generalizability.
>
> ***3. Empirical limitations of Bayesian Flow Networks***
>
> Thank you for this important question. Bayesian Flow Networks (BFNs) have demonstrated several practical advantages over diffusion-based generative models. For instance, the model converges stably with relatively few (e.g., 100) timesteps, and sampling can be performed efficiently in significantly fewer steps, resulting in lower computational costs and generation times.
>
> To date, we have not observed any significant increase in training difficulty or major practical limitations compared to diffusion models. In fact, the reduced timesteps and numerically stable closed-form update rules have contributed to overall training efficiency.
>
> However, we have identified a limitation regarding the propagation of multiple guidance signals. While controlling a single property (e.g., binding affinity) is effective, generating molecules that simultaneously satisfy multiple guidance properties (e.g., binding affinity and synthetic accessibility) remains challenging. This difficulty is particularly relevant in drug discovery scenarios, where multi-property optimization is essential. We recognize this as a crucial open challenge.
>
> In summary, the primary empirical limitation of BFNs lies in the optimization difficulty and fine-grained trade-off control when propagating multiple guidance signals. We plan to address these limitations in future work.
>
> #### [1] Morehead, A., & Cheng, J. (2024). Geometry-complete diffusion for 3D molecule generation and optimization. Communications Chemistry, 7(1), 150.
> #### [2] Schneuing, Arne, et al. "Structure-based drug design with equivariant diffusion models." Nature Computational Science 4.12 (2024): 899-909.
>
> ---
>
> Your comments have been instrumental in significantly improving the quality of our submission. We sincerely appreciate the time and effort you have devoted to reviewing our work.
>
> Should you have any additional inquiries, we would be happy to address them at your convenience.

---

> > ### Comment · Reviewer_kfDY · 2025-08-05
> > **Response to rebuttal**
> >
> > I have read the authors' rebuttal, and after careful consideration, I would like to raise my score to a 6.

---

> > > ### Author Response · Authors · 2025-08-06
> > > **Response by Authors**
> > >
> > > I am deeply grateful that you took the time to thoroughly review my answers. It has been a privilege to have such a substantive dialogue with a researcher of your standing.

---

> ### Comment · Area_Chair_iawv · 2025-08-02
>
> Dear reviewer,
>
> Following the NeurIPS 2025 guidelines, I kindly encourage you to read and respond to author rebuttals as soon as possible. Please engage actively in the discussion with authors during the rebuttal process, update your review by filling in the "Final Justification" and acknowledge your involvement.
>
> Thank you,
> Your AC

---

> ### Comment · Area_Chair_iawv · 2025-08-05
>
> Dear reviewer,
>
> Following the NeurIPS 2025 guidelines, I kindly encourage you to read and respond to author rebuttals **as soon as possible** as the reviewer-author discussion period is coming to the end. Please engage actively in the discussion with authors during the rebuttal process, update your review by filling in the "Final Justification" and acknowledge your involvement.
>
> Thank you,
> Your AC

---

### Official Review · Reviewer_dDy1 · 2025-07-16

**Clarity:** 3
**Significance:** 3
**Originality:** 3
**Rating:** 5
**Confidence:** 3

**Summary:**

This paper addresses a practical and important challenge in structure-based drug design: generating 3D molecules that are not only high-affinity binders, but also selective and synthesizable. The authors propose CByG, a novel framework that integrates Bayesian Flow Networks with gradient-based guidance. Using Tweedie’s formula, they reinterpret Bayesian updates as gradient steps, allowing controllable generation in parameter space. The paper also presents a well-rounded evaluation framework. Results show CByG outperforms existing methods across key metrics.

**Questions:**

1. The "Solved" rate in AiZynthFinder is below 50% despite high SA scores. Could the authors briefly discuss why this gap exists?
2. No runtime comparison is provided. Can the authors include training/inference time versus diffusion models?
3. Line 302 incorrectly refers to “Table 3” for AiZynthFinder results. Please correct.
4. Line 300 mentions six metrics, but only four are shown in Table 2. Please clarify.

**Ethical Concerns:**

["NO or VERY MINOR ethics concerns only"]

**Final Justification:**

The authors’ rebuttal offers a clear and nuanced explanation of the SA-score / AiZynthFinder relationship, promises careful corrections to citations and table labels, and provides useful runtime comparisons. With these points now clarified, I am happy to continue supporting acceptance.

**Limitations:**

Yes

**Paper Formatting Concerns:**

No major formatting issues.

**Quality:**

3

**Strengths And Weaknesses:**

Strengths
• Clear problem statement and strong motivation
• Strong theoretical link between Bayesian updates and gradient guidance
• Addresses practical needs beyond binding affinity by incorporating synthesizability and selectivity into both modeling and evaluation
• Clear writing and well-organized structure
Weaknesses
• The discussion of synthesizability results is somewhat limited.
• The paper does not provide analysis of training or inference efficiency.

---

> ### Author Rebuttal · Authors · 2025-07-31
>
> We have carefully reviewed the valuable comments you provided, and have summarized the main points for revision as follows.
>
> Please note that, due to space constraints, we have grouped similar issues into broader categories for clarity. If any of these points have been misunderstood, we would appreciate your feedback.
>
> 1. Discrepancy between SA score and AiZynthFinder
>
> 2. Citation errors regarding AiZynthFinder and label errors in Table 2
>
> 3. Experimental comparison of computational efficiency between our proposed model and diffusion-based approaches
>
> ---
>
> ***1. Discrepancy between SA score and AiZynthFinder “Solved” rate***
>
> The gap between the SA score and the AiZynthFinder “Solved” rate, as pointed out by the reviewer, stems from fundamental differences in the evaluation mechanisms and interpretations of these two metrics. Below, we provide a detailed explanation of these differences and how they should be interpreted:
>
> *SA (Synthetic Accessibility) Score*
>
> * Evaluation Method:
> The SA score statistically estimates the synthetic accessibility of a molecule by decomposing it into fragments and quantifying how frequently these fragments appear in large chemical databases (e.g., PubChem). Penalties are applied for structural features that increase complexity, such as stereocenters, spiro rings, and macrocycles. The score reflects “average synthetic accessibility” and is normalized between 0 (very difficult) and 1 (very easy).
>
> * Interpretation:
> The SA score is a statistical predictor of how easy a given molecular structure might be to synthesize chemically. It does not account for the actual existence of synthetic routes, available starting materials, or practical experimental constraints.
>
> *AiZynthFinder “Solved” Rate*
>
> * Evaluation Method:
> AiZynthFinder is a procedural retrosynthetic tool that uses Monte Carlo tree search and neural network-based reaction predictors to iteratively break down a target molecule into commercially available starting materials (stock molecules). A molecule is marked as “Solved” if a complete synthetic route is found; otherwise, it is “Unsolved.”
>
> * Interpretation:
> This metric reflects practical synthesizability given current reaction databases, available stocks, and model performance—in essence, whether a concrete, feasible synthesis route exists under realistic laboratory or industrial conditions.
>
> ***Key Causes for Discrepancy***
>
> 1. Stock Compound Limitations:
> The SA score is agnostic to the availability of stock chemicals, while AiZynthFinder requires the route to begin with a predefined set of purchasable starting materials. Thus, a simple molecule may still be “Unsolved” if a necessary intermediate is not available in the stock set.
>
> 2. Fragment Popularity vs. Reaction Feasibility:
> The SA score rewards frequent fragments but does not guarantee that the combination of those fragments is represented in known reactions. AiZynthFinder, on the other hand, demands feasible reaction pathways, so rare or unlearned connection strategies will result in “Unsolved” outcomes.
>
> 3. Multistep or Protection/Deprotection Complexity:
> Even seemingly simple molecules may require complex multistep syntheses or specialized strategies. The SA score does not reflect this procedural complexity, but AiZynthFinder will directly be affected by such challenges when searching for routes.
>
> SA score and AiZynthFinder “Solved” rate complement each other but fundamentally reflect different aspects of synthetic feasibility.
> A high SA score suggests structural simplicity and ease of synthesis, but a low “Solved” rate may still result if there are strict constraints imposed by stock compound availability or reaction databases.
> Thus, both metrics should be interpreted together, and reliance on either metric alone is insufficient for a comprehensive assessment of synthesizability.
>
> ---
>
> ***2. Citation errors and labeling issues in Table 2 for AiZynthFinder***
>
> We will immediately correct the cited reference errors to ensure there are no mistakes in the revised version.
> Additionally, we acknowledge the error in reporting six metrics in the text, and will ensure that such inaccuracies do not persist in the updated manuscript.
> Thank you for your careful attention to these details.
>
> ---
>
> ***3. Experimental comparison of computational runtime***
>
> We have measured the training and inference times of TargetDiff [1], the most widely cited diffusion-based model.
> Additionally, as described in Appendix E, TargetOpt is a guidance model based on TargetDiff that we developed in this study specifically for comparison with CByG , which implements diffusion-based guidance methods.
> Below, we provide a comparative table of computational speed between the diffusion-based models and our approach.
> ||Training Time (h)|Inference Time (h)|
> |:-:|:-:|:-:|
> |TargetDiff|11.2|0.95|
> |TargetOpt|13.4|1.67|
> |CByG|8.7|1.01|
>
> #### [1] Guan, Jiaqi, et al. "3D Equivariant Diffusion for Target-Aware Molecule Generation and Affinity Prediction." The Eleventh International Conference on Learning Representations.
>
> ---
>
> Your insights and constructive feedback have played a crucial role in enhancing the quality of our manuscript. We are deeply grateful for the careful attention and time you have invested in reviewing our work.
>
> If you require further information or would like to discuss any aspect in more detail, please let us know at any time.

---

> > ### Author Response · Authors · 2025-08-07
> >
> > We hope that our responses have addressed your concerns.
> >
> > If there are any points that remain unclear or if you require further clarification, please do not hesitate to let us know.
> >
> > We look forward to your feedback. Thank you very much for taking the time to review our work despite your busy schedule.

---

> > ### Comment · Reviewer_dDy1 · 2025-08-08
> >
> > Thank you for your detailed rebuttal. Your explanations are clear and effectively address the points I raised. The clarification on the discrepancy between the SA score and AiZynthFinder "Solved" rate, as well as the inclusion of runtime comparisons, greatly enhance my understanding of your method. I look forward to seeing the final version.

---

> > > ### Author Response · Authors · 2025-08-09
> > > **Request for the “Final Justification” section.**
> > >
> > > I am very glad to hear that your concerns have been addressed. I understand you must be extremely busy, but if possible, we would greatly appreciate it if you could kindly complete the *Final Justification* section. If it is not too much to ask, we would also be grateful if you might consider raising the final rating, even slightly.
> > >
> > > It has been a privilege to engage in such an insightful discussion with you regarding this study!

---

> ### Comment · Area_Chair_iawv · 2025-08-02
>
> Dear reviewer,
>
> Following the NeurIPS 2025 guidelines, I kindly encourage you to read and respond to author rebuttals as soon as possible. Please engage actively in the discussion with authors during the rebuttal process, update your review by filling in the "Final Justification" and acknowledge your involvement.
>
> Thank you,
> Your AC

---

> ### Comment · Area_Chair_iawv · 2025-08-05
>
> Dear reviewer,
>
> Following the NeurIPS 2025 guidelines, I kindly encourage you to read and respond to author rebuttals **as soon as possible** as the reviewer-author discussion period is coming to the end. Please engage actively in the discussion with authors during the rebuttal process, update your review by filling in the "Final Justification" and acknowledge your involvement.
>
> Thank you,
> Your AC

---

### Note · Authors · 2025-08-13

We are truly grateful for the opportunity to have our submission evaluated from such diverse and insightful perspectives. Our work was reviewed by five distinguished reviewers, whose comments can be summarized as follows:

- Differences between SA scores and AiZynthFinder evaluation metrics
- Computational efficiency of the proposed model
- Inclusion of additional benchmarks and comparison models
- Detailed explanation of categorical variable modeling
- Reliability of the PB-valid evaluation metric
- Broader range of ablation studies
- Theoretical justification of the proposed framework

Each of these comments has had a profoundly positive impact on the refinement of our manuscript. In response, we have conducted additional experiments, provided further proofs, and offered more extensive experimental interpretations to address the reviewers’ concerns. Through continuous dialogue during the discussion phase, it is my understanding that nearly all reviewers expressed agreement with our arguments (reviewer aH1P did not respond to our rebuttal. We were particularly looking forward to a discussion with them, as their initial comments were highly insightful and thought-provoking, and we were disappointed to miss the opportunity for further engagement.).

In brief, our paper addresses 3D molecular generation in the context of structure-based drug design, aiming to generate candidate molecules that satisfy multiple desirable properties. To this end, we reinterpret the Bayesian Flow Network through a gradient-based sampling perspective grounded in a solid theoretical foundation. We then apply guidance to this redefined framework, and experimentally demonstrate that it can generate molecules meeting diverse property requirements. Furthermore, by framing the problem definition and conducting extensive experiments from multiple viewpoints, we provide strong evidence for the superior performance of the proposed model.

We believe that our findings will make a meaningful contribution to the field of structure-based drug design and foster a positive feedback cycle in AI-based drug discovery. We sincerely thank the reviewers and the area chairs for dedicating their valuable time to evaluating our submission, and we hope our work will merit acceptance.

---

### Decision · Program_Chairs · 2025-09-17

**Decision:**

Accept (poster)

**Comment:**

This paper focuses on the structure-based drug design problem and has two main contributions:
1. extend and advocate Bayesian flow networks for structure-based drug design
2. call for practical evaluation for structure-based drug design

The rethinking section J is interesting, in particular J.1 and J.2. However, I could see it stronger as main motivation in the main paper (just a suggestion), this relates to the question reviewers asked about the advantage of BFN over diffusion models. It is argued in the paper J.1 can be solved with BFN, but it seems J.2 is not? J.3 missed citing many relevant efforts from the community, see below.

The reviewers had several questions about (1) claims and related work discussions, (2) additional evaluation on the proposed methods, and (3) theoretical justification of the proposed framework. After rebuttal, all the concerns were addressed and all reviewers vote for acceptance of this paper.

In addition to what several reviewers have mentioned, I would like to point out the study to consider synthesizability [2], selectivity [3] and pharmacophoric constraint [1] in drug design have been extensively studied in the literature, just list a few examples.

I encourage the authors to integrate the reviewers' comments into the revision, especially the discussion to related work and importance missing details, e.g. the experiment setup for selectivity problem.

[1] Adams, K., Abeywardane, K., Fromer, J.C. and Coley, C.W., 2025. ShEPhERD: Diffusing shape, electrostatics, and pharmacophores for bioisosteric drug design. In ICLR 2025.

[2] Liu, S., Zhang, D., Tu, Z., Dai, H. and Liu, P., 2024. Evaluating Molecule Synthesizability via Retrosynthetic Planning and Reaction Prediction. arXiv preprint arXiv:2411.08306.

[3] Schneuing, A., Harris, C., Du, Y., Didi, K., Jamasb, A., Igashov, I., Du, W., Gomes, C., Blundell, T.L., Lio, P. and Welling, M., 2024. Structure-based drug design with equivariant diffusion models. Nature Computational Science, 4(12), pp.899-909.